# Human multilineage pro-epicardium/ foregut organoids support the development of an epicardium/myocardium organoid

Mariana A. Branco [1,2], Tiago P. Dias [1,2], Joaquim M. S. Cabral[1,2], Perpetua Pinto-do-Ó [3,4] & Maria Margarida Diogo [1,2] ✉

The epicardium, the outer epithelial layer that covers the myocardium, derives from a transient organ known as pro-epicardium, crucial during heart organogenesis. The pro-epicardium develops from lateral plate mesoderm progenitors, next to septum transversum mesenchyme, a structure deeply involved in liver embryogenesis. Here we describe a self-organized human multilineage organoid that recreates the co-emergence of pro-epicardium, septum transversum mesenchyme and liver bud. Additionally, we study the impact of WNT, BMP and retinoic acid signaling modulation on multilineage organoid specification. By co-culturing these organoids with cardiomyocyte aggregates, we generated a self-organized heart organoid comprising an epicardium-like layer that fully surrounds a myocardium-like tissue. These heart organoids recapitulate the impact of epicardial cells on promoting cardiomyocyte proliferation and structural and functional maturation. Therefore, the human heart organoids described herein, open the path to advancing knowledge on how myocardium-epicardium interaction progresses during heart organogenesis in healthy or diseased settings.

Human pluripotent stem cells (hPSCs) and their capacity to differentiate into different cell types have been intensely explored in the past years envisaging the development of in vitro models of a broad range of tissues. Specifically, recapitulation of embryonic development in vitro through the generation of hPSC-derived organoids has succeeded in several tissues, including brain, intestine, and kidney. Nevertheless, the recreation of early stages of heart development in vitro has been particularly challenging, with only few very recent studies recapitulating at some extent the structural organization and the multicellularity involved in heart organogenesis[1–3].

One important structure crucial for embryonic development of the heart is the pro-epicardium (PE) organ. PE is a transient structure derived from the lateral plate mesoderm (LPM) that forms at the ventro-caudal base of the developing heart tube of all vertebrate species, specifically at approximately embryonic day 9.0 (E9.0) in the mouse[4] and at fourth week post-conception in the human embryo[5]. This transient organ comprises the epicardial progenitors that migrate towards the developing heart to form the outer epithelial layer, the epicardium, that covers the myocardium[6]. The role of the epicardium during embryonic heart development is well described, constituting not only the main source of cardiac fibroblasts that support and contribute to the formation of a functional connective tissue in the myocardium, but also to coronary smooth muscle cells, critical for coronary vasculature development[6–12]. Moreover, apart from serving as a progenitor cell source of cardiac cells, epicardium is also described to release paracrine signals essential to induce

[1]iBB – Institute for Bioengineering and Biosciences and Department of Bioengineering, Instituto Superior Técnico, Universidade de Lisboa, Lisboa, Portugal. [2]Associate Laboratory i4HB — Institute for Health and Bioeconomy at Instituto Superior Técnico, Universidade de Lisboa, Lisboa, Portugal. [3]Stem Cells in Regenerative Biology and Repair group, Instituto Nacional de Engenharia Biomédica (INEB), Instituto de Investigação e Inovação em Saúde (i3S), Universidade do Porto, Porto, Portugal. [4]Instituto de Ciências Biomédicas Abel Salazar, Universidade do Porto, Porto, Portugal. ✉e-mail: margarida.diogo@tecnico.ulisboa.pt

proliferation, compaction, and maturation of the developing myocardium[13–17].

The PE develops in close proximity with the septum transversum mesenchyme (STM), and morphologically PE cells appear as protrusions extending from the STM region, as observed in the mouse model[12]. However, discrimination between PE and STM populations has been challenging given the lack of molecular boundaries, with different markers, including WT1, TBX18, and TCF21 also co-expressed by the STM cells[12]. At the same stage of PE embryonic development, particularly by E8.75 in mouse, hepatoblasts are specified from the hepatic endoderm and by E9.5 signals from the STM induce their transition into a pseudostratified columnar epithelium that invades the STM[18]. Subsequently, the STM gives rise to part of the liver mesenchyme, the non-parenchymal counterpart of the liver.

Although lineage tracing studies in animal models have given valuable insights[4], the embryonic development path of the PE is still poorly disclosed. Generation of PE cells from hPSC in 2D environment has opened the possibility to decipher some of the cues involved in PE specification in humans[19–22]. However, the recreation of the transient PE embryonic structure in close association with STM and posterior foregut/liver bud has not been achieved yet in vitro. Therefore, the motivation of the present work was to recapitulate in vitro the embryonic development of the human PE through modulation of hPSCs-derived LPM progenitors in a more in vivo-like 3D environment.

Herein we describe a simple and robust platform to generate a self-organized multilineage hPSC-derived organoid that recreates an early embryonic structure comprising both pro-epicardium/septum transversum populations and posterior foregut/hepatic primordium, with three clearly identifiable main regions: (1) an outer layer of WT1+/LHX2-/+low cells identified as PE-like cells, (2) an epithelial-like structure comprising CDX2/AFP positive endoderm gut tube cells and AFP/HNF4 positive hepatoblast-like cells, and (3) an intermediate population between the outer layer and the epithelium that is WT1+low/LHX2+, identified as STM-like cells. We additionally prove the functionality of the developed PE/STM cells through the establishment of a co-culture model with CMs in a 3D environment. From this co-culture strategy resulted an epicardium-myocardium heart organoid (EMO) that self-organizes in a complex structure, comprising a well-defined WT1+ epicardial-like layer that completely surrounds a cTnT+ myocardium-like tissue. Furthermore, we demonstrate how LPM progenitors can be modulated to privilege PE/STM specification at the expense of CMs, and present results indicating retinoic acid (RA) signaling as a potentially important stimulus to induce PE/STM mesoderm specification from LPM.

## Results

### WNT, BMP, and RA activation in LPM progenitors yields WT1+ cells

Although the precise mesodermal origin of PE/STM remains unclear, it is known that the PE is specified from LPM progenitors, from where myocardium is also derived. As a starting point to develop a culture platform for PE cells specification from hPSCs, we adapted our previously reported 3D CM differentiation protocol[23], which is based solely on the temporal modulation of WNT signaling (Fig. 1a), to promote PE specification (PE protocol) at the expense of CM differentiation (CM protocol). According to the literature, BMP4 and WNT signals have been described to be involved in PE cells specification[19,20]. Additionally, RA signal has been also reported to be critical to generate epicardial cells in vitro[19,24].

Therefore, the combined supplementation of BMP4, CHIR, and RA from day 5 (D5) progenitor cells until D7 of differentiation (PE induction period) was performed, which allowed the generation of 73.7 ± 1% of WT1+ cells and completely abolished the generation of cTnT+ CMs (Fig. 1b and Supplementary Fig. 1). To confirm lineage commitment,

the dynamic changes in expression of important genes involved in CMs and PE specification were assessed (Fig. 1c and Supplementary Fig. 2a). The results showed a reduction in the expression levels of the CM progenitor markers NKX2.5 and ISL1, right after PE induction, and a clear up-regulation of the PE markers WT1, TBX18, ALDH1A2, and TCF21. TBX5 and GATA4, which are known transcription factors involved in proper cardiogenic lineage specification, have been also reported to be involved in the development and expressed in PE and STM[25–27]. Accordingly, in the present model, it was observed an increased expression of TBX5 over time in both protocols, however with higher expression levels being observed in the PE protocol compared to the CM protocol. GATA4 expression increased significantly in the common days of both protocols (until D5) and at D11 it presents a higher expression level in PE, compared with CM aggregates. The robustness of the protocol was confirmed with two additional cell lines (hiPSC-F002.1A.13 and hESC-H9 lines; Supplementary Fig. 2b), revealing similar results for WT1+ cell generation efficiency. In addition, we further demonstrated the potential of the WT1+ cells to undergo EMT and differentiate into smooth muscle and fibroblast-like cells (Supplementary Fig. 2c, d).

### PE aggregates recreate pro-epicardium in vivo-like environment

Brightfield (BF) images of PE aggregates at D11 of differentiation, which present a roundness value over 0.7, suggested the presence of lumen-like structures, normally located in the center of the aggregates (Supplementary Fig. 3a, b), which were observed in 87 ± 3% of the aggregates (Supplementary Fig. 3a–c). In order to understand in more detail the cellular composition and assess the presence of any structural organization, sections of PE aggregates generated from the three different hPSC lines tested were analyzed by immunofluorescence (IF) staining (Fig. 1d–f and Supplementary Fig. 3d, e, i, k, and Supplementary Fig. 4). The WT1+ cells were clearly identified as the main population (Fig. 1dI, II) and the presence of lumens lined by an E-cadherin positive and WT1 negative epithelium (Fig. 1dI, II and Supplementary Fig. 3d) were also observed. These epithelial structures were anchored to a basement membrane composed by the ECM proteins laminin, collagen IV and fibronectin (Fig. 1dII, III) and surrounded by CD31+/WT1- endothelial-like cells organized in a concentric pattern (Fig. 1dI). Apparently, these cells do not have an endocardium-like phenotype since they do not co-stain with NKX2.5 (Supplementary Fig. 3e), as previously referred by others[2]. Interestingly, VEGF signaling activation from D7 until D11 of differentiation led to a significant increase in the percentage of CD31+ cells, from 2.1 ± 0.1% to 10.7 ± 1.4%, at D11 of differentiation, without compromising the WT1 positive cell population (Supplementary Fig. 3f, g), suggesting that D7 cell population of PE differentiation comprises progenitors for vascular-like cells that are still responsive to VEGF stimulation.

Since the intensity of WT1 staining was not homogeneous throughout the entire aggregate, with more pronounced staining at the outermost layers and decreasing towards the epithelial structures (Fig. 1dI, II and Supplementary Fig. 3h), we hypothesized that these WT1+ cells could comprise different subpopulations. A recent report from Lupu and colleagues[12] highlighted the similar transcriptomic profiles of PE and STM and divided the PE/STM cluster into three subpopulations, the PE, the STM and a subcluster that was described as being PE cells from STM, which present a different transcriptional profile. Lhx2 expression was observed to be restricted to STM and PE/STM clusters, whereas Wt1 was expressed in all the three subclusters, with a more pronounced expression level in the PE cluster, decreasing in PE/STM and displaying even lower expression in the STM group of cells.

Considering these data, we confirmed the presence of LHX2+ cells within D11 aggregates, with a more pronounced staining in cells surrounding the epithelial structure, and absence or decreased expression towards the periphery of the aggregates, where WT1 staining was

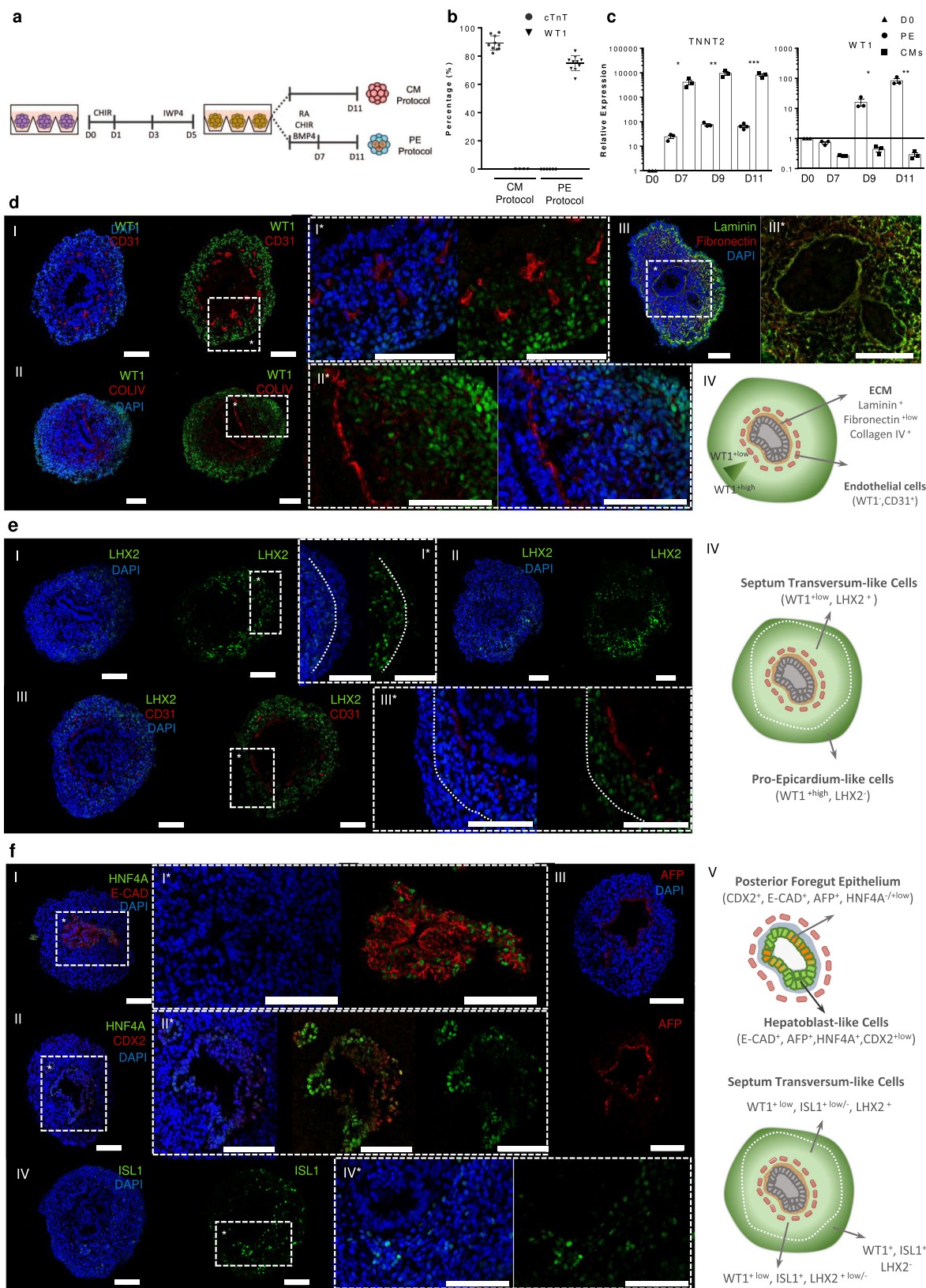

more pronounced (Fig. 1e and Supplementary Fig. 3h). This observation revealed that WT1$^+$ cells comprise, at least, two different populations, a PE-like cell population localized specifically at the periphery of the aggregates that is WT1$^+$/LHX2$^{-/+low}$ and a STM-like population localized between the PE area and the epithelial lumens, which is WT1$^{+low}$/LHX2$^+$ (Supplementary Fig. 3i), recapitulating what was

described in vivo in mouse[12]. Gene expression analysis confirmed also significantly increased expression of the specific PE cell marker *UPK3B*, and the STM markers *LHX2* and *HLX1*, in D11 PE aggregates compared with the CM differentiation protocol (Supplementary Fig. 3j).

Knowing that the STM develops in close proximity with the posterior foregut/hepatic epithelium, and it has been described that the

**Fig. 1 | PE/STM/PFH organoids recapitulate pro-epicardium, septum transversum, posterior foregut, and hepatic epithelium bud embryonic structure.**
**a** Schematic protocol for PE cell differentiation from hPSC in comparison with the CM differentiation platform. **b**, Percentage of WT1 and cTnT positive cells after 11 days of differentiation for both PE and CM differentiation protocols. Data are represented as mean ± SEM of n = 10 independent experiments. **c**, Expression profile of *TNNT2* and *WT1* genes from D0 up to D11 of differentiation for PE and CM differentiation conditions. Values are normalized to *GAPDH* and to D0 of differentiation. Data are represented as mean ± SEM of n = 3 independent experiments.

Exact p-values: TNNT2 D7 p = 0.0144, D9 p = 0.0043, D11 p = 0.0002; WT1 D7 p = 0.0023, D9 p = 0.0169, D11 p = 0.0040. *p < 0.05, ***p < 0.001, and ****p < 0.0001. **d–f** Representative IF staining of D11 PE/STM/PFH organoid sections, highlighting the WT1 positive cell population, and the presence of WT1⁻ epithelium-like structures surrounded by CD31 endothelial-like cells and ECM proteins (**d**), the LHX2 cell population more pronounced near the epithelial structure and with lower expression levels near the aggregate edges (**e**), and the expression of the posterior endoderm gut marker CDX2, the hepatoblast markers HNF4A and AFP within the epithelial structure, and the ISL1 positive cell population (**f**). Scale bars, 100 μm.

nuclear staining of WT1 is weaker in the STM near the foregut endoderm[28], we hypothesized that the epithelial structures observed within D11 aggregates could represent those cells. IF staining for the posterior endoderm marker CDX2 and for the hepatic markers HFN4A and AFP[29–31] showed specific staining within the epithelial structures (Fig. 1fI, III). In particular, it was observed that the majority of the cells within the epithelium were AFP⁺ and HNF4⁻/⁺low/CDX2⁺ and the cells of the epithelium that form protuberances towards the STM area were AFP⁺ and HNF4A⁺/CDX2⁺. These data corroborates the hypothesis that the epithelial structures in D11 PE aggregates could represent an hepatic epithelium, from where hepatoblasts are specified and invade the surrounding STM in a transient structure known as liver bud[18].

Since D11 PE aggregates seem to recapitulate the interaction between STM and posterior foregut/hepatic diverticulum development, we asked if the present aggregates also contain mesenchymal derived cells that have been described to be specified from STM and be involved in the non-parenchymal composition of the liver. Recently, Lotto and colleagues[30] identified at E9.5 a subcluster of STM involved in liver embryogenesis that expresses *Isl1* and low levels of *Wt1*. Surprisingly, the aggregates also show regions with strong ISL1 expression, in the surrounding areas of the epithelial structures (Fig. 1fIV), which seem to not overlap with LHX2 staining (Supplementary Fig. 3k). All of the aforementioned observations were further corroborated for the two additional hPSC lines analyzed (Supplementary Fig. 4). Together, these findings indicate that the developed PE aggregates represent a multilineage organoid that seems to recreate the early transient embryonic structure of PE/STM and posterior foregut/hepatic diverticulum, henceforth designated as PE/STM/PFH organoids.

## Biphasic WNT modulation generates endoderm and mesoderm progenitors

It has been reported that the precise control of aggregate size is important for the controlled and reproducible induction of organoids and in particular for cardiac organoid patterning[2]. Considering this, for both CM and PE protocols, we fixed the aggregate diameter at D0 in 300−320 μm[23], through manipulation of cell seeding density, and observed that at D5 of differentiation the aggregates already increased to an average diameter of 401 ± 6 μm (Fig. 2a, b). This may promote the generation of gradients of soluble factors inside the aggregates and consequently the generation of different subpopulations of progenitor cells that culminate in the emergence of the PE/STM/PFH multilineage organoids. To further confirm this hypothesis, we analyzed the cellular content of organoids at D5 of differentiation for mesoderm and endoderm progenitor cells.

We identified different subpopulations of progenitor cells by flow cytometry analysis for CXCR4, a mesendodermal marker, and c-KIT, a marker expressed in endoderm and vascular progenitor cells but not in mesoderm progenitors, combined with additional mesoderm markers KDR and PDGFRA (Fig. 2c). At D5 of differentiation 24 ± 4% of the cells were CXCR4⁺/c-KIT⁻/KDR⁻/PDGFRA⁻, meaning that, at this time point of differentiation, definitive endoderm (DE) progenitors are present in the aggregates[32–34]. Additionally, ±5% of the analyzed cells corresponded to CXCR4⁻/c-KIT⁺/KDR⁺high vascular progenitor cells[35,36]. The remaining cells were CXCR4⁺/c-KIT⁻/KDR⁺low/PDGFRA⁺, representing a

mesoderm affiliated population[37], accounting for 80 ± 4% of the total cells.

Further analysis of D5 progenitor cell aggregates by IF staining revealed the presence of mesoderm (ISL1⁺), endoderm (SOX17⁺), and vascular cells (CD34⁺; Fig. 2d). Interestingly, SOX17⁺ cells self-organize as a continuous layer, which involves the ISL1⁺ mesoderm progenitor cells, similarly to what is observed during embryonic development when splanchnic mesoderm is in close association with the surrounding endoderm layer[38]. Together, these findings support the hypothesis that PE/STM/PFH organoids derive from a population comprising mesendoderm and endoderm progenitors generated by temporal modulation of WNT signaling under 3D conditions.

## WNT and RA are crucial for PE/STM/PFH organoid development

We next analyzed the impact of WNT, BMP4 and RA modulation during PE induction period (from D5-D7) on the generation of PE/STM/PFH organoids. Modulation of WNT signaling was identified as the main driver for WT1⁺ cell population specification, as clearly demonstrated by the significant decrease in WT1 positive cells with decreasing concentrations of CHIR, from 73.7 ± 1% of WT1⁺ cells, with 3 μM CHIR, to 20 ± 3% WT1⁺ cells, without CHIR supplementation, while maintaining BMP4 and RA activation (Fig. 3a). On the other hand, when only WNT signaling was activated, it was not detected a statistically significant reduction of WT1⁺ cells (Fig. 3b). However, under this condition, we observed the generation of organoids with a different structure compared with PE/STM/PFH organoids, with the emergence of less dense cystic regions that grow with time in culture (Fig. 3c). IF imaging revealed that these cavity-like regions were lined by a CD31⁺ endothelium and AFP⁺/HNF4A⁺ hepatoblast-like cells (Fig. 3d and Supplementary Fig. 5a), which seem to emerge from the CDX2⁺ epithelial structure, resembling the initial stages of embryonic liver development.

These organoids presented also denser regions that were LHX2⁺ and WT1⁺, which may represent the primordium for liver mesenchyme (Fig. 3d and Supplementary Fig. 5a). Interestingly, the specification of these liver organoids was even more evident when a combined activation of WNT and BMP4 was performed (Supplementary Fig. 5b), suggesting that BMP4 signal could have a positive effect in hepatoblast-like cells specification. In fact, the positive role of BMP4 in liver bud specification from posterior foregut has been largely reported in the literature[29,39]. Additionally, these data also suggests that RA activation, at this specific stage of differentiation, could be interfering with further commitment of hepatoblast-like cells from the posterior foregut epithelium present in PE/STM/PFH organoids. It has been reported in mice that liver specification and initial liver bud induction can occur, at least, in Raldh2 mutants, meaning that RA signal at this stage of development is not critical[40]. We additionally observed that the combined activation of WNT and RA signaling pathways was sufficient to generate organoids that resemble the PE/STM/PFH ones, showing that BMP4 supplementation has low impact on the organoid's specification (Fig. 3b and Supplementary Fig. 5c–e).

Removal of WNT signaling activation during the PE induction period, corresponding to RA, RA + BMP4 and BMP4 conditions, opened the path for the specification of cTnT positive cells (Fig. 3e). However, the activation of only BMP4 signaling does not allow the

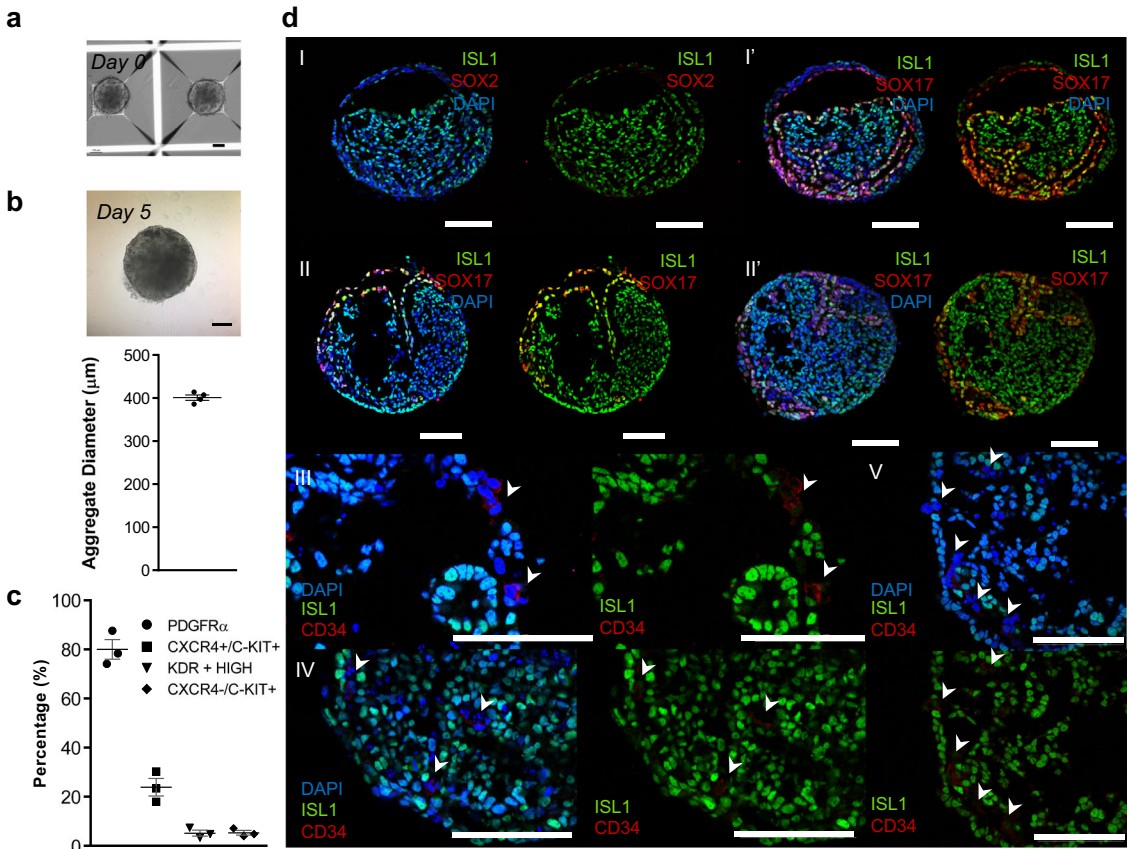

**Fig. 2 | Characterization of D5 hPSC-derived aggregates reveals a cell population that comprises endoderm and mesoderm progenitor cells. a** BF images of aggregates at D0 of differentiation. **b** BF images (top) and diameter distribution of aggregates at D5 of differentiation (μm) (bottom). Data are represented as mean ± SEM of *n* = 4 independent experiments. **c** Flow cytometry quantification of the percentage of DE progenitors (CXCR4$^+$/c-KIT$^+$), mesoderm progenitors (PDGFRα$^+$) and vascular progenitor (KDR$^{high}$ and CXCR4$^-$/c-KIT$^+$) cells. Data are represented as mean ± SEM of *n* = 3 independent experiments. **d** Representative IF staining of D5 cardiac aggregate sections. Mesoderm, endoderm and vascular progenitors were stained for ISL1, SOX17, and CD34, respectively. Scale bars, 100 μm. White arrows represent CD34 positive cells. Source data are provided as a Source Data file.

generation of WT1$^+$ cells and induce CM organoids specification (Supplementary Fig. 6a, b). In both RA and RA + BMP4 conditions were registered similar percentages of WT1$^+$ cells (23.9 ± 3% and 19.9 ± 3% for RA and RA + BMP4, respectively), and the co-generation of cTnT$^+$ cells (9.2 ± 1% and 15.3 ± 5% for RA and RA + BMP4 conditions, respectively) at D11 of differentiation (Fig. 3e). IF images of organoids at D11 and D17, revealed also a very similar structure of the organoids generated from both conditions (Fig. 3f, g). Normally, as it was possible to observe at D17, these organoids present a subpopulation of WT1 positive cells, which do not co-stain with LHX2, lining the periphery of the organoids, including the cTnT positive region and the LHX2$^+$/WT1$^-$ STM-like area (Fig. 3g and Supplementary Fig. 6c, d).

Additionally, it was also observed the presence of an ISL1$^+$ progenitor cell population that overlaps with cTnT$^+$/NKX2.5$^+$ CMs. In fact, PCR analysis revealed increased expression of ISL1 and LHX2 transcripts in RA + BMP4 condition compared with control CHIR + RA + BMP4 (PE/STM/PFH organoid; Supplementary Fig. 6e). In both RA and RA + BMP4 conditions, the percentage of cTnT increased from D11 to D17 of the differentiation (Fig. 3e), which may indicate CM proliferation or later specification of ISL1$^+$ progenitor cells into CMs. Activation of the RA signaling pathway at the cardiac progenitor stage, similarly to what has been largely reported for atrial CMs specification in monolayer differentiation protocols, was recently applied in a 3D environment to generate atrial CM organoids[41]. This resembles the condition where we only use RA supplementation during the PE induction period. We thus hypothesize that the CMs obtained in RA and RA + BMP4

conditions may have a more restricted atrial phenotype compared with that obtained for the control CM protocol.

Altogether, these data demonstrate that (1) WNT and RA are critical stimuli for PE/STM/PFH organoids specification, (2) removal of RA pathway activation results in the specification of liver bud organoids, and (3) the removal of WNT signaling activation between D5 and D7 of differentiation allows the co-emergence of CMs and PE/STM cell populations.

## Co-culture of PE/STM/PFH cells with CMs in 3D recreates epicardium-myocardium interaction

In an attempt to recreate epicardium-myocardium interaction and prove the functionality of PE/STM/PFH organoids, we established a co-culture model between CMs and PE/STM/PFH cells, a system hereafter designated as epicardium-myocardium organoids (EMOs). We then asked whether this system recapitulates in vitro the interaction between myocardium and pro-epicardium/epicardium during embryonic heart development.

PE/STM/PFH organoids and CM aggregates (Supplementary Fig. 7a) were dissociated and combined at 90%:10% ratio (CMs:PE/STM/PFH; Fig. 4a), which was decided based on previous studies[1,12,42,43]. For that, both PE/STM/PFH organoids and CM aggregates, were dissociated at D11 of differentiation since, at this time point, α4-integrin (ITGA4) expression, a cell adhesion protein previously described as being important for PE cell attachment into the myocardium[44,45], is maximized in PE/STM/PFH organoids (Supplementary Fig. 7b). Also, as control conditions, CM aggregates and PE/STM/PFH organoids were

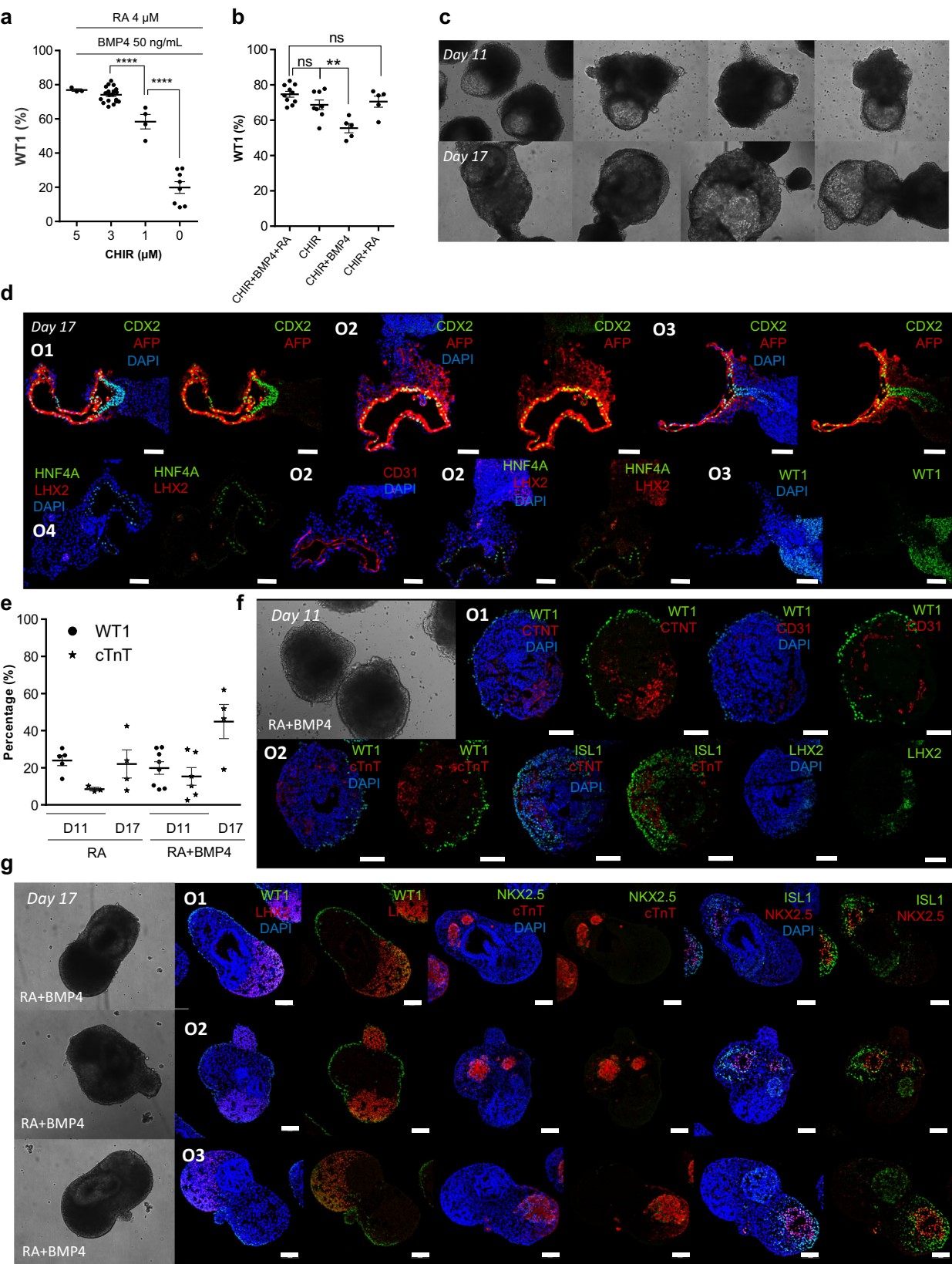

dissociated and reaggregated separately at the time of co-culture establishment (Supplementary Fig. 7c, d).

BF images of EMOs throughout the co-culture period and 15 days post-reaggregation (Fig. 4b and Supplementary Fig. 7e) showed that after 24 h of co-culture, the two origin cell-types were indistinguishable within EMOs, at microscopic examination. However, at D3, it was

observed the establishment of two distinct regions, demonstrating a self-organization behavior. Over time in culture, a distinct layer of cells was observed surrounding the contracting core of the EMOs (Supplementary Fig. 7e), with these organoids reaching an average diameter of $681 \pm 42\,\mu m$ after 15 days of co-culture (Supplementary Fig. 7f). Although within a successful biological replicate the

**Fig. 3 | WNT and RA are the crucial signals for the development of PE/STM/PFH organoids. a** Study of the impact of CHIR concentration variation from 0 – 5 μM from D5 to D7 of differentiation, on the percentage of WT1 positive cells at D11 PE/STM/PFH organoids, assessed by flow cytometry analysis. RA concentration was fixed at 4 μM and BMP4 concentration at 50 ng/mL, for all the tested conditions. Data are represented as mean ± SEM of $n = 8$ (0 μM CHIR), $n = 4$ (1 μM CHIR), $n = 19$ (3 μM CHIR), and $n = 3$ (5 μM CHIR) independent experiments. Exact $p$-values: 3 vs 1 μM CHIR $p < 0.0001$; 1 vs 0 μM CHIR $p < 0.0001$. **b** Flow cytometry analysis of WT1 positive cells at D11 PE/STM/PFH organoids for "CHIR", "CHIR + BMP4", "CHIR + RA" and "CHIR + BMP4 + RA" conditions. Data are represented as mean ± SEM of $n = 10$ (CHIR + BMP4 + RA), $n = 8$ (CHIR) and $n = 5$ (CHIR + BMP4, CHIR + RA) independent experiments. Exact $p$-values: CHIR vs CHIR + BMP4 $p = 0.0072$. **c, d** Representative BF images of D11 and D17 organoids (**c**) and IF images of D17 organoids (**d**) obtained from "CHIR" condition, highlighting the cystic hepatoblast-like structures. **e** Flow cytometry analysis of WT1⁺ and cTnT⁺ cells at D11 and D17 of differentiation for "RA" and "RA + BMP4" conditions. Data are represented as mean ± SEM of $n = 4$ (WT1⁺ in D11 RA, cTnT⁺ in D17 RA and D17 RA⁺BMP4), $n = 3$ (cTnT⁺ D11 RA), $n = 6$ (cTnT⁺ D11 RA + BMP4), and $n = 8$ (WT1⁺ D11 RA + BMP4) independent experiments. **f, g** Representative IF images of D11 (**f**) and D17 (**g**) organoids obtained for "RA + BMP4" condition. Scale bars, 100 μm. ns not statistically significant, $^{**}p < 0.01$, and $^{****}p < 0.0001$. Ox nomenclature identifies different organoids obtained from at least three independent experiments.

proportion of EMOs that present a layer of cells covering the contracting core is almost 100%, in Supplementary Fig. 7g we highlight one representative EMO that failed to establish the mentioned structure.

After 15 days of co-culture, IF and transmission electron microscopy (TEM) analysis were performed in EMOs, using as control age-matched CM aggregates. IF revealed that EMOs are composed by a WT1⁺ epicardial-like layer (EL) surrounding a cTnT⁺ myocardium-like zone (MZ), which was not possible to observe in control CM aggregates, where no WT1 positive cells were present (Fig. 4c and Supplementary Fig. 7h). Specifically regarding CMs, we observed that cTnT⁺ cells, near the EL, show stronger staining for cTnT and lower density of nuclei (Supplementary Fig. 7i), which suggests a more compact fiber content, and CMs appeared also more aligned and following the curvature of the EMOs.

Using Ki-67 and NKX2.5 to identify proliferative CMs, we showed that the more compact CM area, contiguous to the EL, was enriched in proliferative CMs (NKX2.5⁺/Ki-67⁺; Fig. 4d and Supplementary Fig. 8a, b), whereas these cells were almost absent towards the core of EMOs. In control CMs aggregates, it was also observed proliferative CMs, however without presenting a specific organization pattern (Fig. 4d). Interestingly, towards the center of the MZ in EMOs, the intensity of cTnT staining was lower and it was observed the presence of nucleus with a decreased NKX2.5 expression intensity or even that were NKX2.5 negative (Supplementary Fig. 8a). Nevertheless, staining for the apoptotic marker caspase, did not reveal cell death at the center of the organoids (Supplementary Fig. 8c).

In search for other cells that could be present within the MZ, we observed the presence of WT1 positive cells not only surrounding but also within the organoid, accounting for 5 ± 2% of the total cells present in MZ (Supplementary Fig. 8d), which could represent epicardial-derived cells that migrated from the EL towards MZ. An organized vascular-like network of spindle-shaped CD31⁺ cells was also observed throughout EMOs and also in some areas surrounding the EMOs edges (Fig. 4eI–V and Supplementary Fig. 8e), which contrasts with the fewer CD31⁺ cells observed in control CM aggregates (Fig. 4eVIII, f). Interestingly, it was confirmed that at least a part of the CD31⁺ cells co-expressed WT1 (Fig. 4eI, III, white arrows), indicating that these cells may have origin in PE and/or STM-like cells.

We also observed strong deposition of ECM at the periphery, likely overlapping with the endothelium, and also co-localizing with the EL. Collagen I predominated at the outer part of the EMOs, where it may contribute to either endothelial stabilization and/or an EL basal lamina, whereas fibronectin (FN) and laminin (LN), co-localize to a great extent with the EL (Fig. 4eVI, VII). Additionally, ECM deposition was also observed within the MZ, mainly fibronectin and laminin. This conspicuous ECM production may be related with an enrichment of fibroblasts derived from PE cells. Corroborating this idea, the expression levels of fibroblast markers *POSTN* and *DDR2* were enriched in EMOs compared with the CM aggregates control (Fig. 4g).

EMOs were further analyzed to assess for improved ventricle CM maturation. We observed an enriched staining for the ventricular marker MLC2V compared with the control condition, particularly near the WT1⁺ epicardial-like layer, suggesting that the co-culture induces

ventricular CM maturation, an aspect that may also be reinforced by the presence of LN in the same area[46] (Figs. 4eVII and 5aI, II, IV, and Supplementary Fig. 9a). We also observed the expression of cTnI marker in EMOs, particularly near the epicardial-like layer, reinforcing the myocardium-like tissue maturation in that specific area of MZ (Supplementary Fig. 9b). Additionally, IF for the ventricular gap junction marker CX43 showed a stronger staining throughout the entire MZ of EMOs, compared with the control condition (Fig. 5aIII and Supplementary Fig. 9c, d). TEM analysis confirmed that CMs within the EMOs show a well-organized and aligned sarcomere structure with well-defined Z-line, A- and I- bands and H-zone, and present an increased sarcomere length (1.9 ± 0.3 μm) compared with CMs in control condition (1.3 ± 0.2 μm; Fig. 5b, c).

To assess EMOs functionality, spontaneous calcium flux transients were analysed before and after isoproterenol, E-4031 and verapamil drug stimulation (Fig. 5d and Supplementary Fig. 9e), and compared with CM aggregates. As expected, stimulation with isoproterenol, a β-adrenergic agonist, induced a significant increase in frequency of contraction, in both EMOs and control CM aggregates. Additionally, arrhythmic behavior was observed only in EMOs after stimulation with E-4031, a hERG channel blocker (Supplementary Fig. 9e), in agreement with the induction of early after depolarization (EADs) events that have been reported for this drug[47,48]. In age-matched control CM aggregates, after E-4031 stimulation, it was observed an increased frequency of contraction (Supplementary Fig. 9e).

Verapamil, a L-type calcium channel blocker, has been described to decrease the action potential duration by shortening the AP plateau phase, which translates in a less prolonged Ca²⁺ transient profile; an increasing concentration of this drug culminates in CMs ceasing contraction. The results obtained when exposing EMOs to this drug were dependent on the concentration of verapamil. For a concentration of 100 nM, EMOs cease beating, which is recovered after a washout step, and with a concentration of 50 nM, it was possible to observe the expected decrease in the decay time of Ca²⁺ transients[49]. In control condition, upon verapamil stimulation, it was also observed a decreased decay time of Ca2+ transients, however with a not so pronounced effect (Supplementary Fig. 9e), which may indicate a lower expression level of Ca²⁺ channels. Collectively, these results suggest that EMOs may present a functional and improved response to known drugs.

EMOs were kept in culture for up to 60 days to observe how the cell composition and structural organization evolve during time (Supplementary Fig. 9f). With prolonged time in culture, EMOs show two structurally different configurations, a denser and a cavity-like morphology, contrarily to CM aggregates that did not maintain viable for more than 30 days in culture. Although after 60 days of co-culture, the majority of the EMOs display a denser morphology (60% denser vs 40% cavity-like), the cavity-like structure only starts to be visible in culture after 30-40 days, which may indicate that the proportion of cavity-like EMOs may increase and be the tendency with an even longer period of co-culture. This may reflect a lack of extracellular matrix to support aggregate structure and function. IF staining of EMOs sections, for both subtypes, revealed that MZ is

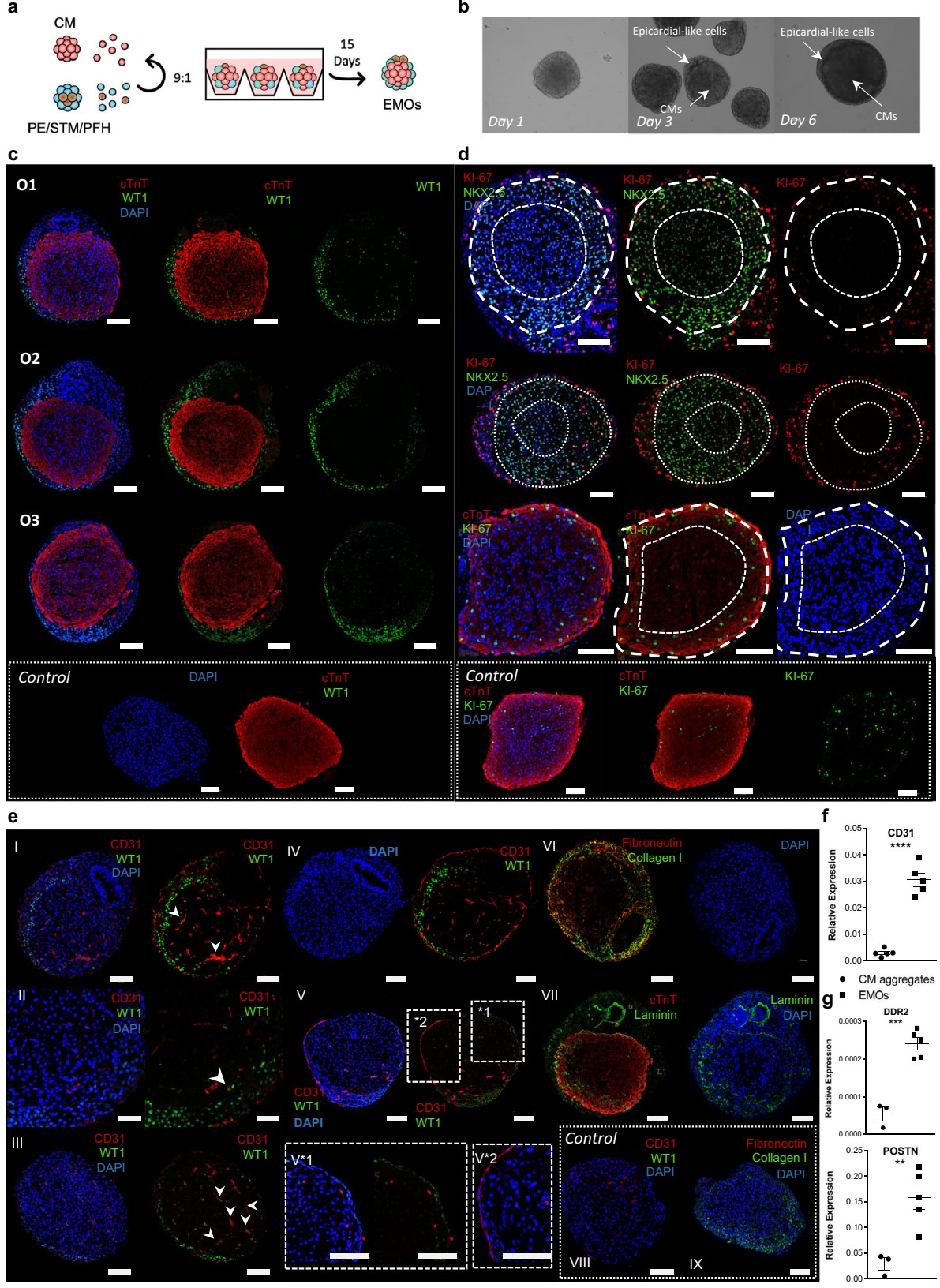

located at the periphery of the organoid, enveloping an E-CAD[+] epithelium in the case of cavity-like organoids. In denser organoids, the cTnT positive cells surround a core that contains WT1 positive cells and also epithelial structures.

Although the arrangement of WT1[+] cells surrounding the cTnT positive cells observed after 15 days of co-culture is mainly lost over time in culture, in some regions, it was possible to observe a few WT1 positive cells surrounding the cTnT positive area (Supplementary Fig. 9eII). At this time of co-culture, the organoids still present an almost continuous layer of CD31 positive endothelial-like cells near the periphery of the organoid. Nevertheless, the culture conditions may be further optimized to allow the maintenance of the

**Fig. 4 | Co-culture of PE/STM/PFH cells with CMs resulted in the development of a heart organoid that shows a self-organized epicardial-like layer surrounding a myocardium-like tissue. a** Schematic representation of EMOs generation. The ratio 9:1 corresponds to 9 CMs: 1 PE/STM/PFH cells. **b**, Representative BF images of EMOs at different time points of co-culture (day 1, day 3 and day 6), highlighting the epicardial-like cells and CMs within the EMOs. **c**–**e** Representative IF staining of EMOs sections after 15 days of co-culture, highlighting the epicardium- and myocardium-like layers (**c**), the difference between cTnT$^+$/NKX2.5$^+$ cells near the epicardium and the inner core of the organoid (**d**), and endothelial-like cells and ECM proteins present within the EMOs (**e**). IF staining of CM aggregates is also presented as control for the key mentioned markers. Scale bars, 100 μm. **f, g** Gene expression profile of CD31 (**e**), DDR2 and POSTN (**f**) in control CM aggregates and EMOs after 15 days of co-culture. Data are represented as mean ± SEM of n = 3 (CD31, POSTN, and DDR2 CM aggregate) and n = 5 (CD31, POSTN, and DDR2 EMOs) independent experiments. Exact p-values: CD31 p < 0.0001; DDR2 p = 0.003; POSTN p = 0.0082. Ox nomenclature identifies different organoids obtained from at least three independent experiments. **p < 0.0¹ ***p < 0.001 ****p < 0.0001.

epicardial-myocardial structural organization observed at earlier stages of co-culture for extended period of time.

In conclusion, co-culture of CMs with PE/STM/PFH cells in a 3D environment, initiates a process of cellular self-organization generating an organoid containing a WT1$^+$-epicardial-like layer surrounding a cTnT positive myocardium-like tissue. The analysis of these structurally organized EMOs shows preliminary evidences that point for the recapitulation of important processes previously associated to the role of epicardium in vivo such as myocardium proliferation, compaction, and maturation. Moreover, presence of a network of CD31$^+$ cells also in the MZ, suggests that within the EMOs mechanisms of coronary vasculature formation and organization are replicated.

## Combined WNT/BMP activation potentiate posterior SHF/ splanchnic mesoderm commitment

Since the development of the PE/STM/PFH organoids was dependent on the initial cell heterogeneity present in D5 progenitors, we next assessed the impact of mesendoderm modulation at early stages of differentiation, not only on PE/STM/PFH organoids specification but also on CM differentiation. It is well described in the literature the complex crosstalk between WNT and BMP4 signaling pathways on the modulation of cardiac/paraxial mesoderm and DE specification from hPSC-derived primitive streak (PS) population[50]. Particularly, it has been demonstrated that activation of BMP signaling after PS induction compromises DE and favors LPM specification, while prolonged WNT signaling activation improves LPM or paraxial mesoderm specification, depending on the level of WNT activation[32,33,50,51].

To reduce endoderm progenitors, without compromising the LPM specification at the expense of paraxial mesoderm, different concentrations of CHIR, between D1 and D3, were tested. It was noticed that increasing concentrations of CHIR compromised KDR$^+$ LPM progenitor cells specification (Supplementary Fig. 10a). However, 1.5 μM of CHIR did not affect the percentage of KDR$^+$ cells and allowed the reduction of c-KIT positive endoderm-like cells (11.7 ± 1.4%) at D5 (Supplementary Fig. 10b). Moreover, a combined activation of WNT and BMP4 induced further reduction of c-KIT positive cells (6.3 ± 1.8%) (Supplementary Fig. 10b), which will be referred from now on as WB condition.

To obtain genetic insight on the impact of CHIR and BMP4 supplementation on progenitor cells specification, bulk-RNA sequencing analysis of D3 and D5 populations in WB condition (D3WB and D5WB) and control conditions (D3 and D5) was performed (raw data deposited in GEO under "GSE184302"). Differential expression analysis comparing D3WB versus D3 populations revealed the up-regulation of genes known to be involved in paraxial mesoderm specification from PS, specifically MSGN1, DLL1, MSX1, and CDX2 (Fig. 6a), further confirmed by PCR analysis (Fig. 6b). This result highlights that even at low level, WNT activation between D1 and D3, was sufficient to allow the specification of early paraxial mesoderm progenitors. However, in D5WB condition we did not observe increased expression of the early somite progenitor markers PARAXIS (TCF15) and FOXC2[33], which may indicate that the initial paraxial mesoderm progenitors specified at D3, after subjected to WNT inhibition for 48 h, do not further progress to somitogenesis.

Interestingly, D3WB vs D3 analysis also highlighted the up-regulation of ALDH1A2, the major RA generating enzyme in the early embryo, and the RA target HOX family genes HOXA1 and HOXA3. These genes continue up-regulated at D5 of differentiation in the WB condition, where the RA related genes NR2F2, HOXB1, HOXA1, HOXA3, HOXA5, and RDH10 were also up-regulated (Fig. 6c). It is known that the specification of pre-somitic and somitic paraxial mesoderm is controlled by a tight spatiotemporal arrangement of FGF and RA signals, being Aldh1a2 expressed in the most anterior pre-somitic mesoderm, expanding anteriorly to the posterior second heart field (SHF). Here, RA forms an antero-posterior gradient, anti-parallel to that of FGF[52], being the posterior SHF characterized by the expression of Aldh1a2, Nr2f2, Hoxb1, and Tbx5, and the anterior SHF characterized by the expression of Tbx1 and Fgf8/10 [53].

Differential expression analysis of D5WB vs D5 shows up-regulation of genes that have been described, in mouse models, to be involved in specification of the posterior region of SHF[53], STM related splanchnic mesoderm[54], STM-derived mesenchymal cells, involved in embryonic liver development[30] and PE[4], including BNC2, IGFBP3, GATA4, MAB21L2, FOXF1, PBX3, and HOXC4 (Supplementary Data 1). This was further confirmed by PCR analysis, which revealed increased expression of FOXF1, MAB21L2, GATA4, TBX5, and HAND2 in D5WB compared with control (Fig. 6d). On the other hand, differential expression analysis also revealed down-regulation of genes related with embryonic heart tube development, anterior SHF specification and positive regulation of SHF CM proliferation, including SALL1, SIX1, EYA1, EYA2, DLK1, and NKX2.6 (Fig. 6e, f). PCR analysis confirmed lower expression levels of cardiac mesoderm markers TBX1, TBX20, and NKX2.5 in the D5WB condition (Fig. 6g).

These results point to the possibility that RA signaling may have an important role in controlling segregation between cardiac/anterior SHF and PE/STM/posterior region of the SHF, which corroborates a recent study showing that the splanchnic mesoderm progenitors, from where STM cells are specified, experience higher levels of RA signaling than the early cardiac mesoderm[54]. The cell surface marker ROR1, recently identified as a marker of cardiac mesoderm that enables early prediction of CM differentiation efficiency[55], was also down-regulated at D3WB and D5WB, compared with control (Fig. 6a, c), supporting further the hypothesis of compromised cardiac mesoderm specification in the WB condition. Additionally, by PCR analysis, we also observed that 1) the expression level of mesendoderm transcription factor T in the WB condition was higher at D2 of differentiation, after the peak of expression at D1 in both conditions, and 2) MESP1, a known early pre-cardiac progenitor marker, presented a significantly higher expression in D3WB compared with D3 condition (Fig. 6b), which may indicate that MESP1 is a common marker for both LPM subpopulations. In fact, the heterogeneity among MESP1$^+$ progenitor cells is well reported[56,57], having been suggested that the majority of epicardial cells arise from an independent population of unipotent Mesp1 progenitors that will give rise to the epicardium lineage and not to CMs[57].

Apart from the up-regulation of RA signaling in the WB condition, Notch signaling was also down-regulated compared with the control condition at D5 of differentiation, with lower expression of genes involved in the regulation of this pathway, DTX1, JAG1, and GRIP2

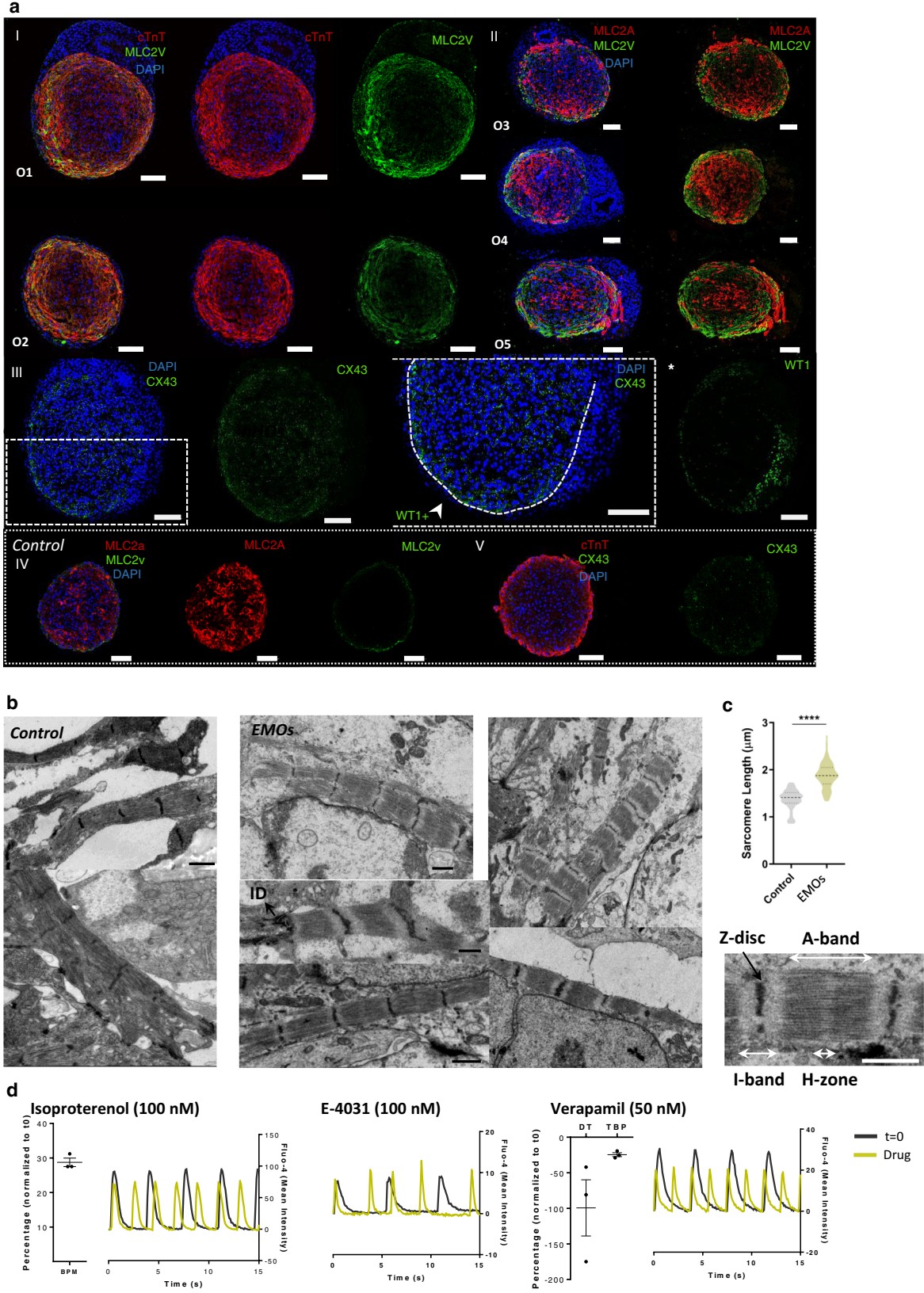

(Fig. 6e). In fact, Notch signaling has been associated to several stages of heart development, particularly during cardiac progenitor specification[58] and further commitment into CMs[59,60]. Regarding endoderm progenitors, it was identified the down-regulation of the endoderm progenitor cell surface marker c-KIT (Fig. 6a) in D3WB and of the transcription factor EOMES in the D5WB condition compared

with the respective control (Fig. 6h). PCR analysis further corroborated the lower expression level of *SOX17*, *EOMES* and *HHEX* in the progenitor population derived from the WB condition (Fig. 6i). Within the top 10 D5WB vs D5 up-regulated genes was also found the CD22 gene, indicating a putative role of this protein as a cell surface marker of PE/STM splanchnic mesoderm progenitors[61].

**Fig. 5 | EMOs present improved signals of ventricle myocardium maturation in comparison with CM aggregates only. a** Representative IF staining of EMOs sections after 15 days of co-culture for the mature ventricle CM marker MLC2V, the atrial/immature ventricle CM marker MLC2A and for connexin 43 (CX43) gap junction, highlighting the more intense staining of the mentioned markers near the edges of the EMO, which are in contact with the WT1$^+$ cells. IF staining of CM aggregates is also presented as controls for the key mentioned markers. Scale bars, 100 μm. **b** Representative transmission electron microscopy (TEM) images showing sarcomeres in EMOs and control CM aggregate sections after 15 days of co-culture.

ID intercalated disks. Scale bars, 1 μm. **c**, Quantification of sarcomere length in EMOs and respective control ($n > 10$ sarcomere measurements from $n = 5$ organoids and $n = 4$ CM aggregates; exact $p$ = value <0.0001). **d** Calcium transient profiles of EMOs after 15 days of co-culture before and after drug stimulation with isoproterenol, verapamil and E-4031. Data are represented as mean ± SEM of $n = 3$ independent experiments. EMOs epicardium-myocardium organoids, O organoid, BPM beats per minute, DT decay time, TBP time between peak. Ox nomenclature identifies different organoids obtained from at least three independent experiments. Source data are provided with this paper.

Together these results suggest that the combined activation of BMP and WNT between D1 and D3 of hiPSCs differentiation may induce the up-regulation of the RA pathway, which may originate the sub-population of early paraxial mesoderm progenitors, and consequently drive the LPM progenitor cell population from cardiac mesoderm to a posterior splanchnic/SHF mesoderm specification (Fig. 6j).

### Enrichment of posterior SHF progenitors compromise CM specification

To evaluate PE and CM specification from D5 progenitors after CHIR and BMP4 treatment between D1–D3 (Fig. 6k), FC and IF analysis at D11 of differentiation were performed. CM commitment from the WB progenitors resulted in a decreased percentage of cTnT positive cells, from 88.7 ± 1.2% to 45.3 ± 7.5% of cTnT$^+$ (Fig. 6l), as expected. With this protocol, the aggregates obtained tend to present two segregated areas, with only one showing a contractile phenotype (Supplementary Fig. 10c). IF staining confirmed the presence of a cTnT$^+$ area and revealed that the non-contractile region represents a cluster of cells that stain for the PE/STM markers WT1 and LHX2 (Supplementary Fig. 10c). This result corroborates the previously suggested hypothesis that a subpopulation of the LPM progenitors that is favored by this protocol is more prone to differentiate into PE/STM cells, and reinforce the idea that CM- and PE/STM- LPM progenitors could have a different molecular signature. Interestingly, treatment of D1 mesoderm progenitors with RA until D3 of differentiation, yielded a similar reduction on the CM differentiation efficiency (46 ± 13 cTnT$^+$ cells) (Supplementary Fig. 10d), reinforcing the hypothesis raised from RNA-seq data that pointed for RA signaling as one of the stimuli behind the generation of this phenotype.

Regarding PE cells specification from the D5WB progenitor population, as expected, its commitment was not compromised, as demonstrated by a statistically significant slight increase in the percentage of WT1 positive cells from 73.7 ± 1% to 84 ± 1% (Fig. 6m) and an increased expression level of the PE/STM markers *WT1*, *TBX18*, and *TCF21* (Supplementary Fig. 10e). However, some differences were observed, regarding the different subpopulations, when compared with the PE/STM/PFH control organoids. Specifically, the organoids generated from the WB condition, revealed an almost absence of ISL1$^+$ cells and, although it was still possible to observe endoderm-derived CDX2/AFP/HNF4A positive cells within the organoids, it was evident that the generation of well-developed and organized epithelial tube-like structures was compromised, which is probably related with the decreased percentage of endoderm progenitors at D5 of differentiation (Fig. 6n and Supplementary Fig. 10f).

Since it has been also reported the induction of PE cells without the WNT signaling inhibition step[19,20], the impact of PE induction period starting at D3WB of differentiation instead of D5WB was assessed. FC analysis showed that induction at D3 allowed efficient generation of WT1 positive cells after 11 days of differentiation (Supplementary Fig. 10g). However, organoids generated from D3WB population show a statistically significant increase in the expression of the PE specific markers *UPK3B*, *UPK1B*, *ANXA8*, and *BCN1* (Supplementary Fig. 10h). These results suggest that although the percentages of WT1$^+$ cells obtained from both progenitor populations, D5WB and D3WB, are the same, induction at D3 seems to favor a more restricted PE phenotype.

In fact, IF of sections of organoids generated from D3WB progenitors revealed an almost absence of LHX2$^+$-STM-like cells (Supplementary Fig. 10i).

Altogether, these results suggest that WNT and BMP4 signaling pathway activation between D1-D3 of differentiation negatively affect CM commitment without compromising PE/STM specification, and the reduction of endoderm progenitors in the WB condition also compromise the development of well-organized epithelial-foregut structures in PE/STM/PFH organoids. Additionally, the results suggest that PE/STM induction from D3WB favors a more PE-like phenotype compared with induction at D5WB that may indicate the existence of different progenitors for the PE and STM populations.

## Discussion

Here we describe a robust and simple platform for the generation of a self-organized multilineage hPSC-derived organoid that combines both PE/STM- and posterior foregut/hepatic diverticulum-like cells (Fig. 7). These organoids recreate the early stages of PE development, showing morphological similarities with the early and transient embryonic structure comprising PE, STM, and posterior foregut. In vivo, PE cells arise adjacently to the STM, and in turn STM provides signaling cues for hepatoblast specification and a territory for the migration of the hepatic bud cells. Similarly, PE/STM/PFH organoids present a PE-like cell population (WT1$^+$/LHX2$^{-/+low}$) at the periphery of the organoid and STM-like cells (WT1$^{+low}$/LHX2$^+$) located between the PE area and the foregut-like epithelium. This arrangement recapitulates what was described by Lupu and colleagues in the mouse, where *Wt1$^+$* cells were found not only in the PE region but also in the STM, and *Lhx2* was further identified as a marker of STM cells[12]. Additionally, a WT1$^+$/ISL1$^+$ cell population, which, according to a recent report[30] may represent a sub-population of the STM-derived early liver mesenchyme, was also identified within the PE/STM/PFH organoids.

In the past years, several cardiac models have been engineered aiming at structurally and functionally mature CMs for different applications, including drug screening and cardiotoxicity studies[62–64]. Nevertheless, the emergence of heart organoids to recreate in vitro the early stages of cardiac development is recent[1–3]. The recapitulation of the transient embryonic structure that comprises an extraembryonic progenitor pool crucial for normal heart development, the PE, and the adjacent STM and posterior foregut/hepatic diverticulum, has not been reported so far. Only recently PE/epicardium cells have been explored in in vitro cardiac models, with few protocols describing the efficient generation of WT1$^+$ PE-like cells from hPSCs in 2D monolayer systems[19–21,65]. In addition, reports describing STM-like cells specification from hPSCs are almost absent from the literature, with only two studies demonstrating the direct generation of STM/liver-mesothelium-like cells in 2D culture that were also identified as WT1$^+$[54,66]. Although it is known that STM and mesoderm-derived populations involved in liver development are a complex array of sub-populations with different molecular signatures, as recently identified in the mouse[30,54], there is a lack of in vitro models that recreate the liver mesenchyme development and its interaction with hepatic endoderm primordium. Therefore, the described simple and robust platform to develop PE/STM/PFH organoids appear as an important tool to study not only PE but also STM and liver bud specification in vitro, and to

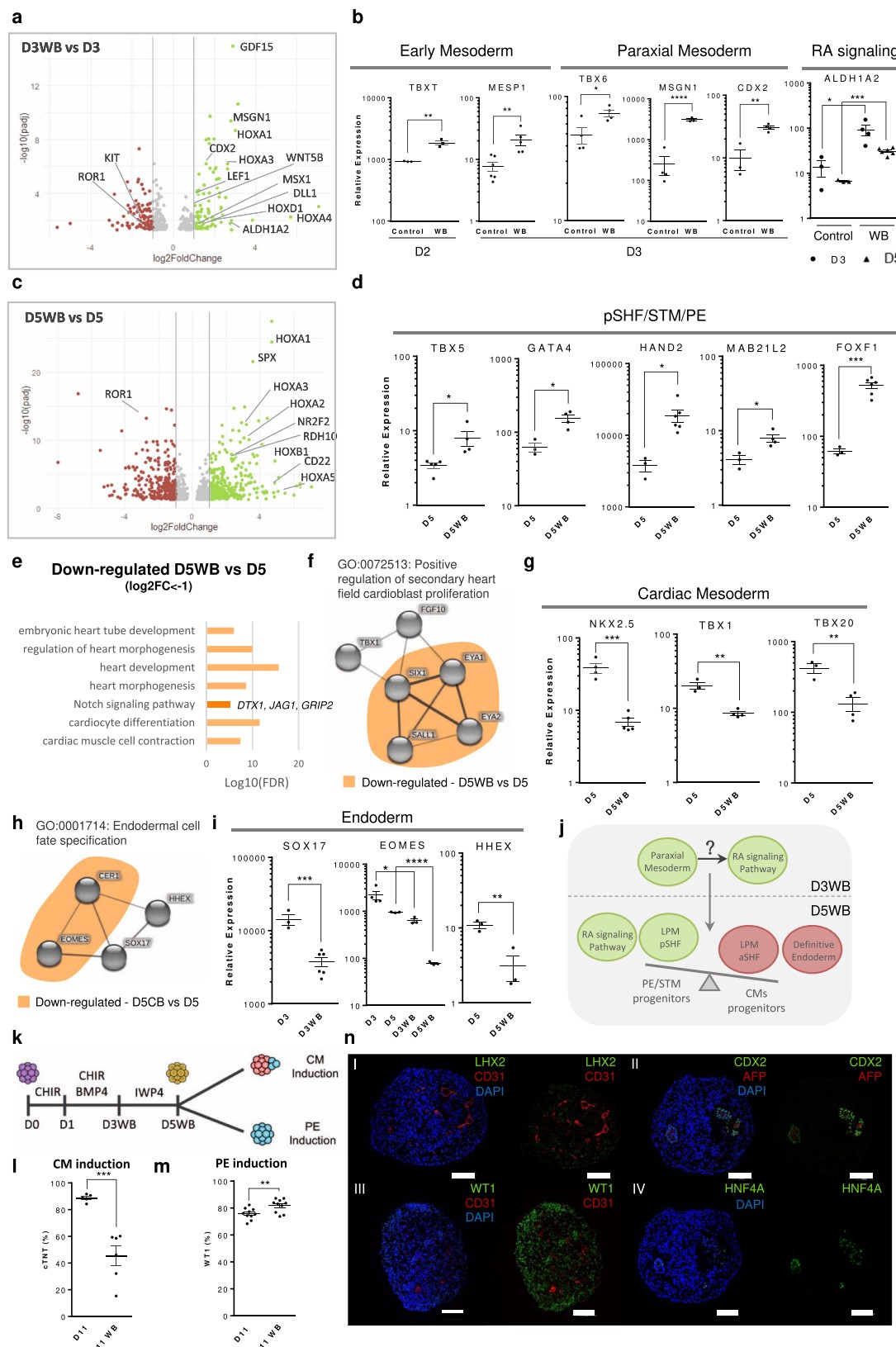

study the interaction between endodermal and mesodermal progenitors involved in this process of specification.

The concept of multilineage organoid has been also recently introduced in the cardiac field by Drakhlis and colleagues that reported an organoid recreating both heart and foregut development[2]. Although the early stages of differentiation, consisting of a simple

WNT signaling modulation, are similar to our protocol, the lack of an additional step for PE/STM specification generated an organoid featuring instead anterior foregut and myocardial-like layer, contrarily to the present PE/STM/PFH organoids.

Although the protocol for PE/STM/PFH organoids generation is based on the activation of the same signaling pathways (BMP, RA

**Fig. 6 | Prolonged and combined activation of WNT and BMP signaling pathways during the first stages of hPSC differentiation induced posterior SHF/splanchnic mesoderm specification potentially through RA signaling activation. a** Volcano Plot highlighting the differentially expressed genes in D3WB vs D3 condition. **b** Gene expression profile of early mesoderm markers *T* and *MESP1*, paraxial mesoderm markers *TBX6, CDX2*, and *MSGN1* and the RA synthesizing enzyme *ALDH1A2* in control and WB conditions. Values are normalized to *GAPDH* and D0 of differentiation. Data are represented as mean ± SEM of $n = 3$ (TBXT, CDX2, ALDH1A2 Control), $n = 4$ (TBX6, MSGN1, ALDH1A2 D3WB), $n = 5$ (MESP1 WB, ALDH1A2 D5WB), and $n = 6$ (MESP1 Control) independent experiments. Exact *p*-values: TBXT $p = 0.0034$; CDX2 $p = 0.0028$; TBX6 $p = 0.0305$; MSGN1 $p < 0.0001$; MESP1 $p = 0.0090$; ALDH1A2 D3 control vs WB $p = 0.0427$, ALDH1A2 D5 control vs WB $p = 0.0001$. **c** Volcano Plot highlighting the differentially expressed genes in D5WB vs D5 condition. **d** Gene expression profile of *TBX5, GATA4, HAND2, MAB21L2, and FOXF1* at D5 and D5WB conditions. Data are represented as mean ± SEM of $n = 4$ (TBX5 D5WB, GATA4 D5WB, MAB21L2 D5WB), $n = 3$ (GATA4 D5, HAND2 D5, FOXF1 D5, MAB21L2 D5), $n = 5$ (TBX5 D5), $n = 6$ (HAND2 D5WB, FOXF1 D5WB) independent experiments. Exact *p*-values: TBX5 $p = 0.0290$; GATA4 $p = 0.0121$; HAND2 $p = 0.0275$; MAB21L2 $p = 0.0242$; FOXF1 $p = 0.0008$. **e** Top gene ontology (GO) terms for biological processes identified (FDR < 0.05) of differentially down-regulated genes (Log$_2$ FC < −1 and adjusted *p*-value < 0.05) for D5WB vs D5. **f** Cluster analysis highlighting GO term "Positive regulation of secondary heart field cardioblast proliferation". In orange are highlighted the genes that were significantly down-regulated in D5WB vs D5 analysis. **g** Gene expression profile of *NKX2.5, TBX1*, and *TBX20* at D5 and D5WB conditions. Data are represented as mean ± SEM of $n = 4$ (NKX2.5 D5, TBX1 D5WB, TBX20 D5WB), $n = 3$ (TBX1 D5, TBX20 D5) and $n = 5$ (NKX2.5 D5WB) independent experiments. Exact p-values: NKX2.5 $p = 0.0005$; TBX1 $p = 0.0021$; TBX20 $p = 0.0070$. **h** Cluster analysis highlighting GO terms: "Endodermal cell fate specification". In orange are highlighted the genes that were significantly down-regulated in D5WB vs D5. **i** Gene expression profile of *SOX17, EOMES*, and *HHEX* at D3/5 and D3/5WB conditions. Data are represented as mean ± SEM of $n = 3$ (SOX17 D3, EOMES D5/D3WB/D5WB, HHEX), $n = 4$ (EOMES D3) and $n = 6$ (SOX17 D3WB) independent experiments. Exact *p*-values: SOX17 $p = 0.0006$; EOMES D3 Control vs D3WB $p = 0.0335$, D5 Control vs D5WB $p < 0.0001$; HHEX $p = 0.0073$. **j** Schematic representation highlighting the main mechanistic outcomes retrieved from the bulk-RNAseq data. **k** Scheme illustrating the process of CM specification and PE induction from D5WB progenitor cell population. **l** Percentage of cTnT$^+$ cells after 11 days of CM differentiation protocol using D5 and D5WB progenitor cell populations. Data are represented as mean ± SEM of $n = 5$ (D11) and $n = 6$ (D11WB) independent experiments. Exact *p*-value = 0.0006. **m** Percentage of WT1$^+$ cells after 11 days of PE induction from D5 and D5WB progenitor cell populations. Data are represented as mean ± SEM of $n = 10$ independent experiments. Exact *p*-value = 0.0087. **n** Representative IF staining of D11 organoid sections obtained using D5WB progenitor cells. Scale bars, 100 μm. *$p < 0.05$, **$p < 0.01$, ***$p < 0.001$ and ****$p < 0.0001$.

and/or WNT) also modulated in previous 2D protocols for PE cells specification from LPM progenitors, we observed different outcomes, which highlights the crucial role played by 3D environment. The results suggest that the 3D configuration, and the consequent generation of soluble factors gradients inside the organoids, were responsible for the generation of an heterogenous progenitor population that comprises self-organized mesoderm (±80%), DE (±20%), and vascular progenitor cells (±5%). Although Drakhlis and colleagues did not characterize the progenitor cell population from which their heart organoids were specified, they corroborate the impact of aggregate size on the generation of the heart/foregut organoids. The co-emergence of these populations within PE/STM/PFH organoids suggests that at least part of the stimuli involved in PE/STM specification from mesoderm progenitors are identical to the stimuli that support posterior foregut/hepatoblasts specification from DE progenitors. BMP4 and WNT, both stimuli used during PE/STM induction period, have also been identified as important signaling cues for DE specification into posterior foregut with further potential to generate liver bud progenitors[29,32]. Interestingly, trough modulation of WNT, BMP4 and RA signaling pathways during PE induction, we demonstrated that removal of RA activation at this stage of differentiation potentiated the specification of the posterior foregut epithelium present in PE/STM/PFH into liver bud organoid models, presenting a hepatoblast-like epithelium and a STM-like population.

As proof-of-principle, we advanced an application for the PE/STM/PFH cells to generate an epicardium-myocardium organoid model (EMOs), through co-culture with CM aggregates. Although we explored different strategies to establish the co-culture model, e.g. the direct fusion between PE/STM/PFH organoids and CM aggregates, only reaggregation of both cell types in a specific ratio allowed to recreate a self-organized structure comprising a continuous well-defined WT1$^+$ epicardial-like layer that fully surrounds a cTnT$^+$ myocardium-like tissue (Fig. 7). Although most of the reported studies describe the epicardium as a uniform single layer of epithelium covering the myocardium, in EMOs we observed the presence of both single and multi-layered WT1$^+$ cells surrounding the MZ. Normally, the regions where we found a multi-cell layer of epicardial-like cells overlap with a prevalence of CD31$^+$ vascular network of cells. Interestingly, a study that characterized the human epicardium during embryonic heart development[5] found differences between atrial and ventricular epicardium and showed that the ventricular epicardium contains regions of multiple layers of epicardial cells near

blood vessels. This may explain the arrangement observed in our EMOs.

Apart from the structural recapitulation of epicardial layer formation, EMOs also recapitulate processes already attributed to epicardial regulation, such as CM proliferation, and functional and structural maturation. Therefore, EMOs represent a reliable tool to recreate the early stages of PE-myocardium interaction, particularly the WT1$^+$ cell arrangement over the myocardium-like layer, opening a clear path to study and model epicardial-myocardial interactions in vitro. Although recent reports have shown evidences for the importance and the positive impact of co-culturing hPSC-CMs with PE cells to obtain more mature and adult-like cardiac models[42,43,67], only two recent reports established heart organoid models that incorporate epicardial cells[13]. However, none of them showed 1) a well-defined self-organized layer of epicardium surrounding the a CM region in a 3D environment, as it is possible to observe in our EMOs, nor 2) clearly demonstrated the direct impact of epicardial cells on myocardial-like tissue organization and maturation, as it was showed here, supporting the increased relevance of the presented heart organoids.

Additionally, although the contribution of epicardial cells for the coronary vasculature of the heart is still controversial[6,11,12], in EMOs we observed within the myocardium-like region the presence of some CD31$^+$/WT1$^+$ endothelial-like cells as part of the CD31$^+$ pool, assembled into an extensive and well-organized vascular-like network. Evaluation of WT1 expression, in mouse and human fetal hearts, revealed that WT1 is expressed in endothelial cells of both arteries and veins at early stages of development[68,69]. Lineage tracing of PE/STM cells also identified WT1$^+$/CD31$^+$ cells in the coronary vasculature of the developing mouse heart[11]. Finally, while identifying cells expressing WT1 in the coronary endothelium of the murine heart, Lupu and colleagues[12] hypothesized a STM origin for these cells. In fact, none of the in vitro models that explored the co-culture between CMs and PE cells have reported the derivation of PE cells into WT1$^+$/CD31$^+$ cells. Since our co-culture model includes not only PE-like but also STM-like cells, it is possible that WT1$^+$/CD31$^+$ cells observed in EMOs may have a STM origin. Therefore, the developed EMOs can also be applied to explore not only PE- but also STM-derived cells specification and their contribution for heart organogenesis in vitro, particularly regarding vascularization.

Apart from the yet unraveled molecular signature of PE cells and their distinction from STM, the characterization of the cells from which PE and STM are specified and how they diverge from the progenitors originating CMs, within the LPM progeny, is not fully known. Here we

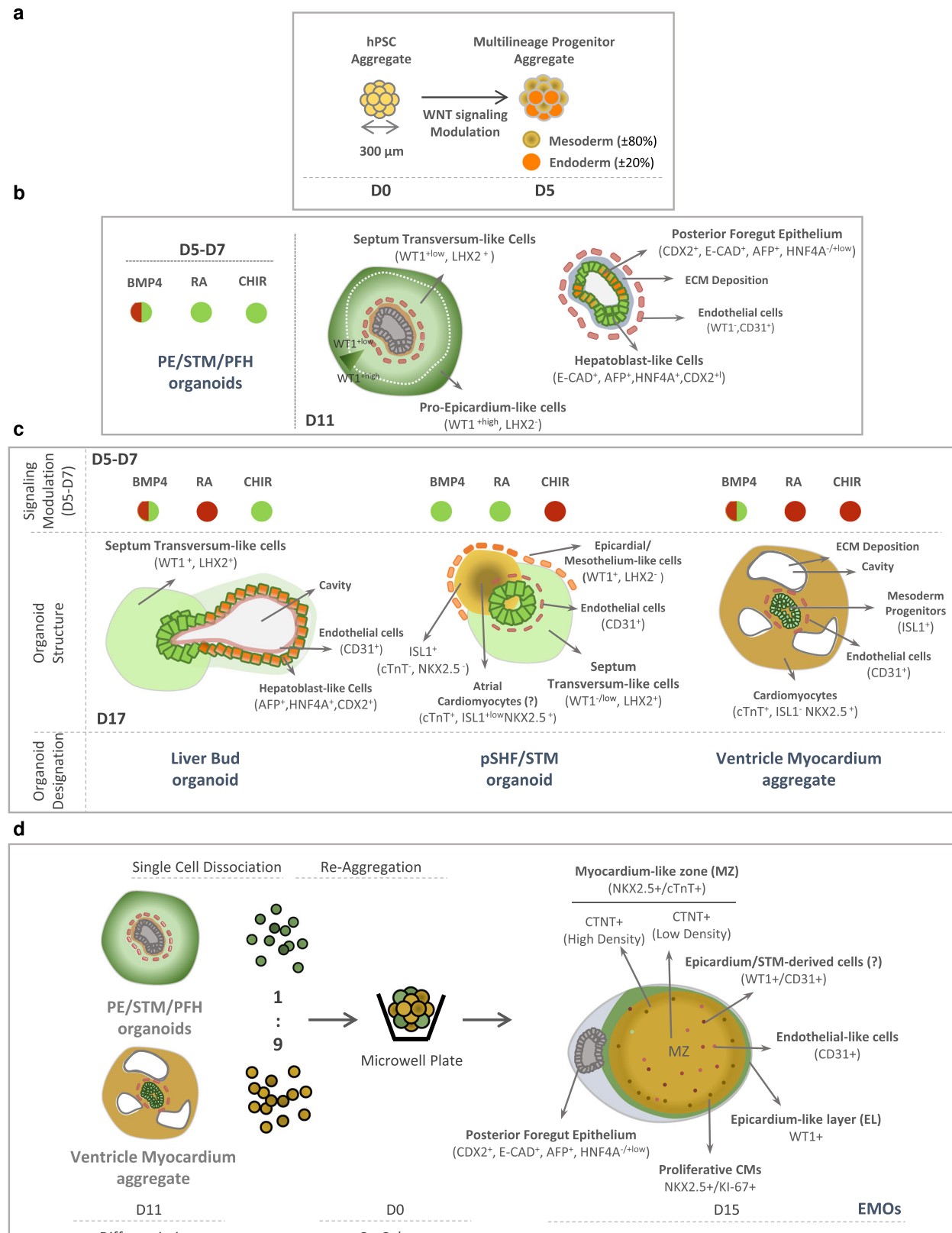

**Fig. 7 | WNT, BMP4, and RA signals modulation conditions PE/STM/PFH organoids specification and these organoids are key to generate a physiologically relevant epicardium-myocardium heart organoid model in vitro.**
**a** Progenitor cell composition of D5 aggregates obtained through temporal modulation of WNT signaling. **b** Structural organization and different cell populations present in PE/STM/PFH organoids highlighting the critical signaling pathways that should be activated during PE induction period (D5-D7) to generate those organoids. **c** Structural organization and different cell populations present in organoids generated through modulation of WNT, BMP4, and RA during PE induction period. Green spheres indicate activated signal and red spheres indicate not activated signal. **d** Co-culture process for the generation of EMOs, and identification of the different regions/cell populations present in EMOs. MZ myocardium-like zone, EL epicardial-like layer.

showed how LPM progenitors can be modulated to potentiate PE/STM specification at the expense of CMs from hPSCs. Moreover, our data indicates (1) specification of PE/STM cells from splanchnic mesoderm with a molecular signature similar to posterior SHF, and (2) RA signaling as a putative stimulus for specification of this posterior SHF progenitor cells[54]. Particularly, we show that the combined activation of WNT and BMP pathways, after the first step of CHIR supplementation for one day (WB condition), induces specification of paraxial mesoderm progenitors, RA signaling pathway activation, and consequently favors posterior SHF/splanchnic mesoderm at the expense of CM progenitors specification. This, in turn, resulted in compromised CM differentiation and unaffected PE/STM specification, compared with the control conditions. Although further studies are needed to assess the dependency of RA signal activation in the WB condition with specification of paraxial mesoderm progenitors, it is well described that RA is a critical morphogen in the establishment of anterior-posterior polarity in the heart. In vivo fate mapping studies in different models describe the emergence of a Aldh1a2-expressing cell population in the LPM that contributes for the specification of the posterior region of the SHF, which ultimately contributes for the specification of the atrial chambers of the heart[70–73].

Altogether, the results herein highlight the potential of the multilineage PE/STM/PFH organoid model itself, and the respective culture platform, as tools to investigate the mechanisms involved in human PE/STM embryogenesis and in hepatic parenchymal and non-parenchymal cells specification, through combinatorial modulation of signaling pathways. Moreover, the simple and reproducible platform for the generation of EMOs, presenting a continuous layer of epicardial-like cells and a patterned vascular-like network, opens a clear path for the study and modeling of epicardial-myocardial interactions in vitro. This may be further explored for modelling congenital heart diseases (CHD), such as coronary vascular diseases and myocardium non-compaction cardiomyopathies, and as models for drug screening and toxicology assays during the embryonic heart development.

## Methods

### Cell Maintenance
hiPSCs (cell lines iPS-DF6-9-9T.B and F002.1A.13) and hESC (H9) lines were maintained in mTeSR™1 (StemCell Technologies) in six-well plates coated with Matrigel™ (Corning). Medium was changed daily. Cells were routinely passaged every three to four days using 0.5 mM EDTA solution (Thermo Fisher Scientific).

### CM aggregates and PE/STM/PFH organoid differentiation from hPSCs
hPSCs were maintained in mTeSR™1 (StemCell Technologies) in six-well plates coated with Matrigel™ (Corning). Cells were passaged every 3 days and medium was changed daily. For CM aggregates and PE/STM/PFH organoids generation, hPSCs were harvested with accutase (Sigma) for 7 min at 37 °C. After dissociation, cells were quickly reaggregated inside microwell plates (AggreWell™800, StemCell Technologies) according to the manufacturer's instructions. Cells were plated at a cell density of $0.9–1.2 × 10^6$ cells/well in mTeSR™1 supplemented with 10 μM ROCKi. 24 h later, the total volume of medium was replaced and cells were maintained in mTeSR™1 without ROCKi for an additional two days. Differentiation was initiated on day 0 by replacing the mTeSR1 by RPMI 1640 medium (Thermo Fisher Scientific) with 2%(v/v) B-27 minus insulin (Thermo Fisher Scientific) (RB27-) supplemented with 11 μM CHIR99021 (Stemgent). After 24 h, medium was changed and on day 3, cells were cultured in RB27- supplemented with Wnt inhibitor IWP-4 (Stemgent) at a final concentration of 5 μM, for two days. At day 5, in the case of CMs differentiation, the medium was changed to RB27-, and in the case of PE/STM/PFH organoids, 3 μM CHIR, 25 ng/mL

BMP4, and 4 μM RA were added from D5-D7, using DMEM/F12 + 2.5 mM Glutamax+100 μg/mL of ascorbic acid (DMEM/F12) as basal medium. At day 7, aggregates were flushed from the Aggre-Well™800 plate and transferred to 6-Well Ultra-Low Attachment plates. Thereafter, medium was changed every 2 days until cell harvest. In the case of CM differentiation, the medium used was RPMI 1640 medium (Thermo Fisher Scientific) supplemented with 2%(v/v) B-27 (Thermo Fisher Scientific) and in the case of PE/STM/PFH organoids was DMEM/F12. Adaptations of the base protocols were also tested and mentioned throughout the entire manuscript. These include 1) testing the impact of VEGF in which case VEGF at 100 ng/mL concentration was added between D7 and D9, followed by VEGF supplementation at a concentration of 50 ng/mL from D9 and D11; 2) WB condition: in which case 1.5 μM CHIR and 2 ng/mL of BMP4 were supplemented between D1 and D3 of differentiation. WT1+-PE/STM cells were replated and further differentiated into SMCs and CFs, and epicardial WT1+ cells. For that purpose, D11-PE/STM aggregates were dissociated are replated in gelatin-coated 12-well plates at a density of $7 ×10^4$ cells/well, and cultured in DMEM/F12 Glutamax+Ascorbic acid with 10 ng/mL of FGF, 5 ng/mL of TGF-β1 and 0.5 μM A83, in the case of CFs and SMCs commitment, or forced PE phenotype maintenance, respectively.

### Establishment of a co-culture system between hPSC-CMs and hPSC-PECs
To establish the co-culture system, D11 CMs and D11 PE/STM/PFH organoids were singularized using 0.25% Trypsin-EDTA for 7 min at 37 °C. After cell counting, both cell types were combined at a proportion of 90%CMs:10%PE/STM/PFH cells and reaggregated using microwell plates (AggreWell™800, StemCell Technologies). For the control condition, CM aggregates and PE/STM/PFH organoids at D11 of differentiation were singularized and reaggregated in microwell plates. Organoids and controls were cultured in DMEM/F12 + Glutamax + Ascorbic acid supplemented with 10 μM ROCKi.

### Flow cytometry analysis
For flow cytometry analysis, day 5 differentiated aggregates, CM aggregates and PE/STM/PFH organoids were washed with PBS and then singularized with TrypLE 1X or 0.25% trypsin-EDTA, respectively, at 37 °C for 7 min. For enzymatic digestion neutralization, FBS-containing medium was added. After centrifugation and washing the cell pellet, cells were fixed with 2% PFA reagent for 20 min at RT or stored at 4 °C. For cell surface marker analysis, cells were washed twice with PBS and re-suspended in primary antibody (Supplementary Table 1) diluted in PBS + 2% (v/v) BSA, at ~500,000 cells per condition, and incubated for 30 min at RT. Afterwards, cells were washed with PBS and re-suspended in secondary antibody diluted in FACS buffer for another 15 min, at RT, in the dark. In the case of the conjugated antibodies (Supplementary Table 1), a single incubation period of 30 min at RT in the dark was performed. Finally, cells were washed twice with PBS for a final volume of 300 μL/FACS tube. For intracellular marker analysis, cells were first incubated with 90% (v/v) cold methanol at 4 °C for 15 min. Cells were then washed 3 times with a solution of 0.5% BSA in PBS. Cell pellet was resuspended and incubated with the primary antibody diluted in 0.1% Triton X-100 and 0.5% BSA in PBS, at RT for 1 h. After incubation, cells were washed twice and cell pellet was resuspended and incubated with the secondary antibody at RT for 30 min in the dark.

### Immunostaining analysis
**Sample collection.** CM aggregates and PE/STM/PFH organoids were fixed in 4% PFA at RT for 30 min in an agitation platform. After PFA removal, cells were stored in PBS at 4 °C for further analysis. Organoids were incubated in 15% (m/v) sucrose in PBS, at 4 °C overnight and afterwards embedded in 7.5%/15% gelatin/sucrose and frozen in iso-penthane at −80 °C. Aggregates with ten/twelve-μm sections were cut on

a cryostat-microtome (Leica CM3050S, Leica Microsystems), collected on Superfrost™ Microscope Slides (Thermo Scientific) and stored at −20 °C. Sections were then de-gelatinized for 45 min in PBS at 37 °C.

**Staining.** Organoid sections were incubated in 0.1 M Glycine (Millipore) for 10 min at RT, permeabilized with 0.1% Triton X-100 (Sigma), at RT for 10 min and blocked with 10% fetal goat serum (FGS, Gibco) in TBST (20 mM Tris-HCl pH 8.0, 150 mM NaCl and 0.05% Tween-20, Sigma), at RT for 30 min. Cells were then incubated with the primary antibody diluted in 10% FBS in TBST solution (Supplementary Table 1) at 4 °C overnight. Secondary antibodies were added for 30 min and nuclear counterstaining was performed using 4′,6-diamidino-2-phenylindole (DAPI, 1.5 μg/mL, Sigma), at room temperature for 5 min.

### Fluorescence intensity quantification
Fluorescence intensity in cryosections was obtained using the Plot Profile command in ImageJ (Fiji). For each cryosection 4 regions were measured.

### Transmission electron microscopy imaging
PE/STM/PFH organoids and CM aggregates were fixed in a solution 2% PFA + 2.5% Glutaraldehyde in 0.1 M PBS for 1 hour on ice. The samples were then embedded in 2% low melting point agarose and let it solidify on ice. After post-fixation in 1% osmium tetroxide for 30 min at room temperature, staining with 1% Tannic Acid for 20 min on ice and in 0.5% uranyl acetate for 1 hour at RT, samples were dehydrated in ethanol and embedded.

### Calcium transient imaging
For calcium imaging, the Fluo-4-AM dye (ThermoFisher) was used. Fluo-4-AM dye-loading solution was prepared according to the manufacturer's instructions and EMOs were incubated with that solution for 30 min at 37 °C. Afterwards, the Fluo-4-AM dye-loading solution was exchanged by DMEM/F12 medium. Before starting image acquisition, EMOs were left in the incubator for another 30 min to stabilize. Videos of beating EMOs and CM aggregates were taken for a period of 30–60 s with an TCS SP5 confocal inverted microscope (DMI6000, Leica).

### RNA isolation
CM aggregates and PE/STM/PFH organoids were washed with PBS and then singularized with 0.25% trypsin-EDTA, at 37 °C for 7-10 min. Total RNA was extracted using High Pure RNA Isolation Kit (Roche) according to manufacturer´s instructions.

### Real-time PCR
Total RNA was converted into cDNA with High Capacity cDNA Reverse Transcription Kit (Thermo Fisher Scientific). PCR reactions were performed with Green Master Mix (nzytech; Supplementary Table 2). Reactions were run in triplicate in ViiA7 Real-Time PCR Systems (Applied BioSystems). For each analysed time point, gene expression was normalized against the expression of the housekeeping gene glyceraldehyde-3-phoshate dehydrogenase (*GAPDH*) and results analysed with the QuantStudio™ RT-PCR Software.

### RNA sequencing and transcriptomic analysis
RNA was extracted from day 3 and day 5 of differentiation from control and WB conditions. At each time point, aggregates were singularized with accutase at 37 °C for 7 min. For enzymatic digestion neutralization, RPMI + 10% FBS was added and cell pellets were stored at −80 °C. Total RNA was extracted using High Pure RNA Isolation Kit (Roche), according to manufacturer´s instructions. RNA samples were sent to the NovoGene facility, where the integrity of the samples was assessed using an Agilent 2100 Bioanalyzer followed by library preparation

using NEBNext® Ultra TM Directional RNA Library Prep Kit for Illumina® and RNA sequencing using an Illumina NovaSeq 6000 system with a paired end 150 bp strategy. For RNA-seq analysis, sample run was aligned to hg38 reference genome using HISAT2 software. Counts were obtained using Featurecounts and differential expression analysis was performed with DESeq2 package of R. Further downstream bioinformatic analysis was performed in R and gene ontology analysis was performed in String web tool platform.

### Statistical analysis and reproducibility
All presented data regarding IF staining, flow cytometry and PCR analysis were representative of at least three biologically independent experiments per condition/marker. All raw data was collected in Microsoft Excel and statistical analysis was performed with GraphPad software. Statistical significance was evaluated with a two-tailed, unpaired Student's $t$ test ($p < 0.05$) when appropriate. The data are presented as mean ± SEM and represent a minimum of three independent experiments. Statistical significance was assigned as not significant (ns), $p > 0.05$; *$p \leq 0.05$; **$p \leq 0.005$; ***$p \leq 0.0005$; ****$p \leq 0.0001$. All micrograph images are representative of at least four independent experiments per condition/marker and calcium transient graphs are representative of three independent experiments.

### Reporting summary
Further information on research design is available in the Nature Portfolio Reporting Summary linked to this article.

## Data availability
The mesendoderm progenitor cell populations RNA-Sequencing data set generated in this study has been deposited in the National Center for Biotechnology Information Gene Expression Omnibus repository under accession code GSE184302. The raw data generated in this study are provided in the Source Data file. Source data are provided with this paper.

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

## Acknowledgements

We thank Instituto de Medicina Molecular (iMM) for the access to the bioimaging facility. We acknowledge the Electron Microscopy Facility at the Instituto Gulbenkian de Ciência for CM sample processing and imaging. RNA-sequencing data used in this publication was generated by Novogene Co., Ltd. This work was funded by national funds from FCT – Fundação para a Ciência e Tecnologia, I.P., in the scope of the project UIDB/04565/2020 and UIDP/04565/2020 of the Research Unit Institute for Bioengineering and Biosciences – iBB, the project LA/P/0140/2020 of the Associate Laboratory Institute for Health and Bioeconomy – i4HB and the project PTDC/EMD-TLM/29728/2017 granted to M.M.D.

## Author contributions

M.A.B., T.P.D., M.M.D., and P.P.O conceptualized the work. M.A.B. and T.P.D. designed the experiments. M.A.B. performed all cell culture experiments and further analysis. M.A.B., T.P.D., P.P.O., and M.M.D. discussed the results and wrote the paper. M.M.D. and J.M.S.C. provided financial support.

## Competing interests

The authors declare no conflicts of interest.
