## [Peer Review File · Nature Communications]

Human Multilineage Organoids Recapitulate
Pro-Epicardium/Septum Transversum/Posterior Foregut and
Support the Development of Epicardium-Myocardium
OrganoidsREVIEWER COMMENTS

Reviewer #1 (Remarks to the Author):

This manuscript provides a cutting-edge organoid model that recapitulate the pro-epicardium migration to form two clear myocardium-/epicardium- like layers, an essential phenomenon that occur in early heart development. Strengths of the study include the novelty of the multilineage nature of the organoid, and the characterization of the spatial relationships of different cell compositions in the model that could be used as an in vitro tool for myocardium-epicardium interaction and the heart organogenesis. Having said that, the myocardium-epicardium interaction of this organoid is poorly studied, and the advantages have not been described in an adequate manner. Rather, the work focuses more on BMP/WNT/RA effect in the early stages of differentiation of the pro-epicardium/septum transversum mesenchyme and the cardiomyocyte, which could potentially be done in 2D monolayer assays, and the conclusion is mostly confirmatory. In addition, there are several weaknesses in the study, and the value of the data presented unfortunately lack information for proper interpretation and to support the conclusion of the paper.

Major concerns

1. The authors conclude that the "hHOs clearly demonstrate the direct impact of epicardial cell on myocardial-like tissue organization and maturation (Page 16)", but the only "maturation" data the authors showed was the sarcomere length in Extended data Fig 3. They could elucidate more on the advantages of "co-culturing PE/STM with CM (hHO)" over "CM aggregate only" (Fig2). For example, is the proliferative CMs (Ki-67+) increased with the co-culture with PE/STM? Are there any differences in the ventricular gap junction (CX43) pattern? Are any of the cardiomyocyte "maturation" markers increased? What differences can you observe in long term cultures (e.g. day 17) in hHOs compared to CM aggregate only or PE/STM aggregate only (Extended Fig 2h)? The quantification for WT1 within MT could be done for CM as well as hHO and presented side by side (Fig2d) to emphasize the increase by PE/STM addition.
2. The analysis and conclusions of STM characterization is concerning. ISL1 and LHX2 are the two mainly used markers in this study to characterize STM, but they are also known markers for pharyngeal mesoderm Harel et al. (doi: 10.1073/pnas.1208690109). Interpretation of the increase of LHX2 and ISL1 should be done carefully preferably supported with additional markers (e.g. in Fig 4).
3. The advantages of adding BMP4 at D5-D7 of PE/STM differentiation is not clearly shown. Unfortunately, there is no data that support the "positive impact of RA signaling on specification of WT1+ cells in the presence of BMP4" (Page 9) since the authors refer to data that compares only "CHIR+BMP4+RA" vs "CHIR+BMP4" vs "CHIR" (Extended Fig4b). To conclude the synergistic effect of BMP4 on RA, "RA+BMP4" vs "RA" should be compared, not "BMP4" vs "BMP4+RA" (Fig 4d, 4f, & Extended Fig4b). Additionally, data in Fig4f show BMP4 only supplemented condition, but is not compared with no BMP4 condition. Together, the effect of BMP4 is not determined, therefore, additional data to support BMP4 addition at D5-D7 in this model is needed.
4. As aforementioned, one of the most interesting aspects of this study is the interaction of PE/STM and CM, i.e. hHO, but in the second half of this study, the authors explore more on the signaling pathways involved in the specification of PE/STM over CM and concluded that CHIR/BMP addition in the early stage of development activated RA signaling that resulted in that favored more PE-like phenotype. However, the authors fail to address questions like; Is CHIR alone or BMP alone also efficient or does it require both CHIR and BMP? Can RA exposure from early stage replace CHIR and BMP that result in RA activation? Moreover, the modifications made in the differentiation process was only analyzed by gene/protein expression changes but not functionally, like by analyzing the efficiency of the myocardium-epicardium two-layer formation when combined with CM.

Minor issues

1. In general, it is easier for the readers to understand the results when the immunofluorescent images are presented side by side with its experimental controls. These data are also essential for analysis and appropriate interpretation. For example, it would be better if Fig 4d and 4f are combined and presented side by side (BMP4+RA vs BMP4) etc.

2. Overall, excessive use of abbreviations in the text and figures makes the context difficult to follow., For example:

Fig1&2: "PE/STM" in Fig1, but the same organoid is renamed as "PE/STM/PFH" in Fig2

Fig2: "hHO" and "MT" is not defined in the figure or the legend, but only in the main text.

Fig5: The experimental setup for this figure is difficult to follow in part since "D3CB, D3C, D5CB and D5C" are not defined in the figure or the legend. C for "Control" and CB for "CHIR and BMP" is misleading since some readers could interpret "C" as "CHIR only".

Extended Data Fig 3d: "hHO" and "Heart Organoids" are both used

Extended Data Fig 5a: "IND" is not defined.

3. The embryonic days stated in the introduction could be speculated, but the authors should clarify the animal "mouse" somewhere in the text. Additionally, some explanation whether the PE organ is conserved across species is preferable.

4. The mixed use of cTnT (Fig1b) and TNNT2 (Fig1c), protein and gene, is difficult to interpret for the readers in other disciplines. Also, mixed use of cTnT (Fig1b) and CTNT (Fig2f) is confusing.

5. The authors should comment in the text if there is a rationale behind the termination of differentiation at Day 11 (Fig1a) compared to Day15 in the original protocol which they refer to (Bronco et al.)?

Reviewer #2 (Remarks to the Author):

The manuscript entitled "Human Multilineage Organoids Recapitulate Pro-Epicardium/Septum Transversum/Posterior Foregut" describes an advanced approach for recapitulating in vitro the embryogenesis of closely developing lineages that is the pro-epicardium (PE), the septum transversum mesenchyme (STM), and the posterior foregut/hepatic diverticulum (PFH). Applying and modulating human pluripotent stem cells (hPSCs) differentiation in a 3D environment, a platform to generate self-organized hPSC-derived organoids is provided, recreating early embryonic structures combining pro-epicardium/septum transversum populations and posterior foregut/hepatic primordium (PE/STM/PFH).

Although none of the applied differentiation strategies and the resulting cell lineages is entirely new, the study is providing substantial advancements to the field and sparks new ideas for studying in the complex mechanisms of early mesendoderm induction, mesoderm and endoderm specification, as well as cardiogenesis and foregut development.

The experimental strategy provides a "Lego-like" approach to modulate (promote versus inhibit) the propensity of distinct mesendodermal structures that is cardiac versus foregut endoderm lineages, enabling welcome opportunities for controlled lineage specification and studying their interplay. In contrast to many other organoid models, the described approach is Matrigel independent. This facilitates the experimental handling, modulation and monitoring of the model and supports the identification of cellular and molecular mechanisms directing the induction of individual lineages, independent of Matrigel-dependent unknown factors.

The manuscript can be roughly divided into 3 parts: 1. Establishing PE/STM/PFH organoids, 2. Establishing "heart organoids" (hHOs) by combining cardiomyocyte only aggregates with PE/STM/PFH

organoids, and 3. Modulations of the underlying hPSC differentiation protocol at early or later stages (i.e. before or after differentiation day 5) to better define signals impacting on PE/STM/PFH and cardiomyocytes (CMs) specification, differentiation and structural organization within organoids.

Major questions and remarks:

Focusing on PE/STM/PFH organoids in Fig.1: For how long can these structures being captured without combination with CMs? It seems mechanistically interesting to test whether the cells/tissues identities in PE/STM/PFH organoids remain stable over time independent of CMs or depend on signals from CMs for long-term stability and eventually progression of differentiation / maturation.

Controlling a precise number of aggregated cells has been described to be important for the specific and reproducible induction of organoids in general and for proper cardiac organoid patterning in particular (e.g. doi: 10.1038/s41596-021-00629-8). What is the role of cell numbers (per aggregate) for differentiation into PE/STM/PFH organoids and subsequent formation of hHOs?

Moreover, a quantitative assessment of the success rate and intra-/inter-experimental reproducibility of the several organoid models across the manuscript is important, including definition of respective quality criteria. Simply stating that the protocol is "robust and reproducible" is insufficient. Was hPSC-line dependence of organoid formation tested?

Results shown in Fig 2 maybe introduced by a schematic in that figure, indicating how the co-culture of PE/STM/PFH cells with CMs is performed for resulting into hHOs.

Authors noted: "As control, CM aggregates were dissociated and reaggregated at the time of co-culture establishment." However, was the dissociation/ re-aggregation of PE/STM/PFH organoids tested as well?

It was noted: "Over time in culture, the PE/STM/PFH cells were observed to migrate and surround the contracting core of the hHOs." However, this intermediate development is not displayed in the manuscript/ figures although it seems of core interest.

Generally, given the prior published studies, the term "heart organoid (hHOs)" used in the manuscript is somewhat unspecific and misleading; alternative designations should be considered.

From the IF staining of hHO sections in Fig. 2b, it is not clear whether – within the center of the myocardial-like zone (MZ) / myocardial tissue (MT) – there is the formation of a cell-free cavity or rather presence of cTNT negative non-cardiomyocytes?

The statement at the end of page 7: "These structurally organized hHOs show preliminary evidences that point for the recapitulation of important features already reported regarding the role of epicardium on myocardium proliferation, compaction, and maturation." is only poorly supported by the data presented in fig.2.

It was noted that PE/STM differentiation induces a population of CD31+/WT1- endothelial-like cells organized in a concentric pattern which may comprise vascular-like cells that are responsive to VEGF stimulation. However, did authors try to further specify these CD31+ cell to test their relation to endocardial cell?

A more "head-to-head" comparison should be provided for hHOs (resulting from mixing CMs with PE/STM/PFH) shown in Fig 2. versus organoids revealed in Fig 4., which apparently represent the co-emergence of CMs and PE/STM cell populations, resulting from differentiation protocol modulations (WNT, BMP, RA) on d5-7.

Moreover, the Results part downstream of "Removal of WNT signaling activation during PE/STM induction allows the co-emergence of CMs in the multilineage organoids" is difficult to follow and should re-structured / shortened.

What is the designation of organoids presented in Fig. 4 d-f?

In the last Result paragraph, authors noted: "To evaluate the impact of the new progenitor cell population (CB condition) on PE/STM and CM specification, the outcome of PE/STM induction and CM commitment starting from this new cell population was evaluated (Fig. 6a)." It remains unclear to this reviewer what the expression "new progenitor cell population (CB condition)" means. This should be better defined.

Given the relative complexity of the paper and the high number of cell lineages (including progenitor and intermediate stages) and tissues, the introduction of table maybe useful to systematically structure the paper. This should include: Designation of the several differentiation strategies and designating of resulting organoid models, definition of the respective lineage phenotypes and their characteristic marker(s) combination(s), indication of putative lineage counterpart(s) established in embryogenesis.

In the Discussion, authors noted: "The concept of multilineage organoid has been also recently introduced in the cardiac field by Drakhlis and colleagues that reported an organoid recreating both heart and foregut development (Drakhlis et al., 2021). Although the early stages of differentiation, consisting of a simple WNT signaling modulation, are similar to our protocol, the lack of an additional step for PE/STM specification generated an organoid featuring instead anterior foregut and myocardial-like layer. Those organoids do not present a clear organization or representation of what is described in vivo for PE, STM and posterior foregut/hepatic-like cells, contrarily to the present PE/STM/PFH organoids that represent a well-defined and in vivo-like organized structure that resemble PE/STM and posterior foregut development." Actually, the publication by Drakhlis et al., 2021 reveals the structures formation and spatial distribution of STM-like cells in neighborhood to posterior foregut endoderm (PFE) representing liver anlagen outside a myocardial layer, thereby closely recapitulating native embryonic patterns. This should be more correctly referred in the submitted manuscript.

In general, the Result and Discussion part is somewhat lengthy, substantially redundant and needs curtailment. Moreover, the paper would benefit from a model integrating the key findings. Authors noted in the discussion: "Characterization of PE/STM/PFH organoids, through single cell transcriptomic analysis, may help to decipher in more detail the different mesoderm-derived subpopulations within PE/STM/PFH organoids and identify new reliable markers that can allow the precise characterization of each sub-types of mesoderm-derived cells." The lack of scRNAseq to confirm (and better defined) the identity of individual cell lineages is indeed a major shortcoming of the manuscript.

Minor:

Typo in the introduction: second heart field (SHF)

Figure 3d: pls double-check designation of IF staining; indication of the DAPI channel deems partially incorrect

Reviewer #3 (Remarks to the Author):

In this manuscript the authors describe a new 3D organoid model of pro-epicardium, septum transversum mesenchyme, and posterior foregut/hepatic cell (PE/STM/PFH) development. They describe a spatial organization of the 3 cell types similar to embryonic development in vivo. Using this model, the study provides insights that are suggestive of the development of PE/STM progenitors from posterior second heart field. The specification of these progenitors was found to be driven by RA signaling based on bulk RNA sequencing analysis. To provide a functional application of these PE/STM/PFH organoids the authors co-cultured them with aggregates of cardiomyocytes. These experiments showed that the PE cells can self-organize into an outer layer of epicardial cells and result in the formation of complex heart organoids.

The major novelty of this work is the detailed characterization of an organoid that contains the three cell types (PE/STM/PFH). While this is an interesting model the study has a number of limitations that need to be addressed.

Major comments:

1) Organoids are great models of development and the representative images shown in the paper demonstrate the overall 3D structure as summarized in the little schematics for example in Figure 1. However, the authors fail to show how reproducible their organoid system is. It is well known that a huge variation of efficiency in organoid formation exists, depending on the system used. The manuscript would greatly benefit from additional data demonstrating how robust the organoid formation is. In other words, how many aggregates are developing the gradient of WT1 expression with PE cells in the outer periphery, followed by STM cells that are engulfing an inner gut epithelial lumen. The authors could address this in several ways e.g. i) by showing images of 10 independently generated organoids; ii) by quantifying the expression gradients of WT1, LHX2, E-CAD throughout the organoids as done in Supp. Fig 2d; and iii) by providing data on the percentage of organoids that form a lumen, and their general shape (round, ovoid, elongated) and size in the horizontal and vertical axis. Excellent examples for this can be found in recent organoid/gastruloid publications from Rossi et al. Cell Stem Cell 2021, PMID: 33176168, Figure S1 and S2; and Moris et al. Nature 2020, PMID: 32528178, Figure 1e, Extended data Figure 1).

2) The authors rely heavily on WT1 and LHX2 as markers for PE vs STM.

a. In Figure 1 staining for WT1 and LHX2 is shown on two different organoids and the conclusion is made the WT1+ cells are localized in the outer rim while LHX2+ cells are expressed in the inner layer surrounding the epithelial gut-like structures. Given that this a central finding of the paper the authors should show this more convincingly by developing a co-staining of WT1 and LHX2 that allows to show clear overlap / separation of the two markers. This is not only relevant for the data presented in Figure 1 but throughout the manuscript. For example, in Figure 4d consecutive sections are stained for WT1 and LHX2 and the conclusion is made that "IF staining revealed the presence of LHX2+/WT1- cells surrounded by a thin layer of WT1+/LHX2- cells (Fig. 4d I, I', II, II' and Extended Data Fig. 4a I, I')". Based on the figures provided it is difficult to appreciate that. In fact, it looks like the WT1+ and LHX2+ cells are overlapping.

b. Using only 2 markers to distinguish PE and STM cells is not ideal. The authors mention additional PE (UPK3B) and STM (HLX1) markers in the manuscript. Adding stainings for these additional markers would strengthen the presented data.

3) To show the functionality of the PE/STM/PFH organoids the authors choose a co-culture system with cardiomyocytes and created a heart organoid. On page 7 the authors make the conclusion that: "PE/STM/PFH cells have the capacity to self-organize and migrate, generating a WT1+-epicardial-like layer surrounding a cTnT positive myocardium like tissue (Fig. 2f)". With the current experimental setup this conclusion cannot be made. In order to make conclusions about the contribution of the PE/STM/PFH cells to the heart organoid the PE/STM/PFH organoids should be generated from a fluorescently labelled cell line (e.g. RFP) that can be lineage traced. With the current dataset it can't be excluded that the WT+ epicardial cells seen in the heart organoids were derived from cells of the cardiomyocyte cultures via inductive signals provided by the PE/STM/PFH cells. In addition, this heart organoid dataset would benefit from a more detailed display of reproducibility along the lines of the first comment.

4) The authors then go on to show that the heart-organoids show signs of cardiomyocyte maturation. While the EM data presented in Extended Data Figure 3d-e is promising more analysis is required to convincingly show a matured cardiomyocyte phenotype.

a. The authors state that the cardiomyocytes show "more compact phenotype, with denser fiber content, and CMs appeared also more aligned and following the curvature of the hHOs." This can't be appreciated from the low magnification images presented in figure 2. The authors should consider providing higher magnification images and to quantify the fiber content and fiber alignment.

b. Markers for trabecular (NPPA, BMP10) vs compact (Hey2, MYCN) cardiomyocytes have been

established (Funakoshi et al. Nat Commun 2021, PMID: 34039977). The authors should use these markers to demonstrate a compact vs trabecular phenotype of the cardiomyocytes in the heart organoids.

c. To further show a more mature phenotype IHC for the marker of immature cardiomyocytes TNNI1 and the marker of mature cardiomyocytes TNNI3 should be performed and quantified. Mature cardiomyocytes should have an increase in the TNNI3/TNNI1 ratio compared to immature controls (Guo et al. Cir Res 2020, PMID: 32271675)

5) The authors further show that the epicardial cells induce proliferation in the out layer of cardiomyocytes in their heart organoids (Fig. 2c). Have the authors checked whether the reduced cardiomyocyte proliferation in the core of the heart organoid is caused by cell death due to lower perfusion of nutrients to the core of the organoid. Additional data providing the size of the heart organoids to address perfusion concerns as well as TUNEL staining to check for cell death in the core of the organoid would be useful.

6) As part of their analysis of the progenitors that are giving rise to the PE/STM/PFH, the authors activate BMP and WNT signaling between day1 and 3 and performed bulk RNAseq expression analysis comparing control and extended BMP+WNT treatment (CB condition). The bulk RNAseq data revealed interesting changes in expression including a shift towards posterior second heart field markers (pSHF), reduction in endoderm markers and reduction in cardiac mesoderm markers. This part of the manuscript would highly benefit from analysis on the single cell level for example by flow cytometric analysis of the proportion of pSHF cells, endoderm progenitors and cardiac mesoderm cells. Alternatively, single cell RNA-sequencing could be considered. This analysis on the single cell level would allow a more conclusive demonstration of the enrichment of a pSHF progenitor population.

7) The authors then go on to analyze the developmental potential of the new "pSHF enriched" progenitor population generated under the CB condition. They report a reduced potential for cardiomyocyte differentiation and an increased potential for PE/STM differentiation. While this is an interesting observation the presented data does not warrant the statement made in the abstract that "evidences for a posterior second heart field/splanchnic mesoderm origin of the PE/STM" are provided in the manuscript. To this end a lineage tracing strategy should have been employed. I agree that the authors have developed a powerful model to analyze the early developmental stages of PE/STM development. However, their experiments in this part of the manuscript fall short of demonstrating this. The manuscript would be a lot more impactful if pSHF progenitors could for example be enriched by sorting for a surface marker, and subsequent analysis of the positive and negative sorted populations for their potential to develop into PE/STM cells. If a clear enrichment of PE/STM cells from the positive sorted pSHF progenitors vs negative sorted progenitors could be shown, this would provide a clear demonstration of their developmental origin. Alternatively, lineage tracing through fluorescent reporters for the pSHF lineage could be used to address this question. Lastly, single cell RNA-sequencing at multiple timepoints throughout the organoid development could also be considered as a powerful tool to reconstruct the development and progenitors of PE and STM. Either of these 3 approaches would be highly beneficial to improve the impact of the manuscript.

8) On page 12, 3rd paragraph the authors draw the conclusion that RA signaling drives the specification of the LMP progenitor to the pSHF phenotype : " induce up-regulation of the RA pathway, which may originate the subpopulation of early paraxial mesoderm progenitors, and consequently drive the LPM progenitor cell population from cardiac mesoderm to a posterior splanchnic/SHF mesoderm specification". Beyond showing the upregulation of enzymes regulating RA signaling (ALDH1A2) and RA target genes (NR2F2, several HOX genes) no further experimental proof for this statement is provided. To clearly show the impact of RA signaling on the specification of the pSHF in their model system the authors should consider inhibiting RA signaling (culture in RA & Retinol free media, or use of small molecule inhibitors such as BMS 493) and activating RA signaling (all-trans-Retinoic acid) during the day1-5 time window.

9) On page 9 the authors state: "these results demonstrate that inhibition of the WNT pathway during this specific stage of differentiation compromise the WT1+ lineage ..." This conclusion cannot be made from the presented data because a WNT inhibition was not experimentally tested. Instead, the WNT activation by CHIR was removed which would have resulted in presence of endogenous WNT signaling levels. Endogenous levels are not the same as complete inhibition of WNT signaling. The authors should therefore consider rephrasing this statement to for example: Endogenous WNT levels are not sufficient to support development of the WT1+ lineage.

A similar refinement of the statement on top of page 10 "that the elimination of WNT signaling between D5-D7 ..." is required.

10) On page 9, bottom the authors state: "This observation highlights the fact that the LHX2 and WT1 positive cells observed in the condition combining BMP4 and RA toward PE/STM induction are mainly dependent on the RA signal for their specification, reinforcing again the role of RA on the generation of a PE/STM population." The authors never tested the effect of BMP + RA treatment vs BMP only treatment. All experiments that manipulated RA signaling were done in the presence of CHIR. Therefore, the statement should be rephrased to state that RA signaling is important in the presence of BMP and CHIR signaling.

Minor comments:

1) On page 7, bottom of first paragraph the authors state: "absence of vascular cells in control CM aggregates (Extended Data Fig 3g,h). The CM aggregates shown in the figure clearly show CD31+ signal. Therefore, this statement needs to be revised.

2) In Figure 2e the co-staining of WT1 and CD31 is not clearly visible. It would help if the authors could provide higher magnification insets for this dataset.

3) In Extended Data Figure 3c the authors show CX43 staining and make the conclusion that the staining is "more pronounced in the region contacting the EL". This conclusion is not supported by the provided images and a quantification of CX43 staining throughout the heart organoid should be considered, if the authors want to make this statement.

4) In Figure 3c the cell types that are present at day 5 are analyzed using flow cytometry. In the manuscript it is stated that the PDGFRa+ mesoderm cells make up 66% of the total cells. In Figure 3c the PDGFR bar seems to be at ~75%. Please clarify this discrepancy.

5) On page 9, 2nd paragraph the authors are providing values for WT1+ cells in CHIR and BMP4 vs CHIR only supplementation: "Removal of RA supplementation from D5-D7, revealed a significant decrease in the percentage of WT1+ cells, more accentuated in the condition with combined CHIR and BMP4 (53±5% WT1+) compared with only CHIR supplementation (63±4% WT1+)" The actual baseline values of WT1+ cells that are present in conditions with RA supplementation are not provided to the reader. Please add these control values in brackets as well e.g. (75±5% vs 53±5% WT1+).

6) On page 9, bottom the authors refer to the endothelial cells that are present in the heart organoid as "myocardial-like endothelial cells". No data is provided to provide proof that these endothelial cells have a myocardial phenotype. The authors should therefore consider removing the "myocardial-like" wording and just refer to the cells as endothelial cells.

7) On page 10, 2nd paragraph the authors introduce the prolonged low level WNT activation combined with BMP4 from day1-day3. It is not immediately apparent to the reader that the initial 11uM CHIR step from day0-1 was kept for both conditions. I recommend including a small schematic of the protocol on top of Figure 5 similar to what the authors did for Figure 6. In addition, the naming of the conditions as CB and C further contributes to the confusion. A clearer labelling for example as C-CB

and C-0 condition would be helpful.

8) On page 10 the authors state: " However, in D5CB condition we did not observe increased expression of the early somite progenitor markers PARAXIS (TCF15) and FOXC2 (Loh et al., 2016),...". The data supporting this statement was not provided. Please include the data in the extended data section.

9) The methods section does not specify what size of AggreWell plates were used. Therefore, the information is not sufficient to recapitulate aggregate formation with the same cell densities. Several studies have shown that the number of cells per aggregate critically impact the outcome of organoid formations. The authors should therefore add this important information.

10) Also related to the aggregate formation the schematic in Figure 1a would benefit from additional detail that shows that these cultures were performed in aggregates. For example, a little "comic" of the cells that shows the culture format throughout the protocol below the existing schematic would be helpful.

11) A couple of datasets / figures are missing statistical analysis. For example, Figure 1c + Extended data Figure 1 a-c and Figure 6b should include statistical comparison between the displayed experimental conditions.

12) Some of the in-text references to Figures are not correct. For example, on page 5, first paragraph Extended data Fig. 3b – should be Fig 3a. There are a couple of these mismatches throughout the manuscript. The authors should go through the manuscript and make sure that all the references are correct.

13) The same nomenclature of ', " is used to indicate staining of consecutive sections (Figure 4d) as well as insets into images (Figure 1e). I recommend using different symbols to indicate insets vs consecutive sections to avoid confusion.

Answers to Reviewers' Comments

Below please find our answers to reviewers' comments concerning the manuscript entitled "Human Multilineage Organoids Recapitulate Pro-Epicardium/Septum Transversum/Posterior Foregut and Support the Development of Epicardium-Myocardium Organoids". First of all, we would like to deeply acknowledge all the comments and questions made by the reviewers. We truly believe that they allowed us to improve the clarity and the impact of our work and to enrich some of the studies that were incomplete or poorly explored, such as, for example, the ones related to epicardium-myocardium interaction. We sincerely hope that the changes that were made throughout the manuscript will clearly address all the comments and questions that were raised. We think that in its actual form, this manuscript has now a major focus not only in the development of the PE/STM/PFH organoids but also in the heart organoids, reason why we propose a small change in the title of the manuscript to more accurately reflect the content of the work.

Reviewer #1 (Remarks to the Author):

This manuscript provides a cutting-edge organoid model that recapitulate the pro-epicardium migration to form two clear myocardium-/epicardium- like layers, an essential phenomenon that occur in early heart development. Strengths of the study include the novelty of the multilineage nature of the organoid, and the characterization of the spatial relationships of different cell compositions in the model that could be used as an in vitro tool for myocardium-epicardium interaction and the heart organogenesis. Having said that, the myocardium-epicardium interaction of this organoid is poorly studied, and the advantages have not been described in an adequate manner. Rather, the work focuses more on BMP/WNT/RA effect in the early stages of differentiation of the pro-epicardium/septum transversum mesenchyme and the cardiomyocyte, which could potentially be done in 2D monolayer assays, and the conclusion is mostly confirmatory. In addition, there are several weaknesses in the study, and the value of the data presented unfortunately lack information for proper interpretation and to support the conclusion of the paper.

Major concerns

1. The authors conclude that the "hHOs clearly demonstrate the direct impact of epicardial cell on myocardial-like tissue organization and maturation (Page 16)", but the only "maturation" data the authors showed was the sarcomere length in Extended data Fig 3. They could elucidate more on the advantages of "co-culturing PE/STM with CM (hHO)" over "CM aggregate only" (Fig2). For example, is the proliferative CMs (Ki-67+) increased with the co-culture with PE/STM? Are there any differences in the ventricular gap junction (CX43) pattern? Are any of the cardiomyocyte "maturation" markers increased? What differences can you observe in long term cultures (e.g. day 17) in hHOs compared to CM aggregate only or PE/STM aggregate only (Extended Fig 2h)? The quantification for WT1 within MT could be done for CM as well as hHO and presented side by side (Fig2d) to emphasize the increase by PE/STM addition.

We agree with the reviewer that a more thorough characterization of the impact of co-culturing CMs and PE/STM/PFH on CM maturation should have been performed. In order to address this issue, we performed additional experiments that are now presented in new Figure 4, Figure 5 and Extended DATA Fig 6, 7 and 8.

As can be seen in Fig. 5 and Extended Data Fig. 8, immunofluorescence (IF) staining of hHOs (now entitled EMOs in the new version of the manuscript) sections shows an increased expression of the mature ventricle CM marker MLC2V, mainly near the WT1⁺ epicardial-like layer, compared with age-matched CM aggregates, in which MLC2V staining was not detected.

We additionally performed IF staining for CX43 and Ki-67 markers in CM aggregates to allow a side-by-side comparison with hHOs and highlight the differences. As can be seen in Fig. 5 and Extended Data Fig. 8, CM aggregates present a lower expression of the ventricular gap junction marker CX43 when compared with hHOs.

Regarding the analysis of proliferative Ki-67⁺CMs, we performed additional IF staining for this marker in different hHOs, as can be seen in Fig. 4 and Extended DATA Fig 7. As can be observed in these figures, in hHOs there is a higher number of proliferative CMs near the epicardial-like layer and an almost absence of proliferative CMs towards the center of the organoids. These results are consistent with the characteristic increased CM proliferation observed at early stages of embryonic myocardium growth development near the epicardial layer and promoted through paracrine signals release from epicardial cells. On the opposite

side, in CM aggregates, it was observed a homogeneous distribution of proliferative CMs throughout the entire aggregate, without a specific distribution pattern (Fig. 4d, control).

We also added new IF images, and the respective higher magnifications, for specific areas of the hHOs, to reinforce our claim that, near the epicardial-like layer, CMs show stronger staining for cTnT and there is a lower nucleus density (Extended Data Fig. 6i), which suggests that CMs present a more compact fiber content, indicating a higher structural maturation degree of the CMs in the co-culture system (Page 10, line 284-288)

Finally, we complemented these studies with a functional analysis of hHO and CM aggregates spontaneous contraction profile, with and without drug stimulation (Fig. 5d and Extended Data Fig. 8c). From this functional analysis we observed that hHOs present the expected response to all the tested drugs, with control CM aggregates showing a not so pronounced effect upon verapamil and E-4031 treatment, which may indicate that CMs present in control condition lack or exhibit a lower expression level of hERG and Ca²⁺ channels, which may indicate a lower functional maturation compared with CMs present in hHOs. Altogether, we believe these results demonstrate a clear impact of the co-culture on myocardium-like cells structural and functional maturation.

Regarding the study of the effect of the long-term culture of hHOs, we extended the culture of these organoids until D50-60 in basal media without exogenous stimuli, and performed IF staining of organoid sections at this stage of co-culture (Extended DATA Fig. 8d). We concluded that the organoids tend to lose the clear myocardial-epicardial organization observed at D15 of co-culture, which indicates that the culture conditions must be optimized to sustain the initial structure. Nevertheless, we observed that these organoids maintain their viability. Regarding the long-term culture of the CM aggregates used as control, we observed that they do not remain viable for more than 30 days in culture, probably due to the lack of extracellular matrix to support aggregate structure and function. This result also demonstrates that the multicellularity and the self-organized structure observed in hHOs bring additional advantages for future in vitro applications. In order to address the last comment of the reviewer regarding the possibility of quantifying the WT1+ cells in CM aggregates, we would like also to clarify that in control CM aggregates there are no WT1 cells, as can be seen in Figure 4c (control). Finally, regarding the PE/STM/PFH organoids, after 30 days in culture, these organoids show a considerable growth of epithelial cells, with a more posterior endoderm signature (CDX2⁺/AFP⁺), the maintenance of the LHX2, WT1 and CD31 cell populations, and their structural organization (Extended DATA Fig. 4e).

2. The analysis and conclusions of STM characterization is concerning. ISL1 and LHX2 are the two mainly used markers in this study to characterize STM, but they are also known markers for pharyngeal mesoderm Harel et al. (doi: 10.1073/pnas.1208690109).

We are aware that ISL1 and LHX2 markers are not specific for STM. Thus, although we only used ISL1 and LHX2 to perform IF staining for STM characterization, we presented additional data that demonstrate a significantly increased expression of an additional STM marker, HLX1, in PE/STM/PFH organoids in comparison with age-matched CM aggregates, by PCR analysis (Extended DATA Fig 2h). This represents an additional evidence pointing for the presence of a STM population in the PE/STM/PFH organoids. Adding to that, the allegation of the existence of a STM cell population in PE/STM/PFH organoids is also strengthened by the presence of pro-epicardial and posterior foregut/hepatic diverticulum-like populations, and their *in-vivo*-like spatial organization. In fact, the spatial arrangement of these three different major populations in the organoids is in agreement with what is described for PE, STM and PFH during embryonic development in mouse.

Interpretation of the increase of LHX2 and ISL1 should be done carefully preferably supported with additional markers (e.g. in Fig 4).

The former Fig. 4 (new Fig. 3), and the respective results section, were extensively reviewed. In particular, we performed additional studies and we re-wrote the paragraph describing the results obtained when WNT signaling is not activated during the PE induction period (D5-D7). We included additional IF staining for the “RA+BMP4” condition at D17 of differentiation that shows the presence of a LHX2⁺/WT1⁻ STM-like region, not observed in PE/STM/PFH organoids and a ISL1⁺ progenitor cell population that overlaps with cTnT⁺/NKX2.5⁺ CMs, both not observed in PE/STM/PFH organoids (Fig. 3g, and Extended Data

Fig 5c and d). These observations may justify the observed PCR results that showed an increased expression of ISL1 and LHX2 in RA+BMP4 condition compared with control CHIR+RA+BMP4 (PE/STM/PFH) (Extended DATA Fig 5e). In both RA and RA+BMP4 conditions, the percentage of cTnT increased from D11 to D17 of the differentiation (Fig. 3e), which may indicate CM proliferation or later specification of ISL1+ progenitor cells into CMs.

3. The advantages of adding BMP4 at D5-D7 of PE/STM differentiation is not clearly shown. Unfortunately, there is no data that support the "positive impact of RA signaling on specification of WT1+ cells in the presence of BMP4" (Page 9) since the authors refer to data that compares only "CHIR+BMP4+RA" vs "CHIR+BMP4" vs "CHIR" (Extended Fig4b). To conclude the synergistic effect of BMP4 on RA, "RA+BMP4" vs "RA" should be compared, not "BMP4" vs "BMP4+RA" (Fig 4d, 4f, & Extended Fig4b). Additionally, data in Fig4f show BMP4 only supplemented condition, but is not compared with no BMP4 condition. Together, the effect of BMP4 is not determined, therefore, additional data to support BMP4 addition at D5-D7 in this model is needed.

We agree that the impact of BMP4 and RA addition during PE induction phase was not clearly demonstrated with the data presented in former Fig. 4. Thus, this figure has been extensively changed to clarify the precise impact of each signaling pathway manipulated during the PE induction period. These results are schematically summarized in a new illustration (Extended Data Fig. 10). The new Fig. 3 and Extended Data Fig. 4 and 5 present additional data that we believe clearly demonstrate that WNT signaling activation between D5-D7 is critical for the generation of the WT1 positive cells (Fig. 3a). However, the sole activation of WNT signaling, although not affecting the percentage of WT1+ positive cells, results in the development of organoids with a more hepatic phenotype (Fig. 3b, c and d). We additionally highlighted that the combined activation of WNT and RA is sufficient to generate PE/STM/PFH organoids, which indicates the crucial impact of RA signal and the reduced effect of BMP4 activation on the generation of these organoids (Extended Data Fig. 4c and d). Regarding the impact of performing "only RA" activation during the PE induction period, we observed the generation of organoids that present a small percentage of WT1+ cells and the co-generation of cTnT+ cells. We added IF images of organoids at D11 and D17, revealing that a subcluster of the WT1 positive cells, which do not co-stain with LHX2, line the periphery of the organoids, including the cTnT positive region and the LHX2+/WT1- STM-like area (Fig. 3f and g, and Extended Data Fig 5c and d). Additionally, it was also observed the presence of an ISL1+ progenitor cell population that overlaps with cTnT+/NKX2.5+ CMs. The combined activation of "RA+BMP4" resulted in similar organoids compared with "only RA", as showed by the flow cytometry results and representative BF and IF images.

4. As aforementioned, one of the most interesting aspects of this study is the interaction of PE/STM and CM, i.e. hHO, but in the second half of this study, the authors explore more on the signaling pathways involved in the specification of PE/STM over CM and concluded that CHIR/BMP addition in the early stage of development activated RA signaling that resulted in that favored more PE-like phenotype. However, the authors fail to address questions like; Is CHIR alone or BMP alone also efficient or does it require both CHIR and BMP?

To address the relevant point raised by the reviewer, we present new data showing the impact of CHIR/BMP4 addition between D1 and D3 (Extended DATA Fig 9). As described in the revised version of the manuscript (Page 13, Line 407- 413), it was demonstrated that although CHIR alone allowed a significant decrease in endoderm progenitor cells (Extended DATA Fig 9a), the combination with BMP4 activation promoted a statistically significant decrease in c-KIT+ cells at D5 of the differentiation (Extended DATA Fig 9b).

"Can RA exposure from early stage replace CHIR and BMP that result in RA activation?"

We performed additional experiments to address this interesting question. In particular, we presented data showing that the direct activation of RA between D1 and D3 of differentiation affects CM specification similarly to what happens when performing CHIR+BMP4 supplementation within the same time frame (Figure 6l and Extended Data Fig. 9d). This reinforces the hypothesis raised from RNA-seq data that pointed for RA as one of the stimuli driving the observed differences in CM and PE specification from D5C and D5CB (mentioned as D5 and D5WB in the new version of the manuscript) progenitor cells.

“Moreover, the modifications made in the differentiation process was only analyzed by gene/protein expression changes but not functionally, like by analyzing the efficiency of the myocardium-epicardium two-layer formation when combined with CM.”

The comment raised by the reviewer is very pertinent and this would certainly constitute a very interesting study. The main focus of this manuscript was the development and characterization of the new PE/STM/PFH organoids and the establishment and characterization of a platform to generate physiological relevant in-vivo-like epicardium-myocardium organoids (the hHOs, now designated EMOs). We believe that our new developed co-culture platform opens interesting paths that could be further explored by us and others with different CM and PE organoid models, such as the new PE organoid that we obtained after progenitor cell modulation, but we believe that including these additional data would go beyond the main objectives of the present manuscript.

Minor issues

1. In general, it is easier for the readers to understand the results when the immunofluorescent images are presented side by side with its experimental controls. These data are also essential for analysis and appropriate interpretation. For example, it would be better if Fig 4d and 4f are combined and presented side by side (BMP4+RA vs BMP4) etc.

We agree with the reviewer and thus the former Fig. 4 (now Fig. 3) was re-structured to comply with this comment.

2. Overall, excessive use of abbreviations in the text and figures makes the context difficult to follow., For example:

- Fig1&2: "PE/STM" in Fig1, but the same organoid is renamed as "PE/STM/PFH" in Fig2

In Fig. 1 we changed the nomenclature and instead of mentioning PE/STM we changed to PE protocol. Only from the second result section beyond, when we demonstrated the presence of the different subpopulation (STM and posterior foregut/hepatoblast-like cells), we start referring the organoid as "PE/STM/PFH".

- Fig2: "hHO" and "MT" is not defined in the figure or the legend, but only in the main text.

We added that information regarding the meaning of the initials on the legend of former Fig. 2 (now Fig. 4).

Fig5: The experimental setup for this figure is difficult to follow in part since "D3CB, D3C, D5CB and D5C" are not defined in the figure or the legend. C for "Control" and CB for "CHIR and BMP" is misleading since some readers could interpret "C" as "CHIR only".

We agree with the reviewer that the nomenclature used is misleading. We changed the "D3C" and "D5C" to only "D3" and "D5", for day 3 and day 5 controls; and changed "D3CB" and "D5CB" to "D3WB" and "D5WB" for day 3 and day 5 after CHIR and BMP4 supplementation from D1 to D3.

Extended Data Fig 3d: "hHO" and "Heart Organoids" are both used.

In order to homogenise the nomenclature, in the former Extended Data Fig. 3e (now Fig. 5) we replaced "Heart organoids" by EMOs.

Extended Data Fig 5a: "IND" is not defined.

We replaced the initials "IND", in former Extended Data Fig 5a (now Extended DATA Fig 9), by "induction".

3. The embryonic days stated in the introduction could be speculated, but the authors should clarify the animal "mouse" somewhere in the text. Additionally, some explanation whether the PE organ is conserved across species is preferable.

We added information on those two relevant points raised by the reviewer in the introduction section (Page 2, lines 39-42, 53 and 57)

4. The mixed use of cTnT (Fig1b) and TNNT2 (Fig1c), protein and gene, is difficult to interpret for the readers in other disciplines. Also, mixed use of cTnT (Fig1b) and CTNT (Fig2f) is confusing.

In order to facilitate the interpretation of the text, we established the use of cTnT for protein (IF staining) and TNNT2 for gene (PCR analysis) throughout the entire manuscript.

5. The authors should comment in the text if there is a rationale behind the termination of differentiation at Day 11 (Fig1a) compared to Day15 in the original protocol which they refer to (Bronco et al.)?

The reviewer comment is very pertinent. In fact, there is rational for that decision. We decided to end the PE differentiation at D11 since this is the time point at which the percentage of WT1+ cells reached its maximum value and it was also the time point that shows the maximum expression of the PE integrin marker ITGA4 (Extended DATA Fig 6b), that has been described to be important to promote the interaction between PE cells and CMs. As requested by the reviewer, we provided this explanation in the revised version of the manuscript (Page 9, Line 263-267).

Reviewer #2 (Remarks to the Author):

The manuscript entitled "Human Multilineage Organoids Recapitulate Pro-Epicardium/Septum Transversum/Posterior Foregut" describes an advanced approach for recapitulating in vitro the embryogenesis of closely developing lineages that is the pro-epicardium (PE), the septum transversum mesenchyme (STM), and the posterior foregut/hepatic diverticulum (PFH). Applying and modulating human pluripotent stem cells (hPSCs) differentiation in a 3D environment, a platform to generate self-organized hPSC-derived organoids is provided, recreating early embryonic structures combining pro-epicardium/septum transversum populations and posterior foregut/hepatic primordium (PE/STM/PFH).

Although none of the applied differentiation strategies and the resulting cell lineages is entirely new, the study is providing substantial advancements to the field and sparks new ideas for studying in the complex mechanisms of early mesendoderm induction, mesoderm and endoderm specification, as well as cardiogenesis and foregut development. The experimental strategy provides a "Lego-like" approach to modulate (promote versus inhibit) the propensity of distinct mesendodermal structures that is cardiac versus foregut endoderm lineages, enabling welcome opportunities for controlled lineage specification and studying their interplay.

In contrast to many other organoid models, the described approach is Matrigel independent. This facilitates the experimental handling, modulation and monitoring of the model and supports the identification of cellular and molecular mechanisms directing the induction of individual lineages, independent of Matrigel-dependent unknown factors. The manuscript can be roughly divided into 3 parts: 1. Establishing PE/STM/PFH organoids, 2. Establishing "heart organoids" (hHOs) by combining cardiomyocyte only aggregates with PE/STM/PFH organoids, and 3. Modulations of the underlying hPSC differentiation protocol at early or later stages (i.e. before or after differentiation day 5) to better define signals impacting on PE/STM/PFH and cardiomyocytes (CMs) specification, differentiation and structural organization within organoids.

Major questions and remarks:

1. Focusing on PE/STM/PFH organoids in Fig.1: For how long can these structures being captured without combination with CMs? It seems mechanistically interesting to test whether the cells/tissues identities in PE/STM/PFH organoids remain stable over time independent of CMs or depend on signals from CMs for long-term stability and eventually progression of differentiation / maturation.

In order to address this comment, we prolonged the time in culture of PE/STM/PFH organoids up to 30 days (Extended DATA Fig 4e), without combination with CM aggregates and without additional exogenous signals. Overall, as main conclusions, these organoids show a considerable growth of epithelial cells, with a more posterior endoderm signature (CDX2⁺/AFP⁺), the maintenance of the LHX2, WT1 and CD31 cell populations, and their structural organization (Extended DATA Fig. 4e).

2. Controlling a precise number of aggregated cells has been described to be important for the specific and reproducible induction of organoids in general and for proper cardiac organoid patterning in particular (e.g. doi: 10.1038/s41596-021-00629-8). What is the role of cell numbers (per aggregate) for differentiation into PE/STM/PFH organoids and subsequent formation of hHOs?

In the case of PE/STM/PFH organoids, it has been previously demonstrated by Branco et. al 2019 that the initial seeding density, and consequently the aggregate size, is in fact critical to generate efficiently cardiac mesoderm. Thus, we fixed the diameter of hPSC aggregate size at D0 to 300-320 μm with an initial cell seeding density of 4000 cells, at D-3 of differentiation. We tested the impact of changing the initial seeding density from 3000-4500 cells at D-3 of differentiation but we did not notice significant changes regarding the structural organization of the organoids or the presence of the different identified cell populations in PE/STM/PFH organoids.

Regarding the hHOs (mentioned as EMOs in the new version of the manuscript), we tested different cell numbers per aggregate during the re-aggregation step and concluded that a number lower than 11.000 cells/aggregate compromise the hHOs structural organization that was reported in the manuscript. We also tested a range of 11-13.000 cells/aggregate, which allowed the successful formation of hHOs.

Moreover, a quantitative assessment of the success rate and intra-/inter-experimental reproducibility of the several organoid models across the manuscript is important, including definition of respective quality criteria. Simply stating that the protocol is "robust and reproducible" is insufficient.

Regarding the PE/STM/PFH organoids, in order to highlight the reproducibility between biological experiments and within the same experiment, we added images of 8 independently generated organoids from 4 independent biological replicates showing the structural similarity between different organoids (Extended DATA Fig 2a). The quality criteria were defined as (1) the presence of one lumen/epithelial structure, normally at the center of the organoid (highlighted in Extended DATA Fig 2a with white arrows), (2) a percentage of WT1 positive cells of $\pm 70\%$ (Fig. 1b), and (3) the presence of a gradient of WT1 expression inside the organoids from the periphery to the epithelial structure (Extended DATA Fig 2f, n = 14 PE/STM/PFH organoids derived from 3 separate differentiation experiments).

In addition to analysing the reproducibility of the organoids across distinct differentiation experiments, we also analysed their reproducibility across different hPSC lines. In Extended DATA Fig 1b we demonstrated the reproducibility of WT1⁺ cell induction for two additional hPSC lines. We added also additional IF data that demonstrate the reproducibility in terms of the structural organization observed in PE/STM/PFH organoids for those two additional hPSC lines (Extended DATA Fig 3).

Concerning the hHOs organoids, we added brightfield images of 10-12 independently generated hHOs from 3 independent differentiation experiments (Extended DATA Fig 6e), where it is evident, despite the intrinsic heterogeneity associated to organoid models in terms of organoid size and shape (Extended DATA Fig 6f), the formation of the two different regions: (1) the outer epicardial-like layer and (2) the core myocardium-like tissue. The existence of this structural organization was defined as the quality criteria. We also presented a representative brightfield image of hHOs that failed to comply with the previously mentioned quality criteria (Extended DATA Fig 6g). Although we do not present data in the manuscript regarding this matter, we found that the ratio between PE/STM/PFH cells and CMs is extremely important to reproducibly obtain the described arrangement of the cell populations (layer of WT1⁺ cells surrounding the myocardium-like tissue). In fact, as explained on the manuscript (line 261-263), we concluded that the optimal ratio of PE/STM/PFH per CMs cells is 90/10 (although 85:15 also works) while for increasing percentages of PE/STM/PFH cells, the cell arrangement is not observed. Moreover, it was also found that the time point to perform the co-culture is critical. In fact, we concluded that the reproducibility is high if the co-culture re-aggregation is performed when the expression of ITGA4 in PE/STM/PFH organoids is maximal (Extended DATA Fig 6b) and

also when CMs aggregates are already contracting and expressing VCAM. Within the same biological run, if the process of re-aggregation works well, the reproducibility between organoids is high, meaning that all the organoids meet the quality criteria.

Results shown in Fig 2 maybe introduced by a schematic in that figure, indicating how the co-culture of PE/STM/PFH cells with CMs is performed for resulting into hHOs.

To facilitate the understanding of the methodology used, we added a schematic illustration (new Fig. 4a and Extended Data Fig.10) that explains how the co-culture of PE/STM/PFH cells with CMs is performed.

3. Authors noted: "As control, CM aggregates were dissociated and reaggregated at the time of co-culture establishment." However, was the dissociation/ re-aggregation of PE/STM/PFH organoids tested as well?

In order to address this comment, we performed extra experiments of re-aggregation of PE/STM/PFH organoids at D11 of differentiation followed by 15 additional days of culture. The re-aggregated PE/STM/PFH organoids were able to form again the epithelial-like structures, demonstrating the self-organization capacity of these cells. Interestingly, upon re-aggregation, each organoid formed frequently contained more than one epithelial structure (Extended Data Fig. 6d).

It was noted: "Over time in culture, the PE/STM/PFH cells were observed to migrate and surround the contracting core of the hHOs." However, this intermediate development is not displayed in the manuscript/ figures although it deems of core interest.

We agree with the reviewer that the existence of a process of migration is not clearly demonstrated in the manuscript/ figures since we do not use reporter cell lines or immunofluorescently labelled cells. What is clearly demonstrated from our work, through brightfield and IF images obtained along discrete time-points during the co-culture period, is the progressive formation of two distinct regions and their structural organization (Fig. 4b and Extended DATA Fig 6e). To ensure language/scientific accuracy, we removed the word "migration", and any statement suggesting the existence of such a migration process, and we state only that the co-culture of CM aggregates with PE/STM/PFH organoids generates a self-organized structure where WT1⁺ cells surround cTnT⁺ cells. The formation of these two distinct regions is now also further strengthened by additional IF images (Fig. 4c and Extended DATA Fig 6h).

4. Generally, given the prior published studies, the term "heart organoid (hHOs)" used in the manuscript is somewhat unspecific and misleading: alternative designations should be considered.

We agree with the reviewer about the lack of specificity of the nomenclature "heart organoids". Thus, we decided to replace the designation "heart organoids" by "epicardium-myocardium organoids" (EMOs), throughout the entire manuscript.

5. From the IF staining of hHO sections in Fig. 2b, it is not clear whether – within the center of the myocardial-like zone (MZ) / myocardial tissue (MT) – there is the formation of a cell-free cavity or rather presence of cTnT negative non-cardiomyocytes?

In order to clarify this doubt, we performed additional IF staining of the hHOs. These results show that in the center of the organoids it is normally observed the presence of some cells that are negative or with a lower staining intensity for the CM marker NKX2.5 (Extended DATA Fig 7a), which may indicate the presence of cTnT negative non-cardiomyocytes at the inner parts of the organoid. We also demonstrated that those inner areas of the organoid do not stain for the apoptotic marker Caspase (Extended DATA Fig 7b), meaning that there is no cell death in the center of the organoid and no cell-free cavity was observed, taking into consideration the presence of cell nucleus (DAPI staining) throughout the entire myocardial region at different sections of hHOs from different depths. These comments can be found in the revised version of the manuscript (Page 10, Line 291-295).

6. The statement at the end of page 7: "These structurally organized hHOs show preliminary evidences that point for the recapitulation of important features already reported regarding the role of epicardium on myocardium proliferation, compaction, and maturation." is only poorly supported by the data presented in fig.2.

We agree with the reviewer that a more thorough characterization of the impact of co-culturing CMs and PE/STM/PFH on CMs maturation should have been performed. In order to address this issue, we performed additional experiments that are now presented in new Figure 4, Figure 5 and Extended DATA Fig 6, 7 and 8.

As can be seen in Fig. 5 and Extended Data Fig. 8, immunofluorescence (IF) staining of hHOs (now entitled EMOs in the new version of the manuscript) sections shows an increased expression of the mature ventricle CM marker MLC2V, mainly near the WT1⁺ epicardial-like layer, compared with age-matched CM aggregates, in which MLC2V staining was not detected.

We also added IF staining for CX43 and Ki-67 markers in CM aggregates that were missing, to highlight the differences compared with hHOs (Extended Data Fig. 7 and 8). As can be seen in Fig. 5, CM aggregates present a lower expression of the ventricular gap junction marker CX43 when compared with hHOs.

Regarding the analysis of proliferative Ki-67⁺ CMs, we performed additional IF staining for this marker in different hHOs, as can be seen in Fig. 4d and Extended DATA Fig 7a. As can be observed in these figures, in hHOs there is a higher number of proliferative CMs near the epicardial-like layer and an almost absence of proliferative CMs towards the center of the organoids. These results are consistent with the characteristic increased CM proliferation observed at early stages of embryonic myocardium growth development near the epicardial layer through paracrine signals release from epicardial cells. On the opposite side, in CM aggregates, it was observed a homogeneous distribution of proliferative CMs throughout the entire aggregate, without a specific distribution pattern (Fig. 4d, control).

We also added new IF images, and the respective higher magnifications, in specific areas of the hHOs, to reinforce our claim that, near the epicardial-like layer, CMs show stronger staining for cTnT and there is a lower nucleus density (Extended Data Fig. 6i), which suggests that CMs present a more compact fibrillar content, indicating a higher structural maturation degree of the CMs in the co-culture system (Page 10, line 284-288)

Finally, we complemented these studies with a functional analysis of hHO and CM aggregates spontaneous contraction profile, with and without drug stimulation (Fig. 5d and Extended Data Fig. 8c). From this functional analysis we observed that hHOs have the expected response to all the tested drugs, with control CM aggregates showing a not so pronounced effect upon verapamil and E-4031 treatment, which may indicate that CM present in control condition lack or present a lower expression level of hERG and Ca²⁺ channels, which may indicate lower functional maturation compared with CMs present in hHOs. Altogether, we believe these results demonstrate a clear impact of the co-culture on myocardium-like cells structural and functional maturation.

7. It was noted that PE/STM differentiation induces a population of CD31⁺/WT1⁻ endothelial-like cells organized in a concentric pattern which may comprise vascular-like cells that are responsive to VEGF stimulation. However, did authors try to further specify these CD31⁺ cell to test their relation to endocardial cell?

According to the reviewer comment, we assessed if the CD31⁺ cells identified in PE/STM/PFH organoids could be related to endocardium, by performing co-staining of NKX2.5 and CD31, since it has been reported that NKX2.5 is transiently expressed in endocardial-like progenitor cells (Mikryukov et al. 2021)(Drakhlis et al. 2021). However, we observed that the CD31 positive cells do not express NKX2.5 which suggests that these CD31⁺ cells may not have an endocardium-like phenotype. These results were not included in the manuscript but if deemed necessary by the reviewer, we may add them.

8. A more "head-to-head" comparison should be provided for hHOs (resulting from mixing CMs with PE/STM/PFH) shown in Fig 2. Versus organoids revealed in Fig 4., which apparently represent the co-emergence of CMs and PE/STM cell populations,

resulting from differentiation protocol modulations (WNT, BMP, RA) on d5-7. Moreover, the Results part downstream of "Removal of WNT signaling activation during PE/STM induction allows the co-emergence of CMs in the multilineage organoids" is difficult to follow and should be re-structured / shortened. What is the designation of organoids presented in Fig. 4 d-f?

The former Fig. 4 and respective results section were extensively reviewed and thus we believe that this part of the manuscript is now easier to be followed by the reader. Regarding specifically the condition in which we only activated RA+BMP4 during PE induction period ("RA+BMP4" condition), it was observed the co-emergence of cTnT positive CMs combined with two distinct populations, WT1⁺/LHX2⁻ and WT1⁻/LHX2⁺ populations. These organoids were designated as posterior second heart field/STM (pSHF/STM) organoids (Extended Data Fig. 10c). In the revised version of the manuscript, we hypothesized that the CMs observed in pSHF/STM organoids may have a more atrial phenotype, whereas hHOs represent a ventricle myocardium-epicardium model. In fact, as explained in the text (Page 8/9, line 228-248), RA activation at the cardiac mesoderm stage has been extensively reported to induce atrial CM specification, which strengthens the hypothesis previously mentioned. Compared with pSHF/STM organoids, hHOs represent a more controllable, reproducible and robust model to study ventricle epicardial-layer formation during embryonic heart development and the impact of epicardium on ventricle myocardial-like tissue maturation. Our main objective when modifying the differentiation protocol during the PE induction period was to shed some light on the relevance of each of the three signals on the specification of PE/STM/PFH organoids and not particularly focused on the development of a heart organoid model. However, we acknowledge that pSHF/STM organoids could be an interesting model to study early SHF progenitor cells specification and/or epicardium-myocardium interaction, particularly in an atrial context, through prolonged time in culture.

9. In the last Result paragraph, authors noted: "To evaluate the impact of the new progenitor cell population (CB condition) on PE/STM and CM specification, the outcome of PE/STM induction and CM commitment starting from this new cell population was evaluated (Fig. 6a)." It remains unclear to this reviewer what the expression "new progenitor cell population (CB condition)" means. This should be better defined.

We agree with reviewer that the expression "new progenitor cell population (CB condition)" is not clear. To make the message more explicit, on page 15/line 479-482 we modified the text from "To evaluate the impact of the new progenitor cell population (CB condition) on PE/STM and CM specification, the outcome of PE/STM induction and CM commitment starting from this new cell population was evaluated (Fig. 6a)." to "To evaluate PE and CM specification from D5 progenitor cell population, after CHIR and BMP4 treatment between D1-D3 (Fig. 6k), FC and IF analyses at D11 of differentiation were performed."

10. Given the relative complexity of the paper and the high number of cell lineages (including progenitor and intermediate stages) and tissues, the introduction of a table maybe useful to systematically structure the paper. This should include: Designation of the several differentiation strategies and designating of resulting organoid models, definition of the respective lineage phenotypes and their characteristic marker(s) combination(s), indication of putative lineage counterpart(s) established in embryogenesis.

We agree with the reviewer that one or more "wrap-up" illustrations may greatly facilitate the job for the readers. For this, in this new version of the manuscript, we introduced additional illustrations throughout the figures of the manuscript to schematize the different culture protocols that were employed throughout the studies (Fig. 1a; Fig. 4a, Fig. 6k). Moreover, we added a new Figure (Extended Data Fig. 10) that summarizes the different tissue models/organoids that were obtained and the respective signalling manipulation strategies that were performed throughout the work.

11. In the Discussion, authors noted: "The concept of multilineage organoid has been also recently introduced in the cardiac field by Drakhlis and colleagues that reported an organoid recreating both heart and foregut development (Drakhlis et al., 2021). Although the early stages of differentiation, consisting of a simple WNT signaling modulation, are similar to our protocol, the lack of an additional step for PE/STM specification generated an organoid featuring instead anterior foregut and myocardial-like layer. Those organoids do not present a clear organization or representation of what is described in vivo for PE, STM and

posterior foregut/hepatic-like cells, contrarily to the present PE/STM/PFH organoids that represent a well-defined and *in vivo*-like organized structure that resemble PE/STM and posterior foregut development." Actually, the publication by Drakhlis et al., 2021 reveals the structures formation and spatial distribution of STM-like cells in neighborhood to posterior foregut endoderm (PFE) representing liver anlagen outside a myocardial layer, thereby closely recapitulating native embryonic patterns. This should be more correctly referred in the submitted manuscript.

We agree with the point raised by the reviewer and as such we re-phrased the mentioned paragraph to more correctly refer the embryogenesis aspects studied by Drakhlis et al., 2021 (Page 18, Line 559-566)

In general, the Result and Discussion part is somewhat lengthy, substantially redundant and needs curtailment. Moreover, the paper would benefit from a model integrating the key findings.

Authors noted in the discussion: "Characterization of PE/STM/PFH organoids, through single cell transcriptomic analysis, may help to decipher in more detail the different mesoderm-derived subpopulations within PE/STM/PFH organoids and identify new reliable markers that can allow the precise characterization of each sub-types of mesoderm-derived cells." The lack of scRNAseq to confirm (and better defined) the identity of individual cell lineages is indeed a major shortcoming of the manuscript.

To comply with these more general reviewer comments, we re-structured the entire manuscript and re-wrote some sections to make the message clearer, more focused and less redundant. We also added additional data that strengthens the more important conclusions of our studies. We also added a new figure presenting a model integrating the main findings of the manuscript (Extended Data Fig. 10).

We understand the comment of the reviewer and plan to use scRNAseq in future work, beyond the studies performed in this manuscript, to help us to provide a better definition of the different cell populations present in the PE/STM/PFH organoids. In this new version of the manuscript, we tried to compensate for the absence of the scRNAseq and provide robust information about the cell populations by performing additional IF studies. More specifically, we added additional IF images for the co-staining of WT1 and LHX2 to clarify the presence of WT1⁺/LHX2⁻ cells at the periphery of the organoids and the presence of LHX2⁺/WT1⁺ cells more prevalent towards the epithelial structure. We believe that these studies reinforced our claim that the mentioned organoids contain PE-like and STM-like cell populations. Adding to that, we think that is also valid to highlight that the allegation of the existence of a PE and STM cell population in PE/STM/PFH organoids is also corroborated by their *in-vivo*-like spatial organization. In fact, the spatial arrangement of these two populations, combined with the presence of posterior foregut/hepatic diverticulum-like structure, is in agreement with what is described for PE, STM and PFH during embryonic development in mouse.

In addition to these IF studies, we performed additional experiments to provide a more detailed characterization of the impact of WNT, BMP4 and RA on the generation of PE/STM/PFH organoids. We believe that altogether this new data provides a robust demonstration that we are generating an organoid platform that, upon modulation, can recreate the interface between heart, PE/STM and liver bud embryonic structures (Fig. 3 and Extended DATA Fig 4 and 5).

Minor:

Typo in the introduction: second heat field (SHF)

We corrected this typo.

Figure 3d: pls double-check designation of IF staining; indication of the DAPI channel deems partially incorrect

We checked and rectified those typos throughout the manuscript.

Reviewer #3 (Remarks to the Author):

In this manuscript the authors describe a new 3D organoid model of pro-epicardium, septum transversum mesenchyme, and posterior foregut/hepatic cell (PE/STM/PFH) development. They describe a spatial organization of the 3 cell types similar to embryonic development in vivo. Using this model, the study provides insights that are suggestive of the development of PE/STM progenitors from posterior second heart field. The specification of these progenitors was found to be driven by RA signaling based on bulk RNA sequencing analysis. To provide a functional application of these PE/STM/PFH organoids the authors co-cultured them with aggregates of cardiomyocytes. These experiments showed that the PE cells can self-organize into an outer layer of epicardial cells and result in the formation of complex heart organoids. The major novelty of this work is the detailed characterization of an organoid that contains the three cell types (PE/STM/PFH). While this is an interesting model the study has a number of limitations that need to be addressed.

Major comments:

1) Organoids are great models of development and the representative images shown in the paper demonstrate the overall 3D structure as summarized in the little schematics for example in Figure 1. However, the authors fail to show how reproducible their organoid system is. It is well known that a huge variation of efficiency in organoid formation exists, depending on the system used. The manuscript would greatly benefit from additional data demonstrating how robust the organoid formation is. In other words, how many aggregates are developing the gradient of WT1 expression with PE cells in the outer periphery, followed by STM cells that are engulfing an inner gut epithelial lumen. The authors could address this in several ways e.g. i) by showing images of 10 independently generated organoids; ii) by quantifying the expression gradients of WT1, LHX2, E-CAD throughout the organoids as done in Supp. Fig 2d; and iii) by providing data on the percentage of organoids that form a lumen, and their general shape (round, ovoid, elongated) and size in the horizontal and vertical axis. Excellent examples for this can be found in recent organoid/gastruloid publications from Rossi et al. Cell Stem Cell 2021, PMID: 33176168, Figure S1 and S2; and Moris et al. Nature 2020, PMID: 32528178, Figure 1e, Extended data Figure 1).

To highlight the reproducibility of PE/STM/PFH organoids between biological experiments and within the same experiment, we added images of 8 independently generated organoids from 4 independent biological replicates showing the similarity between different organoids (Extended DATA Fig 2a). The quality criteria were defined as the (1) presence of one lumen/epithelial structure, normally at a more central position in the organoid (highlighted in Extended DATA Fig 2a with white arrows), (2) percentage of $\pm 70\%$ WT1 positive cells (Fig. 1b), and (3) presence of a gradient of WT1 expression inside the organoids, from the periphery towards the epithelial structure (Extended DATA Fig 2f, n = 14 PE/STM/PFH organoids derived from 3 separate differentiation experiments).

In addition to analysing the reproducibility of the organoids across distinct differentiation experiments, we also analysed their reproducibility across different lines of human pluripotent stem cells (PSC), both embryonic and induced. In Extended DATA Fig 1b we demonstrated the reproducibility of WT1⁺ cell induction for two additional hPSC lines. We added also IF data that demonstrate the reproducibility in terms of the structural organization observed in PE/STM/PFH organoids for these two additional hPSC lines (Extended DATA Fig 3).

2) The authors rely heavily on WT1 and LHX2 as markers for PE vs STM.

a. In Figure 1 staining for WT1 and LHX2 is shown on two different organoids and the conclusion is made the WT1⁺ cells are localized in the outer rim while LHX2⁺ cells are expressed in the inner layer surrounding the epithelial gut-like structures. Given that this a central finding of the paper the authors should show this more convincingly by developing a co-staining of WT1 and LHX2 that allows to show clear overlap / separation of the two markers.

The point raised by the reviewer is very pertinent. To address this comment, we performed co-staining of LHX2 and WT1 that indeed demonstrates the separation of the two markers (Extended DATA Fig 2g) with LHX2 being expressed near the epithelium and the WT1 more intensively expressed near the periphery of the organoid. Furthermore, in Extended DATA Fig 3, that shows the same IF staining for organoids derived from two additional hPSC lines, it is also evident the separation of the two markers.

This is not only relevant for the data presented in Figure 1 but throughout the manuscript. For example, in Figure 4d consecutive sections are stained for WT1 and LHX2 and the conclusion is made that "IF staining revealed the presence of LHX2+/WT1- cells surrounded by a thin layer of WT1+/LHX2- cells (Fig. 4d I, I', II, II' and Extended Data Fig. 4a I, I')". Based on the figures provided it is difficult to appreciate that. In fact, it looks like the WT1+ and LHX2+ cells are overlapping.

The former Fig. 4 and respective results section have been extensively changed and rewritten to be easier to follow. Regarding the specific comment raised by the reviewer, we prolonged the time in culture of organoids obtained for the "RA+BMP4" condition until D17 of differentiation, where it is now more clear the presence of WT1+ cells lining the periphery of the organoids and that do not co-stain with LHX2 (Fig. 3f and g), and the presence of a distinct LHX2+/WT1- cell population (Fig. 3f and g and Extended Data Fig. 5c and d). The WT1 positive mesothelium lines the cTnT positive areas but also the LHX2 cell population (Extended Data Fig. 5d).

b. Using only 2 markers to distinguish PE and STM cells is not ideal. The authors mention additional PE (UPK3B) and STM (HLX1) markers in the manuscript. Adding stainings for these additional markers would strengthen the presented data.

We understand that using these two markers only, may not be ideal. Although we presumed the existence of STM and PE populations based on immunostaining for LHX2 and WT1, respectively, we additionally demonstrated a significantly increased gene expression of other STM and PE markers, including HLX1 and UPK3B, for the PE protocol in relation to the CM protocol, which point to the emergence of these two populations in our organoids (Extended Fig.2h). We are aware that LHX2 and WT1 are not specific markers for the STM and PE populations, but the combined generation of the two different mentioned sub-populations, in an in vivo-like organization (as previously explained to reviewer 1), also corroborates our claims concerning the characterization of the PE/STM/PFH organoids.

3) To show the functionality of the PE/STM/PFH organoids the authors choose a co-culture system with cardiomyocytes and created a heart organoid. On page 7 the authors make the conclusion that: "PE/STM/PFH cells have the capacity to self-organize and migrate, generating a WT1+-epicardial-like layer surrounding a cTnT positive myocardium like tissue (Fig. 2f)". With the current experimental setup this conclusion cannot be made. In order to make conclusions about the contribution of the PE/STM/PFH cells to the heart organoid the PE/STM/PFH organoids should be generated from a fluorescently labelled cell line (e.g. RFP) that can be lineage traced. With the current dataset it can't be excluded that the WT+ epicardial cells seen in the heart organoids were derived from cells of the cardiomyocyte cultures via inductive signals provided by the PE/STM/PFH cells.

We agree with the reviewer that our experimental setting does not allow to infer about the precise mechanism by which the WT1+ cells are generated. Thus, to address this comment, we removed the statement claiming that PE/STM/PFH cells have the capacity to migrate (page 9, line 274) and form this layer of WT1+ cells since we agree that we do not present enough evidences to prove that. In the revised manuscript we only state that the co-culture of the PE/STM/PFH cells with the CM aggregates allows the generation of hHOs (now EMOs) showing the WT1+ cells surrounding the cTnT positive CM core without concluding by which mechanisms the epicardial cells lining the myocardium were generated.

In addition, this heart organoid dataset would benefit from a more detailed display of reproducibility along the lines of the first comment.

To address this comment, we added brightfield images of 10-12 independently generated hHOs from 3 independent differentiation experiments (Extended DATA Fig 6e), where it is evident, despite the intrinsic heterogeneity associated to organoid models in terms of organoid size and shape (Extended DATA Fig 6f), the formation of the two different regions: (1) the outer epicardial-like layer and (2) the core myocardium-like tissue. The existence of this structural organization was defined as the quality criteria. We also presented a representative brightfield image of hHOs that failed to recapitulate the previously mentioned quality criteria (Extended DATA Fig 6g). Although we do not present data in the manuscript regarding this matter,

we found that the ratio between PE/STM/PFH cells and CMs is extremely important to reproducibly obtain the described arrangement of the cell populations (layer of WT1+ cells surrounding the myocardium-like tissue). In fact, as explained on page 261-263 of the manuscript, we concluded that the optimal ratio of PE/STM/PFH per CMs cells is 90/10 (although 85:15 also works) while for increasing percentages of PE/STM/PFH cells, the cell arrangement is not observed. Moreover, it was also found that the time point to perform the co-culture is critical. In fact, we concluded that the reproducibility is high if the co-culture re-aggregation is performed when the expression of ITGA4 in PE/STM/PFH organoids is maximal (Extended DATA Fig 6b) and also when CMs aggregates are already contracting and expressing VCAM. Within the same biological run, if the process of re-aggregation works well, the reproducibility between organoids is high, meaning that all the organoids meet the quality criteria.

4) The authors then go on to show that the heart-organoids show signs of cardiomyocyte maturation. While the EM data presented in Extended Data Figure 3d-e is promising more analysis is required to convincingly show a matured cardiomyocyte phenotype.

a. The authors state that the cardiomyocytes show "more compact phenotype, with denser fiber content, and CMs appeared also more aligned and following the curvature of the hHOs." This can't be appreciated from the low magnification images presented in figure 2. The authors should consider providing higher magnification images and to quantify the fiber content and fiber alignment.

b. Markers for trabecular (NPPA, BMP10) vs compact (Hey2, MYCN) cardiomyocytes have been established (Funakoshi et al. Nat Commun 2021, PMID: 34039977). The authors should use these markers to demonstrate a compact vs trabecular phenotype of the cardiomyocytes in the heart organoids.

c. To further show a more mature phenotype IHC for the marker of immature cardiomyocytes TNNI1 and the marker of mature cardiomyocytes TNNI3 should be performed and quantified. Mature cardiomyocytes should have an increase in the TNNI3/TNNI1 ratio compared to immature controls (Guo et al. Cir Res 2020, PMID: 32271675).

We agree with the reviewer that a better characterization of the impact of CM and PE/STM/PFH co-culture on CM maturation should have been performed. To address this issue, we added new data in new Figure 4, Figure 5 and Extended DATA Fig 6, 7 and 8.

Regarding CM maturation, as can be seen in Fig. 5 and Extended Data Fig. 8, IF staining of hHOs sections shows an increased expression of the mature ventricle CM marker MLC2V, mainly near the WT1+ epicardial-like layer, compared with age-matched CM aggregates. In which concerns to compaction, we added new IF images, and the respective higher magnifications, for areas of the hHOs that reinforce the statement that CMs, near the epicardial-like layer, show stronger staining for cTnT and lower nucleus density (Extended Data Fig. 6i), which suggests a more compact fiber content.

Regarding the "more compact phenotype" analysis, we performed additional IF staining for the proliferative marker Ki-67 in different hHOs (Fig. 4d and Extended DATA Fig 7a). As can be observed in these figures, in hHOs there is a higher number of proliferative CMs near the epicardial-like layer and an almost absence of proliferative CMs towards the center of the organoids. These results are consistent with the characteristic increased CM proliferation observed at early stages of embryonic myocardium growth development near the epicardial layer and promoted through paracrine signals released from epicardial cells. Likewise, in CM aggregates, it was observed a homogeneous distribution of proliferative CMs throughout the entire aggregate, without a specific distribution pattern (Fig. 4d, control).

5) The authors further show that the epicardial cells induce proliferation in the out layer of cardiomyocytes in their heart organoids (Fig. 2c). Have the authors checked whether the reduced cardiomyocyte proliferation in the core of the heart organoid is caused by cell death due to lower perfusion of nutrients to the core of the organoid. Additional data providing the size of the heart organoids to address perfusion concerns as well as TUNEL staining to check for cell death in the core of the organoid would be useful.

In order to address the two points of this comment, we immunostained hHOs for apoptotic cells using an antibody to detect caspase-3 activity and we also quantified hHOs diameter at D15 of co-culture. Although, at this time point, these organoids already presented an average diameter of 680 μm (Extended DATA Fig 6f), we did not observe appreciable cell death in the center of the organoids (Extended DATA Fig 7b).

6) As part of their analysis of the progenitors that are giving rise to the PE/STM/PFH, the authors activate BMP and WNT signaling between day1 and 3 and performed bulk RNAseq expression analysis comparing control and extended BMP+WNT treatment (CB condition). The bulk RNAseq data revealed interesting changes in expression including a shift towards posterior second heart field markers (pSHF), reduction in endoderm markers and reduction in cardiac mesoderm markers. This part of the manuscript would highly benefit from analysis on the single cell level for example by flow cytometric analysis of the proportion of pSHF cells, endoderm progenitors and cardiac mesoderm cells. Alternatively, single cell RNA-sequencing could be considered. This analysis on the single cell level would allow a more conclusive demonstration of the enrichment of a pSHF progenitor population.

7) The authors then go on to analyze the developmental potential of the new "pSHF enriched" progenitor population generated under the CB condition. They report a reduced potential for cardiomyocyte differentiation and an increased potential for PE/STM differentiation. While this is an interesting observation the presented data does not warrant the statement made in the abstract that "evidences for a posterior second heart field/splanchnic mesoderm origin of the PE/STM" are provided in the manuscript. To this end a lineage tracing strategy should have been employed.

I agree that the authors have developed a powerful model to analyze the early developmental stages of PE/STM development. However, their experiments in this part of the manuscript fall short of demonstrating this. The manuscript would be a lot more impactful if pSHF progenitors could for example be enriched by sorting for a surface marker, and subsequent analysis of the positive and negative sorted populations for their potential to develop into PE/STM cells. If a clear enrichment of PE/STM cells from the positive sorted pSHF progenitors vs negative sorted progenitors could be shown, this would provide a clear demonstration of their developmental origin. Alternatively, lineage tracing through fluorescent reporters for the pSHF lineage could be used to address this question. Lastly, single cell RNA-sequencing at multiple timepoints throughout the organoid development could also be considered as a powerful tool to reconstruct the development and progenitors of PE and STM. Either of these 3 approaches would be highly beneficial to improve the impact of the manuscript.

We thank the reviewer for comments 6) and 7), which raised very pertinent questions. We agree that the analysis that we performed is not robust enough and does not provide strong "evidences for a posterior second heart field/splanchnic mesoderm origin of the PE/STM". Thus, we re-wrote the manuscript in order to tone down our speech regarding this claim (Abstract and Page 3/Line 77-83) and we reduced the emphasis that was initially provided to this topic in the initial version of the manuscript. In fact, our emphasis was/is more on understanding if the manipulation during the first days of the differentiation could allow the generation of a PE organoid that do not present the endoderm-derived posterior foregut/hepatic epithelium. We believe that we were capable of demonstrating, by flow cytometry, that CHIR+BMP4 treatment between D1-D3, significantly reduced the percentage of endoderm progenitors at D5 (Extended DATA Fig 9b) and also compromised the structural organization observed in PE/STM/PFH organoids with the absence of the epithelial structure (Figure 6n). Additionally, we also observed that the mentioned treatment compromised CM specification, but did not compromise WT1+ cells generation, which suggested a decreased cardiac mesoderm progenitor cell population and/or a favoured PE/STM progenitor cell specification. To broaden our insight on the extent to which this system replicates developmental processes, we performed bulk RNAseq data that pointed for a decreased expression of CMs progenitor markers but also for an enrichment of the posterior region of SHF and STM related splanchnic mesoderm genes. Although we did not progress further with a deeper characterization of the developed subpopulations of progenitor cells and consequent specific derivations, as suggested by the reviewer, we believe that the presented data comprise a valuable and novel information that opens the path for further studies by the scientific community.

8) On page 12, 3rd paragraph the authors draw the conclusion that RA signaling drives the specification of the LMP progenitor to the pSHF phenotype : " induce up-regulation of the RA pathway, which may originate the subpopulation of early paraxial mesoderm progenitors, and consequently drive the LPM progenitor cell population from cardiac mesoderm to a posterior splanchnic/SHF mesoderm specification". Beyond showing the upregulation of enzymes regulating RA signaling (ALDH1A2) and RA target genes (NR2F2, several HOX genes) no further experimental proof for this statement is provided. To clearly show the impact of RA signaling on the specification of the pSHF in their model system the authors should consider inhibiting RA signaling (culture in RA & Retinol free media, or use of small molecule inhibitors such as BMS 493) and activating RA signaling (all-trans-Retinoic acid) during the day1-5 time window.

We performed additional experiments to address this question. We presented data showing that the direct activation of RA between D1-D3 of differentiation affects CM specification at a similar level compared with the condition for which we performed CHIR+BMP4 supplementation within the same time frame (Extended Data Fig. 9d), which reinforces the hypothesis raised from RNA-seq data that pointed for RA as one of the stimuli behind the observed differences in CM and PE specification from D5 and D5CB progenitor cells.

9) On page 9 the authors state: "these results demonstrate that inhibition of the WNT pathway during this specific stage of differentiation compromise the WT1+ lineage ..." This conclusion cannot be made from the presented data because a WNT inhibition was not experimentally tested. Instead, the WNT activation by CHIR was removed which would have resulted in presence of endogenous WNT signaling levels. Endogenous levels are not the same as complete inhibition of WNT signaling. The authors should therefore consider rephrasing this statement to for example: Endogenous WNT levels are not sufficient to support development of the WT1+ lineage. A similar refinement of the statement on top of page 10 "that the elimination of WNT signaling between D5-D7 ..." is required.

We thank the reviewer for this comment. We considered the suggestion and we re-wrote the mentioned statement (Page 7/8, Lines 202-252).

10) On page 9, bottom the authors state: "This observation highlights the fact that the LHX2 and WT1 positive cells observed in the condition combining BMP4 and RA toward PE/STM induction are mainly dependent on the RA signal for their specification, reinforcing again the role of RA on the generation of a PE/STM population." The authors never tested the effect of BMP + RA treatment vs BMP only treatment. All experiments that manipulated RA signaling were done in the presence of CHIR. Therefore, the statement should be rephrased to state that RA signaling is important in the presence of BMP and CHIR signaling.

We thank the reviewer for the comment. We tested and presented data for the mentioned conditions (BMP + RA treatment vs BMP only) in new Figure 3 and Extended Fig. 4. This result section was intensively re-structured and re-wrote to clarify the impact of BMP4, RA and CHIR manipulation.

Minor comments:

1) On page 7, bottom of first paragraph the authors state: "absence of vascular cells in control CM aggregates (Extended Data Fig 3g,h). The CM aggregates shown in the figure clearly show CD31+ signal. Therefore, this statement needs to be revised.

We agree with the reviewer and we changed this statement to a more accurate one (Page 10, Line 301)

2) In Figure 2e the co-staining of WT1 and CD31 is not clearly visible. It would help if the authors could provide higher magnification insets for this dataset.

To address this issue, we added Figure 4e II where co-staining of WT1 and CD31 can be clearly seen in a higher magnification inset.

3) In Extended Data Figure 3c the authors show CX43 staining and make the conclusion that the staining is "more pronounced in the region contacting the EL". This conclusion is not supported by the provided images and a quantification of CX43 staining throughout the heart organoid should be considered, if the authors want to make this statement.

We added additional IF images of CX43 staining in hHOs (Extended DATA Fig 8b) and also additional IF staining images for CX43 for the control condition (CMs aggregates) (Figure 5a control) to highlight the differences between both conditions. However, since we did not perform quantification of CX43 staining, we decided to remove the sentence "more pronounced in the region contacting the EL".

4) In Figure 3c the cell types that are present at day 5 are analyzed using flow cytometry. In the manuscript it is stated that the PDGFRa+ mesoderm cells make up 66% of the total cells. In Figure 3c the PDGFR bar seems to be at ~75%. Please clarify this discrepancy.

The reviewer is correct. Thus, we rectified the data presented in the manuscript (Page 7, Line 194) that is now in agreement with the data presented in Figure 2.

5) On page 9, 2nd paragraph the authors are providing values for WT1+ cells in CHIR and BMP4 vs CHIR only supplementation: "Removal of RA supplementation from D5-D7, revealed a significant decrease in the percentage of WT1+ cells, more accentuated in the condition with combined CHIR and BMP4 (53±5% WT1+) compared with only CHIR supplementation (63±4% WT1+)" The actual baseline values of WT1+ cells that are present in conditions with RA supplementation are not provided to the reader. Please add these control values in brackets as well e.g. (75±5% vs 53±5% WT1+).

We followed the suggestion of the reviewer and added the mentioned values in the manuscript (Page 7/8, Lines 202-252).

6) On page 9, bottom the authors refer to the endothelial cells that are present in the heart organoid as "myocardial-like endothelial cells". No data is provided to provide proof that these endothelial cells have a myocardial phenotype. The authors should therefore consider removing the "myocardial-like" wording and just refer to the cells as endothelial cells.

We followed the reviewer suggestion and thus we removed the "myocardial-like" wording (Page 7/8, Lines 202-252).

7) On page 10, 2nd paragraph the authors introduce the prolonged low level WNT activation combined with BMP4 from day1-day3. It is not immediately apparent to the reader that the initial 11uM CHIR step from day0-1 was kept for both conditions. I recommend including a small schematic of the protocol on top of Figure 5 similar to what the authors did for Figure 6. In addition, the naming of the conditions as CB and C further contributes to the confusion. A clearer labelling for example as C-CB and C-0 condition would be helpful.

To address this comment, we re-phrased the text (Page 15, Line 479-482) to make clearer that the step of WNT activation from D0-D1 is maintained in both conditions. We also added a scheme illustrating both the WB and control protocols to be easier to understand the protocol employed for the two tested conditions (Fig. 6k).

8) On page 10 the authors state: " However, in D5CB condition we did not observe increased expression of the early somite progenitor markers PARAXIS (TCF15) and FOXC2 (Loh et al., 2016),...". The data supporting this statement was not provided. Please include the data in the extended data section.

This information is presented in Supplementary Table 1, which includes the differential expression analysis performed with the RNA-seq data.

9) The methods section does not specify what size of AggreWell plates were used. Therefore, the information is not sufficient to recapitulate aggregate formation with the same cell densities. Several studies have shown that the number of cells per aggregate critically impact the outcome of organoid formations. The authors should therefore add this important information.

We agree with the reviewer that this is an important technical information to allow recapitulation of the aggregate formation. We use Aggrewell 800 plates, information that is specified in the material and methods section (Page 21, Line 660).

10) Also related to the aggregate formation the schematic in Figure 1a would benefit from additional detail that shows that these cultures were performed in aggregates. For example, a little "comic" of the cells that shows the culture format throughout the protocol below the existing schematic would be helpful.

To address this comment, we re-formulated the illustration in Figure 1a. We believe it is now clearer the culture conditions that were used.

11) A couple of datasets / figures are missing statistical analysis. For example, Figure 1c + Extended data Figure 1 a-c and Figure 6b should include statistical comparison between the displayed experimental conditions.

The reviewer is correct. We added the statistical analysis on the mentioned Figures.

12) Some of the in-text references to Figures are not correct. For example, on page 5, first paragraph Extended data Fig. 3b – should be Fig 3a. There are a couple of these mismatches throughout the manuscript. The authors should go through the manuscript and make sure that all the references are correct.

To address this point we have checked all the mentioned figures in the text to make sure all of them are now in agreement with the respective Figure legends.

13) The same nomenclature of ', " is used to indicate staining of consecutive sections (Figure 4d) as well as insets into images (Figure 1e). I recommend using different symbols to indicate insets vs consecutive sections to avoid confusion.

We followed the suggestion made by the reviewer. In the revised manuscript the symbols “ ’ ” are used to indicate staining of consecutive sections whereas “ * ” is now used to refer insets.

REVIEWER COMMENTS

Reviewer #1 (Remarks to the Author):

The revised manuscript improved substantially which has made the manuscript easier for the readers to follow. The authors have provided substantial amount of additional data that support their conclusion of the paper. Although most of the concerns were addressed with these data, there are still several weaknesses to be clarified.

Major concerns

- Fig5a: Quantification of MLC2V and CX43 is needed, especially CX43, since it is unclear if CX43 is significantly increased in EMOs compared to controls with the representative images. It is also misleading to use "white" for EMOs (Fig5a III) and "green" for controls (Fig5a IV).

- Fig5d & Fig 8c: Verapamil is a calcium blocker and decreases heart rate (as is the result in "Maddah et al., 2015" that the authors refer to) but the Fluo-4 intensity after drug treatment (yellow line) is showing increase in heart rate and decrease in time between peaks. Authors should elaborate further in the difference they see in their system compared to others.

Additionally, the summary of these data, "Collectively, these results indicate that EMOs present a functional and improved response to known drug (Line 342)" is not supported with a statistical comparison. Does any of the data comparing Fig5d & Fig 8c have significant difference?

Minor concerns

- Fig3, 4 etc: Abbreviations O1, O2, O3, O4 is used throughout the new images, but is not defined in the figure legends. Are they "organoid No.1"... etc? Does O1, O2, O3, O4 each come from different cell lines?

- Fig3c, 3d: It would be easier for the readers to interpret if the condition of the culture "CHIR" is inserted to the BF images as in Fig3f.

- Fig3g: Mislabeled, as "RA" in O3 where it should be "RA+BMP4"

- Fig4d: Lacking the names of the proteins stained in "Control". Assuming that it is cTnT (Red) & Ki-67 (Green) but the authors also have NKX2.5 in the same panel and it is not stated in the Figure legend. Additionally, Ki-67 and NKX2.5 should be quantified as in Extended data Fig 2f, if the authors are to conclude "whereas these cells were almost absent towards the core of EMOs (line 290)"

Reviewer #2 (Remarks to the Author):

Generally, I'd like to thank the authors for comprehensively replying to all questions raised. Still, a number of topics requires further attention:

- The authors have added additional data to prove reproducibility of their method. However, a statistic outlining the success rate/ efficiency of PE/STM organoid generation for each cell line should be generated e.g. a respective bar chart. This analysis should be based on how many organoids are generated e.g. per experiment and how many of those fulfill the quality criteria defined by authors?

- It was found that the endothelium does not have an endocardial phenotype. These data should be included in the Extended Data figures and mentioned in the main text.

- For generating the EMOs: What was the rationale behind dissociating the CM and PE/STM aggregates

and not simply co-culture them (e.g. within a hydrogel) to see if proepicardial cells would migrate to the CM aggregates and cover them (similarly to the approach of Hofbauer et al.)? Have the authors tried this? Performing additional experiments on that is not expected by this reviewer; however, if there was a rationale behind the dissociation strategy, this should be discussed.

- On the one hand, it is fascinating how epicardial cells form a layer around the MZ in EMOs. However, this layer is not a single-cell layer as it appears in vivo and has been achieved in vitro, e.g. in cardiac organoids from the Hudson group (Mills et al., PNAS, 2017). This should be outlined in the discussion.

- In their long-term culture experiments, the authors describe two types of resulting organoids (Extended Data Fig. 8d). What is the percentage of organoids of each type developing per experiment? Is there an hypothesis on why one or the other type of organoid develops and how this may be controlled in future work? No additional experiments are required here but the authors should write a couple of sentences about that in the discussion.

Reviewer #3 (Remarks to the Author):

The authors did address most of the comments and issues raised which did improve the quality of this revised manuscript. However, there are still some issues remaining that need to be addressed.

1) The maturation and compaction of the cardiomyocytes in the EMOs is still not sufficiently shown. The authors did opt to remove the term "compact myocardium" from this revised version of the manuscript in response to the reviewer comments. However, they are still using the terms "compact" and "compaction" in the abstract (line 17), main text (line 288 and line 361) and discussion (593). To justify the use of these terms a compact cardiomyocyte phenotype needs to be more clearly demonstrated using established markers in the field. Specifically, the authors should test whether a compact cardiomyocyte phenotype adjacent to the epicardial layer and a trabecular phenotype in the core of the EMO are present. As pointed out before this could be achieved by antibody staining, RNAscope or in situ hybridization for compact markers (Hey2, MYCN) and trabecular markers (NPPA, BMP10).

2) The authors did add staining for the ventricular marker MLC2V to show a more mature ventricular phenotype in the EMO vs control CM aggregates. While this new data shows a clear difference between the EMOs and the CM aggregates, MLC2V is rather marking a specific cardiomyocyte subtype than a mature cardiomyocyte phenotype. In directed ventricular differentiation protocols MLC2V staining can be detected as early as day20 of differentiation at a timepoint at which the myocytes are still very immature. To clearly demonstrate the maturation effect in the EMO, staining for additional maturation markers that are used in the field should be performed. As suggested before this could for example be achieved by staining for TNNI1 and TNNI3 in EMO and control CM aggregates.

3) New data is provided (Extended data Fig. 9d) to support the statement that RA signaling between d1-d3 of differentiation restricts cardiac mesoderm specification. However, the authors also claim that this RA treatment induces a PE/STM mesoderm (abstract, line 19-20). No data is provided to show that RA treatment between d1-d3 does result in an increased proportion of WT1+/LHX+ cells at day 11. The finding that RA signaling does promote PE/STM mesoderm specification is an important novelty of this study and should be clearly shown.

4) Functional data to support a more mature cardiomyocyte phenotype in the EMOs has been added to the revised manuscript (Fig 4d and Extended Data Figure 8c). The authors state: "arrhythmic behavior was observed only in EMOs after E-4031 stimulation" (line 330). This statement is not supported by the data. The calcium transients shown in the figures suggest arrhythmogenic behavior in both EMOs and Controls CM aggregates. Specifically, the distance between each of the calcium transients is not

stable = arrhythmic in both EMOs and Control CM aggregates following E-4031 treatment. Authors should rephrase this sentence to state that arrhythmogenic behavior was seen following E-4031 treatment for EMOs and Controls. Adding a short explanation that E-4031 is a hERG channel blocker would help the readers understanding these experiments better. Similarly, the authors should consider introducing that isoproterenol is beta-adrenergic agonist.

Minor:

- 1) In Figure 4d the labels in the bottom panel overlap.
- 2) In Figure 5d and Extended Data Figure 8c the label of the bar graph should read "BPM" not "BMP".
- 3) Some of the arrows in the schematics in Extended Data Figure 10 c, d are missing a label.

REVIEWER COMMENTS

Reviewer #1 (Remarks to the Author):

The revised manuscript improved substantially which has made the manuscript easier for the readers to follow. The authors have provided substantial amount of additional data that support their conclusion of the paper. Although most of the concerns were addressed with these data, there are still several weaknesses to be clarified.

Major concerns

- Fig5a: Quantification of MLC2V and CX43 is needed, especially CX43, since it is unclear if CX43 is significantly increased in EMOs compared to controls with the representative images. It is also misleading to use "white" for EMOs (Fig5a III) and "green" for controls (Fig5a IV).

We agree with the reviewer that quantification of CX43 is needed to clearly demonstrate that EMOs have an increased expression of this marker in comparison with the control condition. We performed and added the requested quantification in Extended DATA Fig 8c, which demonstrates that EMOs have a higher prevalence of CX43 positive cells when compared with the control condition. Regarding MLC2V staining, we think that the presented IF images in Fig. 5a I and II, Fig. 5a control and Extended DATA Fig 8a clearly demonstrate that the staining of MLC2V in control CM aggregates is practically inexistent compared with EMOs, which shows a clear staining. Regarding the staining of CX43 in "white" in Fig. 5a III, we changed to "green" to be coherent with Fig. 5a IV.

- Fig5d & Fig 8c: Verapamil is a calcium blocker and decreases heart rate (as is the result in "Maddah et al., 2015" that the authors refer to) but the Fluo-4 intensity after drug treatment (yellow line) is showing increase in heart rate and decrease in time between peaks. Authors should elaborate further in the difference they see in their system compared to others. Additionally, the summary of these data, "Collectively, these results indicate that EMOs present a functional and improved response to known drug (Line 342)" is not supported with a statistical comparison. Does any of the data comparing Fig5d & Fig 8c have significant difference?

We appreciate the reviewer comment. However, there are few studies that evaluate verapamil effect at the level of calcium transients. In fact, the majority of the studies are focused on action potential profiles. Therefore, the effect of verapamil on Ca²⁺ transients is not well documented in the literature, which precludes an accurate comparison of our data with data from others. Although in our system we observed an increased frequency of contraction, contrarily to the study by Maddah et al., 2015 in which it was observed a decreased frequency of contraction, a decreased duration of the Ca²⁺ transient, which is the most accepted outcome upon verapamil stimulation, is coherent with the results in Maddah et al., 2015 work.

When considering the 3 independent biological experiments that were performed for each drug and for each condition, we did not find statistical significance for the differences observed between EMOs and CM aggregates in the case of verapamil exposure; however, it was clear that in the case of E-4031 exposure the arrhythmic behaviour was only observed in EMOs (see answer to reviewer 3). Nevertheless, we changed the text to "Collectively, these results suggest that EMOs may present a functional and improved response to known drugs."

Minor concerns

- Fig3, 4 etc: Abbreviations O1, O2, O3, O4 is used throughout the new images, but is not defined in the figure legends. Are they "organoid No.1"... etc? Does O1, O2, O3, O4 each come from different cell lines?

We added the meaning of the abbreviations O1, O2, O3, O4 in Fig. 3, 4, 5, and in Extended Fig. 5, 6 and 7.

•Fig3c, 3d: It would be easier for the readers to interpret if the condition of the culture "CHIR" is inserted to the BF images as in Fig3f.

We sympathize with the reviewer comment but we decided to not combine Fig. 3c and 3d, as it is in Fig. 3f, because we wanted to present a higher number of BF and IF images to clearly highlight the points that were raised for the "CHIR" condition. Considering those, we believe the current arrangement is the best to convey the message.

-•Fig3g: Mislabeled, as "RA" in O3 where it should be "RA+BMP4"

The reviewer is correct. We changed "RA" to "RA+BMP4" in Fig.3g O3.

•Fig4d: Lacking the names of the proteins stained in "Control". Assuming that it is cTnT (Red) & Ki-67 (Green) but the authors also have NKX2.5 in the same panel and it is not stated in the Figure legend.

The reviewer is correct. We added the names of the proteins stained in Fig. 4d "Control".

Additionally, Ki-67 and NKX2.5 should be quantified as in Extended data Fig 2f, if the authors are to conclude "whereas these cells were almost absent towards the core of EMOs (line 290)"

We agree with the reviewer and as such we added the quantification of the number of NKX2.5 and KI67 positive cells in the periphery and center of EMOs in MZ (Extended Data Fig. 7b) to strengthen the mentioned statement "whereas these cells were almost absent towards the core of EMOs".

Reviewer #2 (Remarks to the Author):

Generally, I'd like to thank the authors for comprehensively replying to all questions raised. Still, a number of topics requires further attention:

- The authors have added additional data to prove reproducibility of their method. However, a statistic outlining the success rate/ efficiency of PE/STM organoid generation for each cell line should be generated e.g. a respective bar chart. This analysis should be based on how many organoids are generated e.g. per experiment and how many of those fulfill the quality criteria defined by authors?

We agree with the reviewer that this is an important quantification, still missing in the manuscript. We have thus added a bar chart (Extended DATA Fig 2b) providing information on the percentage of organoids that fulfill the quality criteria in each one of the 4 independent experiments, and added the efficiency value in the revised version of manuscript (page 4, line 114).

- It was found that the endothelium does not have an endocardial phenotype. These data should be included in the Extended Data figures and mentioned in the main text.

We agree with the reviewer. To comply with this comment, we added the mentioned IF staining in Extended Data Fig. 2e and we provided the information collected from those studies in the revised version of the manuscript (Page 4, Line 122-123).

- For generating the EMOs: What was the rationale behind dissociating the CM and PE/STM aggregates and not simply co-culture them (e.g. within a hydrogel) to see if proepicardial cells would migrate to the CM aggregates and cover them (similarly to the approach of Hofbauer et al.)? Have the authors tried this? Performing additional experiments on that is not expected by this reviewer; however, if there was a rationale behind the dissociation strategy, this should be discussed.

The point raised by the reviewer is very pertinent. In fact, we tried to perform first the fusion between CM aggregates and PE/STM organoids without dissociation and further re-aggregation. However, by using this methodology, the model that we obtained did not recapitulate the formation of the epicardial-like layer. This is probably because by using this methodology it was not possible to control the proportion of PE/STM cells and CMs,

which is a critical parameter, since the PE/STM organoids are bigger than CM aggregates at D11. In fact, we found that the epicardial-like layer is only formed when using an optimal proportion of 10-15% of PE cells to 90-85% of CMs. Additionally, we also thought about dissociating PE/STM organoids and re-aggregate the cells inside 96-well plates to obtain smaller and size-controlled organoids that would then respect better the proportion of 10-15% of PE cells to 90-85% of CMs. However, the latter would still include an additional technical limitation that of putting one CM aggregate in each well of the 96-well plate already containing the re-aggregate PE/STM organoid. Overall, the selected strategy was the one that, according to our experience, was technically the simplest to perform and the one that allowed achieving, in a more reproducible and in-vivo like way, a model that recreates epicardium-myocardium interaction. To comply with the reviewer comment, we added a sentence in the revised version of the manuscript raising this point and advancing the respective discussion (Page 18, Line 558-561).

- On the one hand, it is fascinating how epicardial cells form a layer around the MZ in EMOs. However, this layer is not a single-cell layer as it appears in vivo and has been achieved in vitro, e.g. in cardiac organoids from the Hudson group (Mills et al., PNAS, 2017). This should be outlined in the discussion.

The reviewer raised a very interesting point that we had also thought about. Although there are areas in the EMOs where the epicardial cells form a multiple cell layer, there are also regions where epicardial cells form a single-cell layer around the MZ. It is also important to highlight that normally the regions where we found more than one layer of WT1⁺ cells surrounding the MZ, overlap with the regions in epicardial layer (EL) where it is observed a prevalence of a CD31 positive vascular network of cells. Interestingly, although most of the reported studies describe the epicardium as an uniform single layer of epithelium covering the myocardium, as mentioned by the reviewer, a study that characterized the human epicardium during embryonic heart development (Risebro et al., 2015) found differences between atrial and ventricular epicardium and showed that the ventricular epicardium contains regions of multiple cell layers near blood vessels. This may justify the arrangement that we observed in our EMOs. To comply with this interesting observation and the comment raised by the reviewer, we incorporated the discussion above in the revised version manuscript (Page 18, Line 563-569).

- In their long-term culture experiments, the authors describe two types of resulting organoids (Extended Data Fig. 8d). What is the percentage of organoids of each type developing per experiment? Is there an hypothesis on why one or the other type of organoid develops and how this may be controlled in future work? No additional experiments are required here but the authors should write a couple of sentences about that in the discussion.

The long-term culture experiments were performed 3 times (3 independent experiments) with 8-14 organoids for each run. The ratio of dense- vs cavity-like organoids after 60 days of co-culture (CC) was 60% for dense-like morphology and 40% of cavity-like morphology. Between day 15 and 60 of CC, the morphology of organoids evolve a lot and all of them up to day 30-40 show a dense structure, with the cavity-like morphology only appearing at later time points of CC for some organoids. Thus, we reason that these organoids, in the culture conditions provided, have a tendency to evolve to the cavity-like structure with prolonged time in culture, but some of them take longer to achieve that morphology. To comply with the reviewer comment, we added these data and the respective discussion in the revised version of the manuscript (Page 11, Line 333-336).

Reviewer #3 (Remarks to the Author):

The authors did address most of the comments and issues raised which did improve the quality of this revised manuscript. However, there are still some issues remaining that need to be addressed.

1) The maturation and compaction of the cardiomyocytes in the EMOs is still not sufficiently shown. The authors did opt to remove the term "compact myocardium" from this revised version of the manuscript in response to the reviewer comments. However, they are still using the terms "compact" and "compaction" in the abstract (line 17), main text (line 288 and line 361) and

discussion (593). To justify the use of these terms a compact cardiomyocyte phenotype needs to be more clearly demonstrated using established markers in the field. Specifically, the authors should test whether a compact cardiomyocyte phenotype adjacent to the epicardial layer and a trabecular phenotype in the core of the EMO are present. As pointed out before this could be achieved by antibody staining, RNAscope or in situ hybridization for compact markers (Hey2, MYCN) and trabecular markers (NPPA, BMP10).

We understand the reviewer argument regarding the term “compact myocardium”. However, when we mention “compact MZ” or “compact myocardium”, we only want to highlight that we observe, within the MZ, that the CM region that is near the periphery is denser compared with the center/core of the organoid. We do not intend to state that we are observing compaction and trabeculation in our model. In fact, we simply want to describe an observation, similar to what Drakhlis et al did also in their work when they stated “HFOs recapitulate patterns of early cardiomyogenesis. At d10, HES3 NKX2.5-eGFP-derived HFOs consisted of a compacted, NKX2.5-eGFP-positive myocardial layer (ML) enclosing an eGFP-negative inner core (IC). The ML was further covered by an OL of more loosely appearing NKX2.5-eGFP-positive and -negative cells (Fig. 1b)”. Thus, to comply with the point raised by the reviewer, we suggest changing the term “compact” by “denser”, in case the reviewer agrees that this would describe more accurately our observations, without performing additional immunostaining with additional markers. However, if the reviewer considers that the inclusion of markers specific for compact myocardium is mandatory, then we may perform additional IF staining for the HEY2 marker.

2) The authors did add staining for the ventricular marker MLC2V to show a more mature ventricular phenotype in the EMO vs control CM aggregates. While this new data shows a clear difference between the EMOs and the CM aggregates, MLC2V is rather marking a specific cardiomyocyte subtype than a mature cardiomyocyte phenotype. In directed ventricular differentiation protocols MLC2V staining can be detected as early as day20 of differentiation at a timepoint at which the myocytes are still very immature. To clearly demonstrate the maturation effect in the EMO, staining for additional maturation markers that are used in the field should be performed. As suggested before this could for example be achieved by staining for TNNI1 and TNNI3 in EMO and control CM aggregates.

We agree with the reviewer that additional maturation markers should be evaluated and thus we added additional IF staining for the marker cTnI in EMOs. These new data clearly demonstrated an increased staining for this marker in EMOs compared with CM aggregates, as can be observed in Extended Data Fig. 8b. These observations were added to the revised version of the manuscript (Page 10, Line 305-307).

3) New data is provided (Extended data Fig. 9d) to support the statement that RA signaling between d1-d3 of differentiation restricts cardiac mesoderm specification. However, the authors also claim that this RA treatment induces a PE/STM mesoderm (abstract, line 19-20). No data is provided to show that RA treatment between d1-d3 does result in an increased proportion of WT1+/LHX+ cells at day 11. The finding that RA signaling does promote PE/STM mesoderm specification is an important novelty of this study and should be clearly shown.

The reviewer is right. We do not show evidences that RA treatment induces PE/STM mesoderm. We only show evidences that RA treatment, between D1-D3 of differentiation, restricts CM differentiation (Extended data Fig. 9d) and does not compromise PE/STM specification. We did not observe an improved PE/STM specification with RA treatment between D1-D3 of differentiation. Thus, to comply with this comment, we changed the abstract text to “Through mesendoderm progenitor cells modulation, we also show evidences that point for retinoic acid signaling as a restrictive stimulus for cardiac mesoderm specification, without compromising induction of PE/STM mesoderm from LPM.”

4) Functional data to support a more mature cardiomyocyte phenotype in the EMOs has been added to the revised manuscript (Fig 4d and Extended Data Figure 8c). The authors state: "arrhythmic behavior was observed only in EMOs after E-4031 stimulation" (line 330). This statement is not supported by the data. The calcium transients shown in the figures suggest arrhythmogenic behavior in both EMOs and Controls CM aggregates. Specifically, the distance between each of the calcium transients is not stable = arrhythmic in both EMOs and Control CM aggregates following E-4031

treatment. Authors should rephrase this sentence to state that arrhythmogenic behavior was seen following E-4031 treatment for EMOs and Controls. Adding a short explanation that E-4031 is a hERG channel blocker would help the readers understanding these experiments better. Similarly, the authors should consider introducing that isoproterenol is beta-adrenergic agonist.

We did not observe arrhythmic behaviour in control CM aggregates after E-4031 exposure, and we do not understand the reviewer comment that suggests that the selected graphic in Extended DATA Fig 8c (E-4031) for the control condition (CM aggregates) shows an arrhythmic behavior. We present below a graphic that shows the TBP, in seconds, for CM aggregates and EMOs after E-4031 exposure to highlight the arrhythmic behaviour in EMOs and the regular transient in CM aggregates after E-4031 exposure. Apart from that, we followed the pertinent reviewer suggestion and added the information that E-4031 is a hERG channel blocker and isoproterenol is a beta-adrenergic agonist in the revised version of the manuscript (Page 11, Line 314 and 316).

Minor:

1) In Figure 4d the labels in the bottom panel overlap.

The reviewer is right. We corrected that mistake.

2) In Figure 5d and Extended Data Figure 8c the label of the bar graph should read "BPM" not "BMP".

The reviewer is right. We changed "BMP" to "BPM" in both figures.

3) Some of the arrows in the schematics in Extended Data Figure 10 c, d are missing a label.

The reviewer is right. We added the missing labels in Extended Data Figure 10 c and d.

REVIEWERS' COMMENTS

Reviewer #2 (Remarks to the Author):

After the second round of revision, authors have properly replied to all key questions raised to their manuscript. I would therefore like to thank the authors for their comprehensive additional work.

Reviewer #3 (Remarks to the Author):

In their 2nd revision the authors have further improved the manuscript. However, there are still three issues that remain and need to be addressed before warranting publication.

1) This reviewer still thinks that the manuscript would benefit from the inclusion of a staining for compact (HEY2) and trabecular markers (NPPA) in control myocyte aggregates and EMOs. This would allow to conclude that the epicardial cells do show expected functional properties including the induction of a compact myocyte phenotype. The authors do propose to use EMOs in the future to model diseases like non-compaction (Discussion, page 20, line 618) which is an exciting avenue, but requires the formal demonstration of the formation of compact cardiomyocytes.

2) To address the maturation status of the cardiomyocytes the authors did add staining for TNNI3 (cTNI) (Extended Data Figure 8b). However, they only provide a staining for one EMO and are not providing the cTNI staining of a control cardiomyocyte aggregate. This is important data that needs to be added to show a difference between the EMO and a myocyte only control aggregate. An additional minor comment, the color legend in the Extended Data Figure 8b in the 2nd panel is not correct and needs to be fixed.

3) The authors still fail to show the direct effect of RA signaling on PE/STM mesoderm specification. Only results for the cardiomyocyte content at day11 are shown after day1-3 RA treatment. What happens to the PE/STM specification under this condition? The authors do show that the D5WB progenitors are not comprised in their potential to give rise to PE/STM (page 15, 475-478). However, this is not providing direct evidence for the effect of early stage RA treatment on PE/STM specification. If the authors want to claim that RA signaling does not comprise the induction of PE/STM mesoderm (abstract line 20), the experimental data for the effects of RA treatment from day 1-3 on PE/STM specification need to be shown. Alternatively, the authors could consider changing their statement in the abstract to better reflect their findings i.e. instead of RA signaling the statement could focus on the findings for WNT and BMP signaling.

4) The authors are providing convincing data in their rebuttal letter, showing that E-4031 treatment is not arrhythmogenic in control CM aggregates. My apologies for misreading the Extended Data Figure 8e in my previous assessment. The graph showing the TBP provided to the reviewer is much clearer than the original calcium traces currently provided in the manuscript. I suggest adding the TBP graph to the supplemental material.

Minor:

1) Page 15, line 473, refers to the wrong figure. It should refer to Extended Data Figure 9d.

REVIEWERS' COMMENTS

Reviewer #2 (Remarks to the Author):

After the second round of revision, authors have properly replied to all key questions raised to their manuscript. I would therefore like to thank the authors for their comprehensive additional work.

We also would like to deeply acknowledge the contribution of Reviewer #2 throughout the process of revisions of our manuscript which we sincerely believe contributed significantly to improve the quality of the paper.

Reviewer #3 (Remarks to the Author):

In their 2nd revision the authors have further improved the manuscript. However, there are still three issues that remain and need to be addressed before warranting publication.

We deeply acknowledge the contribution of Reviewer #3 throughout the entire process of revisions and his commitment in supporting us to improve the quality and the impact of our manuscript. We hope the answers provided below will properly reply to the remaining issues that were raised in this final round of revisions.

1) This reviewer still thinks that the manuscript would benefit from the inclusion of a staining for compact (HEY2) and trabecular markers (NPPA) in control myocytes aggregates and EMOs. This would allow to conclude that the epicardial cells do show expected functional properties including the induction of a compact myocyte phenotype. The authors do propose to use EMOs in the future to model diseases like non-compaction (Discussion, page 20, line 618) which is an exciting avenue, but requires the formal demonstration of the formation of compact cardiomyocytes.

We acknowledge and understand the point raised by the reviewer. However, we believe we have already provided a very extensive and robust structural and functional characterisation of the EMOs, clearly demonstrating the physiological relevance of this model. Thus, for the moment we do not feel it is absolutely necessary to add these particular studies to our manuscript. In the future, we and others will certainly extend this characterization further with more detailed analysis of EMOs directed towards specific applications.

2) To address the maturation status of the cardiomyocytes the authors did add staining for TNNI3 (cTNI) (Extended Data Figure 8b). However, they only provide a staining for one EMO and are not providing the cTNI staining of a control cardiomyocyte aggregate. This is important data that needs to be added to show a difference between the EMO and a myocyte only control aggregate. An additional minor comment, the color legend in the Extended Data Figure 8b in the 2nd panel is not correct and needs to be fixed.

The reviewer is right. The manuscript was lacking the immunostaining for cTNI in control CM aggregates. This important data is now included in the manuscript (Extended Data Fig. 9). We also corrected the colour legend in the Extended Data Figure 8b in the 2nd panel.

3) The authors still fail to show the direct effect of RA signaling on PE/STM mesoderm specification. Only results for the cardiomyocyte content at day11 are shown after day1-3 RA treatment. What happens to the PE/STM specification under this condition? The authors do show that the D5WB progenitors are not comprised in their potential to give rise to PE/STM (page 15, 475-478). However, this is not providing direct evidence for the effect of early stage RA treatment on PE/STM specification. If the authors want to claim that RA signaling does not comprise the induction of PE/STM mesoderm (abstract line 20), the experimental data for the effects of RA treatment from

day 1-3 on PE/STM specification need to be shown. Alternatively, the authors could consider changing their statement in the abstract to better reflect their findings i.e. instead of RA signaling the statement could focus on the findings for WNT and BMP signaling.

The reviewer is right. We do not provide a direct and irrefutable evidence of the effect of early stage RA treatment on PE/STM specification. To comply with this issue, we decided to modify the abstract text that is now focused on our findings about WNT, BMP and RA signalling on PE/STM/PFH organoids specification. Moreover, we also toned down our claims about RA signalling in the introduction (Page 3, Line 78 to 81) and discussion (Page 19, Line 594 to 598) sections. Overall, we believe our manuscript is now more accurate concerning the conclusions about the role of this particular signalling pathway.

4) The authors are providing convincing data in their rebuttal letter, showing that E-4031 treatment is not arrhythmogenic in control CM aggregates. My apologies for misreading the Extended Data Figure 8e in my previous assessment. The graph showing the TBP provided to the reviewer is much clearer than the original calcium traces currently provided in the manuscript. I suggest adding the TBP graph to the supplemental material.

We acknowledge the comment of the reviewer and we agree that the graph showing the TBP is much clearer than the original calcium traces. To comply with this comment, we added the graph with the TBP values after E-4031 exposure for both EMOs and CM aggregates in Extended Data Fig. 9 to clearly demonstrate that E-4031 treatment is not arrhythmogenic in control CM aggregates.

Minor:

1) Page 15, line 473, refers to the wrong figure. It should refer to Extended Data Figure 9d.

The reviewer is right. This mistake is now corrected. We changed the figure reference in that part of the text to "Extended Data Fig. 9d".